# Reverse metabolomics for the discovery of chemical structures from humans

Emily C. Gentry[1,2,3], Stephanie L. Collins[4], Morgan Panitchpakdi[1,2], Pedro Belda-Ferre[5,6], Allison K. Stewart[7], Marvic Carrillo Terrazas[8], Hsueh-han Lu[8], Simone Zuffa[1,2], Tingting Yan[9], Julian Avila-Pacheco[10], Damian R. Plichta[10], Allegra T. Aron[1,2], Mingxun Wang[1,2], Alan K. Jarmusch[1,2,11], Fuhua Hao[12], Mashette Syrkin-Nikolau[13], Hera Vlamakis[10,14], Ashwin N. Ananthakrishnan[15], Brigid S. Boland[16], Amy Hemperly[13], Niels Vande Casteele[16], Frank J. Gonzalez[9], Clary B. Clish[10], Ramnik J. Xavier[10,14,17,18], Hiutung Chu[8,19], Erin S. Baker[7,20], Andrew D. Patterson[12], Rob Knight[5,6,21,22], Dionicio Siegel[1] & Pieter C. Dorrestein[1,2 ✉]

Determining the structure and phenotypic context of molecules detected in untargeted metabolomics experiments remains challenging. Here we present reverse metabolomics as a discovery strategy, whereby tandem mass spectrometry spectra acquired from newly synthesized compounds are searched for in public metabolomics datasets to uncover phenotypic associations. To demonstrate the concept, we broadly synthesized and explored multiple classes of metabolites in humans, including *N*-acyl amides, fatty acid esters of hydroxy fatty acids, bile acid esters and conjugated bile acids. Using repository-scale analysis[1,2], we discovered that some conjugated bile acids are associated with inflammatory bowel disease (IBD). Validation using four distinct human IBD cohorts showed that cholic acids conjugated to Glu, Ile/Leu, Phe, Thr, Trp or Tyr are increased in Crohn's disease. Several of these compounds and related structures affected pathways associated with IBD, such as interferon-γ production in CD4[+] T cells[3] and agonism of the pregnane X receptor[4]. Culture of bacteria belonging to the *Bifidobacterium*, *Clostridium* and *Enterococcus* genera produced these bile amidates. Because searching repositories with tandem mass spectrometry spectra has only recently become possible, this reverse metabolomics approach can now be used as a general strategy to discover other molecules from human and animal ecosystems.

Annotation of the human metabolome is far from complete. A typical untargeted metabolomics study of human-derived samples can annotate about 10% of the data with a structure[5–9]. Therefore, strategies are needed to structurally assess which molecules are found in humans. Structural knowledge is required for the scientific community to propose how molecules are made and their functional role in biochemical pathways and to determine their involvement in disease development and progression. For the past two centuries, compound isolation followed by structural characterization has been the cornerstone of small-molecule discovery, but this is not practical for human

samples. Recently, several methods for the structure elucidation of biological molecules have been developed, including single-molecule atomic force microscopy[10], co-crystallization of small molecules within crystalline lattices[11,12] and methods using crystalline powders[13]. Nevertheless, none of these approaches reach the throughput necessary to obtain structural insights for the thousands of molecules detected by high-resolution mass spectrometers. Many computational strategies exist to match structures to mass spectrometry (MS) data, but predictions have no easy path forward to verify which of the possible matching structures are correct[14–16].

[1]Skaggs School of Pharmacy and Pharmaceutical Sciences, University of California, San Diego, La Jolla, CA, USA. [2]Collaborative Mass Spectrometry Innovation Center, Skaggs School of Pharmacy and Pharmaceutical Sciences, University of California, San Diego, La Jolla, CA, USA. [3]Department of Chemistry, Virginia Tech, Blacksburg, VA, USA. [4]Department of Biochemistry and Molecular Biology, The Pennsylvania State University, University Park, PA, USA. [5]Department of Pediatrics, University of California, San Diego, La Jolla, CA, USA. [6]Department of Computer Science and Engineering, Jacobs School of Engineering, University of California, San Diego, San Diego, CA, USA. [7]Department of Chemistry, North Carolina State University, Raleigh, NC, USA. [8]Department of Pathology, University of California, San Diego, La Jolla, CA, USA. [9]Laboratory of Metabolism, Center for Cancer Research, National Cancer Institute, National Institutes of Health, Bethesda, MD, USA. [10]Broad Institute of MIT and Harvard, Cambridge, MA, USA. [11]Immunity, Inflammation, and Disease Laboratory, Division of Intramural Research, National Institute of Environmental Health Sciences, National Institutes of Health, Research Triangle Park, NC, USA. [12]Center for Molecular Toxicology and Carcinogenesis, Department of Veterinary and Biomedical Sciences, The Pennsylvania State University, University Park, PA, USA. [13]Division of Gastroenterology, Department of Pediatrics, Rady Children's Hospital University of California San Diego, La Jolla, CA, USA. [14]Center for Microbiome Informatics and Therapeutics, Massachusetts Institute of Technology, Cambridge, MA, USA. [15]Division of Gastroenterology, Massachusetts General Hospital, Boston, MA, USA. [16]Division of Gastroenterology, University of California, San Diego, La Jolla, CA, USA. [17]Center for Computational and Integrative Biology, Massachusetts General Hospital and Harvard Medical School, Boston, MA, USA. [18]Department of Molecular Biology, Massachusetts General Hospital and Harvard Medical School, Boston, MA, USA. [19]CU-UCSD, Center for Mucosal Immunology, Allergy and Vaccine Development, University of California, San Diego, La Jolla, California, USA. [20]Department of Chemistry, University of North Carolina at Chapel Hill, Chapel Hill, NC, USA. [21]Center for Microbiome Innovation, Jacobs School of Engineering, University of California, San Diego, San Diego, CA, USA. [22]Department of Bioengineering, University of California, San Diego, San Diego, California, USA. ✉e-mail: pdorrestein@health.ucsd.edu

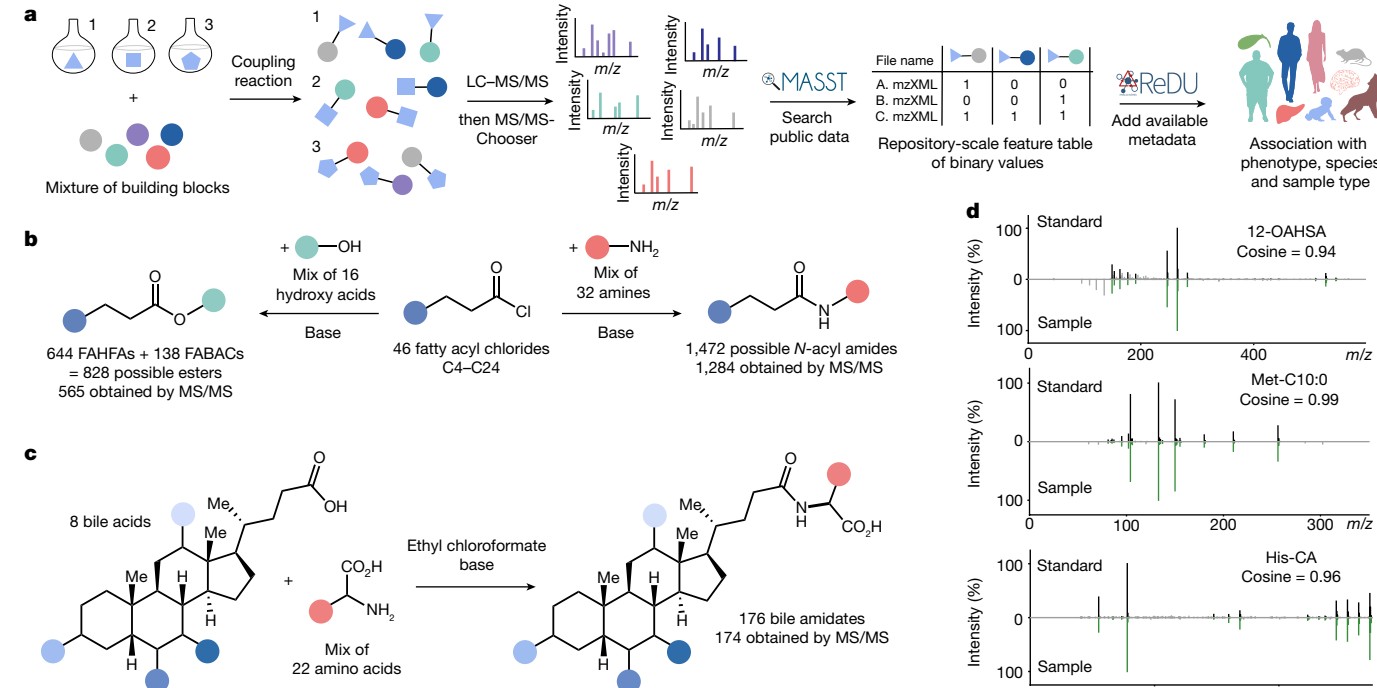

**Fig. 1 | Overview of reverse metabolomics and the synthetic strategies used to obtain standards for MS/MS in this work.** The reverse metabolomics portion starts with MS/MS to associate with sample information (phenotype, species and sample type), whereas the synthesis portion is an approach to obtain the MS/MS spectra. **a**, Workflow for the reverse metabolomics strategy using LC–MS/MS and MASST and ReDU data analyses tools and platforms. **b**, Synthesis scheme for acyl amides and esters. FAHFA, fatty acid ester of hydroxy fatty acid. **c**, Combinatorial bile acid conjugation reaction performed for the discovery of bile amidates. **d**, Representative mirror plots of example MS/MS matches of one of the synthesized standards for each class of molecule with MS/MS data in the public domain.

## Reverse metabolomics

To complement existing structure elucidation strategies and to enrich the biological information available for known or newly discovered molecules, we introduce reverse metabolomics. Reverse metabolomics evaluates whether a specific tandem MS (MS/MS) spectrum can be found in public untargeted metabolomics datasets. Reverse metabolomics allows us to gain insight into phenotypes (for example, health and disease), species and sample types associated with a particular MS/MS spectrum through the mining of metadata associated with each public dataset.

To demonstrate the effectiveness of reverse metabolomics in this work, we first utilized combinatorial synthesis to create complex mixtures of potential metabolites. MS/MS data were then acquired for the compounds in the synthetic mixtures[17]. To assess the capabilities of our reverse metabolomics strategy, we created MS/MS spectra for four metabolite classes of interest: *N*-acyl amides, fatty acid esters, bile acid esters and bile amidates. The presence of these molecules was searched in public untargeted metabolomics datasets[18] using the mass spectrometry search tool (MASST)[1], and the sample information from the available reanalysis of data user interface (ReDU)[2] was summarized (Fig. 1a–d). MASST searches MS/MS fragmentation spectra against 1.2 billion public MS/MS spectra. ReDU enables the analysis and filtering of public data by organism, disease state, phenotype, biospecimen and other metadata or sample descriptors. MS/MS spectra for 2,430 molecules were acquired, and 31% of these spectra were found in human data (level 2 or 3 annotation)[19]. Based on reverse metabolomics analyses, it was possible to propose that there is a strong association between microbial bile amidates and IBD. This association was then validated across multiple IBD cohorts.

Reverse metabolomics offers a distinct approach to investigate biological phenotypes within a specific compound class. Once a new

structure is verified, it becomes integrated into the existing biochemical and metabolic model knowledgebase, which enables the scientific community to formulate hypotheses about biological roles and to identify potential diagnostic biomarkers.

## Acyl amide and ester searches

Acyl amides and esters are important signalling molecules in humans, and many structures have not been fully identified or characterized[20–23]. We synthesized a library of acyl amides and esters through reactions of 46 fatty acyl chlorides with 32 amines and 17 hydroxy acids, respectively, under basic conditions (Fig. 1b; an explanation of key fragment ions are shown in Extended Data Fig. 1). This library was then subjected to a search in public metabolomics datasets. In total, this synthetic scheme theoretically yielded 1,472 acyl amides and 782 acyl esters. We collected MS/MS spectra of the [M+H]+ adduct for 87% and 72% of all desired amides and esters, respectively. We obtained 60,277 spectral matches to the synthesized *N*-acyl amides across 25,463 unique files using MASST, and 31% of the compounds searched returned spectral matches (Fig. 2a and Extended Data Figs. 2 and 3). For the esters, 5,884 spectral matches were found across 5,273 unique data files. However, only 13 esters (2.3%) of those synthesized by MS/MS were detected in public data (Extended Data Fig. 3a–c), which is probably because esters do not ionize well in positive mode. We opted to consistently query the [M+H]+ spectrum to maximize our search space because about 90% of all available public data have been acquired in positive mode. Six of the acyl esters detected in public data were previously undescribed fatty acid bile acid conjugates (FABACs). Although several FABACs have been synthesized for the treatment of gallstones, to the best of our knowledge, they have not been reported in biological samples[24,25].

Next, we used ReDU to investigate where these compounds are found in public data and how they associate with health phenotypes.

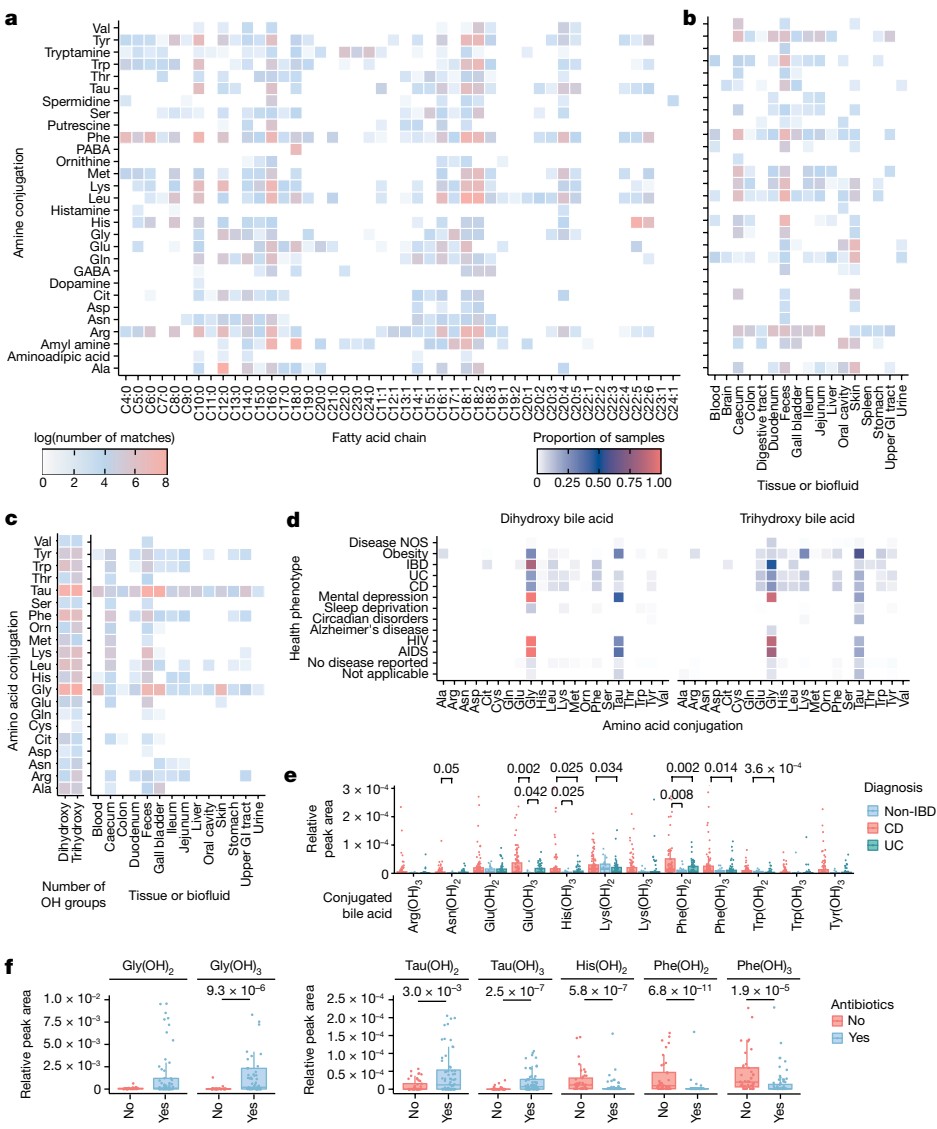

**Fig. 2 | Repository-scale analysis of public MS data. a**, Heatmap representing the log value of spectral matches for 1,472 *N*-acyl amides in the entire GNPS metabolomics repository. **b**, Heatmap showing the log value of unique MS/MS spectral matches for each amine conjugation in different tissues and biofluids using GNPS public data with metadata available in ReDU. GI, gastrointestinal. **c**, Heatmap showing the log value of unique spectral matches for conjugated bile acids across different tissues and biofluids using GNPS public data with metadata available in ReDU. **d**, Heatmap showing the proportion of samples that each MS/MS-synthesized bile acid was detected for health phenotypes across the public metabolomics repositories that have metadata available in ReDU. Disease NOS, disease not otherwise specified. From top to bottom for health phenotype, *n* = 1,556, *n* = 679, *n* = 84, *n* = 144, *n* = 207, *n* = 46, *n* = 713, *n* = 317,

*n* = 195, *n* = 56, *n* = 59, *n* = 13,108 and *n* = 6,092. **e**, Relative MS1 abundance of conjugated bile acids across clinical groups in the public project with MassIVE MS repository accession number MSV000084908 (data collected using an Orbitrap, positive mode). CD, *n* = 103; UC, *n* = 60. **f**, Relative MS1 abundance of conjugated bile acids in relation to antibiotic usage in data from a paediatric IBD cohort deposited as MSV000088735 (data collected using an Orbitrap, positive mode). Antibiotic use, *n* = 72; no antibiotic use, *n* = 45. For **d**–**f**, boxplots show first (lower) quartile, median, and third (upper) quartile and whiskers are 1.5 times the interquartile range. Significance was tested using a pairwise two-sided Wilcoxon rank-sum test. Only *P* values 0.05 or less are shown, which were adjusted using Benjamini–Hochberg correction.

The majority of spectral matches to both the synthetic amides and esters were from animal samples, primarily from humans and mice (Extended Data Figs. 2a and 3b). The synthesized amides were most frequently observed in fecal, caecum and skin samples, whereas esters were observed across many different tissue types (Fig. 2b and Extended Data Figs. 2c and 3b). Some of them were relatively common in specific sample types. For example, 48% of human breast milk samples deposited in ReDU contained at least one synthesized ester, and oleic acid ester of hydroxy stearic acid (OAHSA) was the most frequently observed. *N*-acyl amides were widely detected in microbial cultures (Extended Data Fig. 2b), but only two esters were found in microbial

datasets. MS/MS matches to OAHSA were detected in cultures of *Vibrio mediterranei*. Another MS/MS match, to the palmitoleic acid ester of hydroxy palmitic acid, was detected in a *Leucoagaricus* dataset. *N*-acyl amides were found more frequently in samples labelled as IBD, but the dataset in which they were detected originated from samples collected from a single person monitored for multiple years (Extended Data Fig. 2c). These associations did not hold for either the Crohn's disease (CD) or the ulcerative colitis (UC) categories, which contain data from other IBD cohorts. This result highlights the importance of being able to support an observation across many datasets and large populations. In this way, an important benefit of our method is that

it enables greater confidence when proposing potential biomarkers. This is because all public data are searched and associations are made across multiple datasets of similar or identical phenotype.

## Analysis of bile amidates

First described in 1848 (ref. 26), structural characterization of bile acids has a long history. Bile acids were initially discovered through their role as emulsifiers in fat digestion. They are now known to regulate host immune response and signalling pathways and to play a crucial part in host–microbe interactions[27]. They exist in multiple forms–free carboxylic acids, hydroxyl group conjugates and amide conjugates (called bile amidates)–and all are vital for maintaining host health[28].

Here we focused on bile amidates, the most frequent of which observed are Gly and taurine (Tau) conjugates, although some rare analogues have been described[29–32]. We have previously reported that gut microbes in humans and mice can conjugate cholic acid (CA) to Leu, Phe and Tyr[33]. Other conjugates have been found since then, including Ser conjugated to CA and Phe conjugated to deoxycholate[1]. In silico predictions of conjugated bile acids have also been identified, for which Phe and Trp chenodeoxycholic acid (CDCA) conjugates were validated using standards[34]. Following these discoveries, we proposed that these microbial-conjugated bile acids are part of a much larger class of bile acid amidates that includes other naturally occurring amino acids, both proteinogenic and non-proteinogenic. Therefore, we performed combinatorial amide coupling reactions between 8 dihydroxylated and trihydroxylated bile acids and 22 amino acids and collected MS/MS data for each combination of bile acid and amino acid pair (Fig. 1c). As different instruments can give rise to slightly different MS/MS spectra, we collected MS/MS data on the synthetic mixtures using two of the most common instruments used to obtain public metabolomics data: Orbitrap and quadrupole time-of-flight (Q-ToF) mass spectrometers (Extended Data Fig. 3a). Spectra of these synthesized bile acids had 16,587 matches from 6,603 unique files using MASST compared with 27,337 files that contained either Gly or Tau conjugates (Fig. 2c and Supplementary Tables 1–3). We found MS/MS matches to 145 of the synthesized compounds in public metabolomics data. Of those, to our knowledge, 139 had not been previously described, and approximately 30 of these were reported in bacterial cultures while this paper was under review[35]. Bile acids conjugated to all amino acids except for L-DOPA and Pro were found in public data, although conjugations to Cys and Gln were rare (found in ≤20 files). Of the three non-proteinogenic amino acid conjugations synthesized, we found matches to citrulline (Cit) and ornithine, but not DOPA. It is proposed that other non-proteinogenic amino acid conjugates will be discovered in the future[36]. We recapitulated a 1960s report of an ornithine conjugate in humans, which, to our knowledge, is the only reaffirmation of this reported finding[31]. The most frequently observed amino acid conjugations were those with aromatic rings such as His, Phe, Trp and Tyr, as well as Lys and Ile/Leu, although frequencies of these conjugations varied depending on the type of bile acid. Because it is not possible to perform retention time matching on the repository scale, and because the MS/MS spectra of isomers were relatively indistinguishable, it is only possible to specify whether the amino acids are conjugated to a dihydroxylated or trihydroxylated bile acid when performing a MASST search. Consequently, annotation was at a level 2 or 3 according to 2007 metabolomics standards initiative guidelines[19] (Fig. 1d). However, targeted analysis was performed to confirm the presence of these bile acids in human fecal samples and to determine the absolute concentration of various conjugated and unconjugated bile acids. Seven out of the 15 analysed human fecal samples contained 1 or more of the new conjugated bile acids at concentrations above their well-studied Gly and Tau counterparts (Extended Data Fig. 4b).

In total, 1,742 files, or about one quarter of the files with spectral matches to these new bile acids, were also in ReDU, which has sample information that uses standardized ontologies and vocabularies that we leveraged for reverse metabolomics. Of these, 21% were from human samples. The synthesized conjugated bile acids were detected in fecal material and intestinal tissue, but they were less frequently observed or undetected in the blood and liver, which suggests that these molecules do not enter the enterohepatic circulation, are diluted below detection levels or are modified before they enter the circulation (Fig. 2c). Twelve, mostly aromatic or charged amino acid conjugates, were detected in gall bladder samples from diverse vertebrate species other than humans, mice and rats. We next examined how different conjugated bile acid structures associate with health status in public data. Certain conjugations, specifically those with Glu, Ile/Leu, Phe and Trp, were observed more frequently in all types of IBD samples (described as IBD, CD and UC) relative to those from healthy individuals with no disease reported (Fig. 2d). Samples were also separated in ordination space (Jaccard) by health status based solely on compositional differences of these synthesized bile acids (Extended Data Fig. 4c). Given that these associations are based on presence–absence data from MS/MS matching, we postulated that these bile acids were more likely to be found in patients with IBD compared with healthy individuals, which will lead to MS/MS scans of these molecules being triggered more frequently in IBD samples. To test this hypothesis, we looked at the relative abundance of the new bile amidates in a dataset from patients with IBD and from individuals without IBD[37]. Relative peak areas of several bile acids were higher in samples from patients with CD than in samples from healthy individuals. This result was consistent with the spectral count observations that searched all compatible public data (Fig. 2e). Using a cohort of paediatric patients with IBD, we also examined how the abundance of these bile acids change in response to antibiotic intake. Infants receiving antibiotics presented lower levels of microbial-derived bile acids than host-derived Gly and Tau conjugates, for which the levels were higher in the patients not receiving antibiotics (Fig. 2f). Because these are putative associations, we set out to confirm our findings in two separate human IBD cohorts using an entirely different metabolomics platform. This experiment also served as an independent confirmation of the existence of these bile acids in humans using data that are not part of the Global Natural Product Social Molecular Networking (GNPS), MASST or ReDU ecosystem.

## Validation of the IBD association

Independent validation of the association of these bile acids with IBD was achieved using data and samples from the longitudinal integrative Human Microbiome Project 2 (iHMP2) and the cross-sectional PRISM IBD cohorts[38,39]. In these studies, fecal samples were collected from participants without IBD, with UC or with CD. Fecal extracts were then pooled and analysed with the mixed bile acid standards to match retention times and MS/MS spectra in the same experiment (Fig. 3a and Extended Data Fig. 6a). In the majority of examples, retention time and MS/MS data matched to one single standard out of four possible isomers with the same retention time. However, we cannot exclude the possibility that other isomers that are not yet known (and therefore not tested) could match, which is a general limitation of MS-based metabolomics experiments. Most isomers, such as Glu-CDCA and Glu-conjugated deoxycholic acid (DCA), could be resolved by chromatographic separation (Fig. 3a). However, in a few cases, isomers were not separated by chromatography, such as Met-CDCA or DCA and Tyr-conjugated hyodeoxycholic acid (HDCA) or ursodeoxycholic acid (UDCA). In such cases, both names are indicated (Fig. 3a,b). In total, we observed 63 of the bile acid conjugates in the iHMP2 IBD dataset, 45 of which have relative quantification data (Fig. 3a and Extended Data Fig. 6a) and 19 of which were also detected in the PRISM cohort (Extended Data Fig. 5). In alignment with our previous findings, bile acids conjugated to Cit, Glu, His, Ile/Leu, Phe, Thr, Trp and Tyr were detected in higher abundance

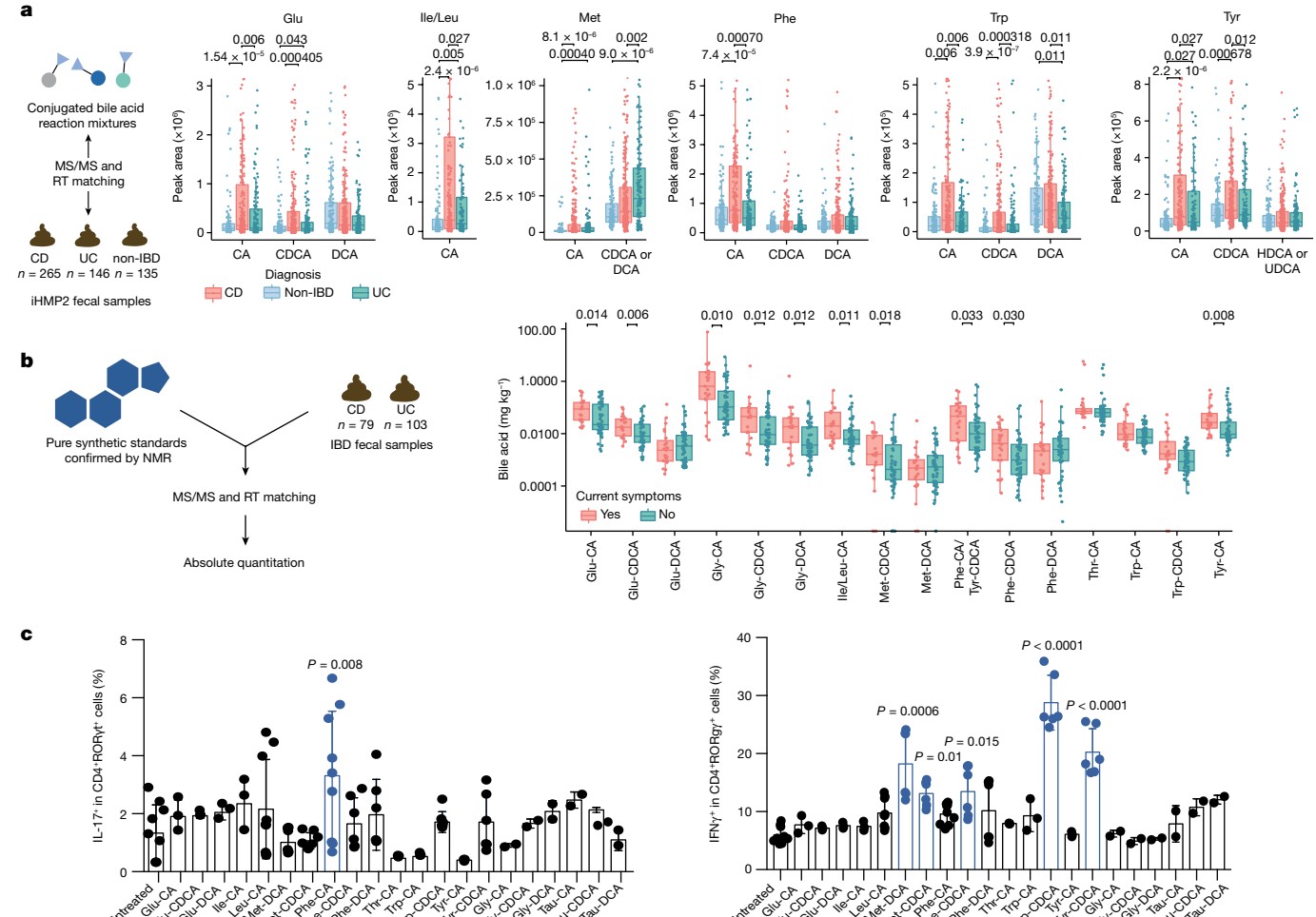

**Fig. 3 | IBD association of new conjugated bile acids. a**, Examples of retention time (RT) matching and MS/MS spectra of a standard to a pooled fraction for which only one isomer matched by retention time (for example, the Glu-CA conjugate) and for which isomers could not be resolved (for example, Tyr-HDCA acid and Tyr-UDCA). This dataset was collected using a Q-Exactive in negative mode (accession number MSV000087562). Relative peak area abundances of selected bile acids that were higher in patients CD (red) and/or in patients with UC (green) compared with individuals without IBD (blue) in the iHMP2 study, as determined by pairwise two-sided Wilcoxon tests. This is a re-analysis of data collected using a Orbitrap, negative mode (Metabolomics Workbench data repository accession numbers PR000639 and PR000677). Boxplots show first (lower) quartile, median and third (upper) quartile with whiskers as 1.5 times the interquartile range. Significance is shown as Benjamini–Hochberg corrected

*P* values. CD, *n* = 265; non-IBD, *n* = 135; UC, *n* = 146 **b**, Concentrations of conjugated bile acids in fecal samples from individuals with active (*n* = 23) or inactive (*n* = 72) Crohn's disease. Values on the *y* axis represent mg of bile acid per kg of fecal matter. This dataset was collected using a Q-TOF in positive mode (accession number MSV000092337). Boxes represent the interquartile range, centre line is the median and whiskers are 1.5 times the interquartile range. *P* values < 0.05 by one-sided Wilcoxon test are shown. **c**, Flow cytometric quantification of IL-17 (left) and IFNγ (right) production in naive CD4[+] T cells from *Foxp3*-hCD2 reporter mice. Cells were treated with 100 μM of bile acids on day 0 and CD4[+] T cells were gated for analyses on day 3. *n* = 6 for controls and *n* > 3 for biologically independent samples for every substrate tested. Bar plot shows mean and error bars represent s.d. One-way Kruskal-Wallis test provided significance.

in patients affected by CD compared with individuals without IBD (Fig. 3a and Extended Data Fig. 6a). Specifically, primary bile acid amides (for example, cholic and chenodeoxycholic amides) were higher in abundance in CD, whereas amides of deoxycholates and related secondary bile acid isomers remained unchanged relative to samples from individuals without IBD. Using an additional separate IBD cohort, we quantitatively examined how the abundance of conjugated bile acids relate to symptom activity in CD compared with UC (Fig. 3b and Extended Data Fig. 6b). Overall, 11 out of 19 bile acids that were quantified were significantly increased in individuals with active symptoms, but only in the CD group. The same trend was not observed for UC. As the observed bile acids may have crucial roles in IBD, we set out to evaluate whether they had biological activity. Therefore, we individually synthesized bile acid conjugates that had the highest abundance in the iHMP2 IBD cohort and tested them for activities in pathways associated with IBD.

As bile acids can be immunomodulatory, one pathway of interest is bile-acid-mediated immune dysfunction. A series of recent studies discovered that CA, CDCA and two secondary lithocholic acid metabolites can affect T cell homeostasis[40,41]. We postulated that some of our lead bile acids may also dysregulate host immune responses; therefore, we tested our synthesized bile acids for immunomodulatory activities. Five of the conjugated bile acids (Met-CDCA, Met-DCA, Phe-CDCA, Trp-CDCA and Tyr-CDCA) had increased levels of interferon-γ (IFNγ) but limited effects on interleukin-17 (IL-17) (Fig. 3c). Notably, Trp-CDCA resulted in about a sixfold increase in IFNγ, a key cytokine involved in the pathogenesis of CD[3].

The pregnane X receptor (PXR), a bile acid nuclear receptor involved in xenobiotic transport and metabolism, is also thought to play a pivotal part in IBD. Specifically, agonists of PXR that are semisynthetic derivatives of rifamycin (for example, rifaximin and rifampicin) have been studied for the treatment of IBD in clinical trials, and decreased

*PXR* expression has been associated with IBD and CD[4,38,39,42–46]. Therefore, as bile acids are also known PXR agonists, we tested the most abundant bile acids in the iHMP2 dataset against human PXR and found that Thr-CA, Glu-DCA and Glu-CDCA act as agonists (Extended Data Fig. 7a,b). These candidate compounds were then added to small intestinal organoid tissue to test their effect on PXR activity. The addition of Glu-CA and Glu-CDCA increased PXR activation and significantly increased the expression of the downstream PXR target gene *Cyp3a11*, even at concentrations as low as 10 μM (Extended Data Fig. 7c). Taken together, these findings support the hypothesis that some of the newly discovered bile acids might have an important role in IBD through PXR and/or immune-mediated processes. We propose that the results could also explain why not all patients respond to rifaximin treatment in clinical studies, as some patients may already have large quantities of PXR agonists produced by their microbiota. More work is needed to fully understand the physiological roles of these bile acids in IBD biology. However, these findings demonstrate that reverse metabolomics can discover new molecules that are biologically active in humans.

## Microbial production of bile amides

Bacteria are involved in the conversion of host-derived bile acids to secondary bile acids. Previous reports from our group have shown that bacteria are able to produce some conjugated bile acids ex vivo[33,35]. Furthermore, the results from the paediatric patients with IBD and treated with antibiotics support a hypothesis that these bile acids might be produced by the gut microbiota (Fig. 2f). To further explore the extent of this microbial conjugation chemistry, we screened 202 isolates from the first Human Microbiome Project[47] for bile acid conjugation capabilities. Isolated from skin, gut and vaginal microbiomes, the bacteria included in this experiment were diverse and spanned 36 distinct genera, including *Bacteroides*, *Bifidobacterium*, *Clostridium* and *Lactobacillus*. Each strain was cultured in duplicate for 72 h in amino-acid-rich medium that included mucin and 500 mg l$^{-1}$ of supplemental bile acid mixture that included CA, DCA, CDCA and their Tau and Gly conjugates (Extended Data Fig. 8a,b).

Liquid chromatography (LC)–MS/MS analyses of extracts from these cultures revealed that *Actinobacterium*, *Bacillus*, *Clostridium* and *Fusobacterium* genera were the main bacterial producers, although some conjugated bile acids were also found in cultures of *Bacteroidia* species. (Fig. 4a–c and Supplementary Table 4). Overall, most bacteria were capable of making several different amino acid conjugations. We detected cholic and deoxycholic amidates for 15 amino acids, which were not detected in control samples (medium controls and extraction controls) or culture samples at the initiation of the experiment (*t* = 0 h; Fig. 4c). More deoxycholic conjugates were detected than CA conjugates. Additionally, precursor ions for other bile acids, namely Asp, Trp and Val conjugates, were detected in cultures with the correct retention times, but were too low in abundance to trigger MS/MS fragmentation and could only be putatively identified without MS/MS spectral matching.

To further characterize bacterial production of conjugated bile acids, we used LC coupled to ion mobility spectrometry MS (IMS-MS), which enables matching of the precursor *m/z*, MS/MS, retention time and drift time against standards run on the same instrument within the same experiment (Fig. 4d,e and Supplementary Tables 5 and 9). With the additional specificity of ion mobility separation, all synthesized isomers within our eight synthetic mixtures, except Ile and Leu conjugates, could be separated. Similar to our LC–MS/MS results in positive ionization mode, we observed a variety of amino acid conjugations in the culture extracts, 18 in total, that matched to CA and DCA. There were also CDCA amidates, specifically those conjugated to the polar amino acids Arg, Asp and Glu. Notably, Arg conjugates were exclusively matched to the CDCA form[35] (Fig. 4e

and Extended Data Fig. 8a). None of the new bile acid conjugates was observed in medium controls or at the *t* = 0 time point of incubation. Gram-positive bacteria belonging to the *Bifidobacterium*, *Enterococcus*, *Clostridium, Cellulosilyticum* and *Catabacter* (recently renamed to *Christensenella*) genera[48], most of which are members of the Firmicutes phylum, produced the most conjugated bile acids in terms of abundance and diversity. These observations align with a previous study[35] that screened 72 microbes for bile acid conjugation and proposed structures of conjugated bile acids, but none of these were verified using standards. We also revealed that Gram-negative *Fusobacterium* can produce conjugated bile acids. Together, these results show that bacteria are involved in more bile acid modifications than was previously appreciated, many of which are now also being discovered in other studies[36,49,50].

## Conclusion

The reverse metabolomics approach resulted in an expansion of metabolic knowledge using simple chemical transformations. Here we targeted three metabolite classes of interest and searched for around 2,000 unique compounds in public data and found some relevant disease associations. This method can be expanded to any synthetically accessible compound class that can ionize and fragment in a mass spectrometer. Our study highlighted the value of public metabolomics data, a growing resource that has begun to double every 2 years. As more data become publicly available with the help of the community, reverse metabolomics will become even more informative. We demonstrated that we could connect newly discovered bile acids to disease phenotypes, something that may eventually have diagnostic value to accurately and non-invasively detect CD. We also found that these bile acids may mediate IBD processes through the regulation of host immune function and PXR signalling. Ultimately, to fully understand human (and other animal) metabolism and to propose functional roles of metabolites we must be able to detect and annotate the majority of molecules. Once the structures of molecules are established and chemical standards are available, then those molecules will become part of the traditional 'forward' direction of metabolomics and quantitative analyses. For molecules that ionize and fragment well by MS, reverse metabolomics will be one of the key tools in the toolbox for structure elucidation and discovery of new host and microbiome-derived metabolites.

Apart from the general limitations of MS already addressed in the text and the Supplementary Methods, metabolites must be present in sufficient quantities or ionize well to be detected and selected for MS/MS acquisition, as reverse metabolomics requires a MS/MS spectrum. Here we addressed this limitation through combinatorial synthesis; however, to use this strategy, the investigator must have some hypothesis with respect to structures of interest. Basic biological building blocks and simple metabolic transformations can lead to millions of candidate molecules, but more complex molecules are more challenging to access, which leads to a much reduced throughput. Other strategies to obtain candidate MS/MS spectra exist, such as computational prediction or directly from a LC–MS/MS untargeted metabolomics experiment, and could be used for MASST searches in reverse metabolomics experiments. We validated our experiments using biological samples (human fecal and/or cultured microbial data) to match to the chromatographic retention times, MS/MS and drift time (if available) of the standards to support the structural annotation output. However, without isolation and NMR or X-ray structural analyses, one cannot exclude the possibility that different isomers are represented by the data. Last, the 1.2 billion spectra that are currently part of MASST searches do not encompass all molecules. This limits re-analyses of public data in the discovery phase of this approach. One approach to address this limitation is to encourage public data deposition, particularly with metadata[2,18].

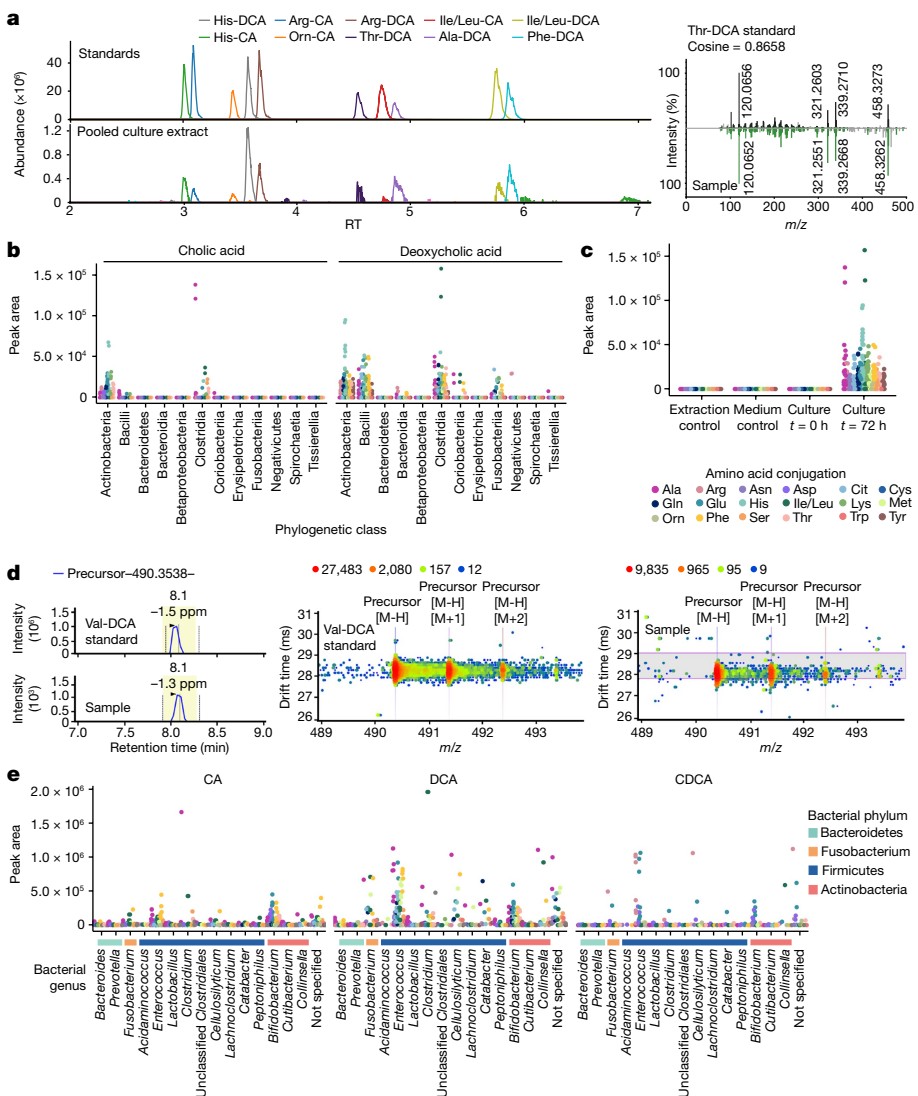

**Fig. 4 | Bile acid conjugations observed in HMP isolates cultured in fecal growth medium containing CA and DCA.** Each strain was cultured in duplicate. **a**, Representative chomatography and MS/MS spectrum for the LC–MS/MS data of microbes cultured with bile acids, in comparison to the synthetic standards. **b**, Scatter plot for positive-mode LC–MS/MS data showing bile acid abundance across phylogenetic classes of bacteria using feature intensities. **b**, Peak area abundance of conjugated bile acids in culture samples at the genus level. Orn, ornitine. **c**, Peak area abundance compared with controls. **d**, Representative retention and drift time for IMS-MS data. **e**, Scatter plots for IMS-MS data collected in negative mode showing bile acid abundance across bacterial genera, organized according to the phylogenetic tree shown and coloured according to their bacterial phylum. Those not specified are unclassified *Lachnospiraceae*.

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

# Methods

## Combinatorial reactions

**N-acyl amides.** In a vial equipped with a magnetic stir bar, a solution containing 33 different amines (0.036 eq., 0.00036 mmol each, 1.2 eq., 0.012 mmol total) with NaOH (1.2 eq., 0.012 mmol) was prepared in 0.1 ml $H_2O$. A solution of acyl chloride (1.0 eq., 0.01 mmol) in tetrahydrofuran (THF; 0.1 ml) was then added in one portion and the mixture was stirred at room temperature for 2 h.

**Acyl esters.** To a vial equipped with a magnetic stir bar, $NaHCO_3$ (1.2 eq., 0.012 mmol) was added followed by a solution of 19 different hydroxy fatty acids or bile acids (0.063 eq., 0.00063 mmol each, 1.2 eq., 0.012 mmol total) in 0.05 ml $CH_2Cl_2$. A solution of acyl chloride (1.0 eq., 0.01 mmol) in 0.05 ml $CH_2Cl_2$ was then added and the reaction was stirred at room temperature for 2 h.

**Conjugated bile acids.** Bile acid (0.125 mmol) was added to a small round-bottom flask equipped with a magnetic stir bar and dissolved in anhydrous THF (2.5 ml). The solution was cooled to 0 °C with an ice bath. Ethyl chloroformate (0.15 mmol, 1.2 eq.) and triethylamine (0.15 mmol, 1.2 eq.) were added sequentially, producing a cloudy white mixture that was stirred in an ice bath for 1.5 h. After that time, a basic aqueous solution of 22 amino acids (0.07 eq., 0.00875 mmol each or 1.54 eq., 0.193 mmol total) was prepared with NaOH (1.5 eq., 0.188 mmol) or $NaHCO_3$ (1.5 eq., 0.188 mmol) in 2.5 ml $H_2O$ then added to the reaction in one portion at 0 °C. The mixture was stirred for 2 h and allowed to gradually warm to room temperature to yield a clear homogeneous solution.

## LC–MS/MS analysis of synthetic mixtures

**N-acyl amides.** The reaction mixture was concentrated in vacuo, then the residue was reconstituted by sonication in 1.5 ml $CH_2Cl_2$–methanol (2:1). An aliquot of 150 μl was transferred from each sample and concentrated in vacuo. The dried extract was resuspended in 1 ml of HPLC-grade MeOH–$H_2O$ (1:1) and diluted for analysis. LC–MS/MS analysis was performed using a Thermo UltiMate 3000 UPLC system coupled to a high-resolution Q-ToF mass spectrometer (Bruker Daltonics MaXis HD). A polar C18 column (Kinetex polar C18, 100 × 2.1 mm, 2.6 μm particle size, 100 Å pore size; Phenomenex) was used as the stationary phase, and a high-pressure binary gradient pump was used to deliver the mobile phase, which consisted of solvent A (100% $H_2O$ + 0.1 % formic acid (FA)) and solvent B (100% acetonitrile + 0.1% FA). The flow rate was set to 0.5 ml min$^{-1}$ and the injection volume for each sample was 10 μl. Following injection, samples were eluted using the following gradient: 0–1.0 min, 5% B; 9.0–11.0 min, 100%; 11.5–14.0 min, 5%.

**Acyl esters.** The reaction mixture was concentrated in vacuo, then the residue was redissolved by sonication in 1.0 ml $CH_2Cl_2$ and diluted 20× for analysis. LC–MS/MS analysis was performed using a Thermo UltiMate 3000 UPLC system coupled to a high-resolution Q-ToF mass spectrometer (Bruker Daltonics MaXis HD). A polar C18 column (Kinetex polar C18, 100 × 2.1 mm, 2.6 μm particle size, 100 Å pore size; Phenomenex) was used as the stationary phase, and a high-pressure binary gradient pump was used to deliver the mobile phase, which consisted of solvent A (100% $H_2O$ + 0.1% FA) and solvent B (100% acetonitrile + 0.1% FA). The flow rate was set to 0.5 ml min$^{-1}$ and the injection volume for each sample was 10 μl. Following injection, samples were eluted using the following gradient: 0–1.0 min, 5% B; 9.0–11.0 min, 100%; 11.5–14.0 min, 5%.

**Conjugated bile acids.** An aliquot of 20 μl was transferred from each reaction mixture and concentrated in vacuo, then the crude residue was resuspended in 2 ml of HPLC-grade methanol and diluted threefold for LC–MS/MS analysis. LC–MS/MS analysis was performed using a Thermo UltiMate 3000 UPLC system coupled to a high-resolution Q-ToF mass spectrometer (Bruker Daltonics MaXis HD) or a Thermo Vanquish UPLC system coupled to a Q-Exactive Orbitrap mass spectrometer (ThermoFisher Scientific). A polar C18 column (Kinetex polar C18, 100 × 2.1 mm, 2.6 μm particle size, 100 Å pore size; Phenomenex) was used as the stationary phase, and a high-pressure binary gradient pump was used to deliver the mobile phase, which consisted of solvent A (100% $H_2O$ + 0.1% FA) and solvent B (100% acetonitrile + 0.1% FA). The flow rate was set to 0.5 ml min$^{-1}$ and the injection volume for each sample was 2 μl or 5 μl for the Orbitrap and Q-ToF, respectively. Following injection, samples were eluted using the following gradient: 0–1.0 min, 5% B; 1.0–1.1 min, 25%; 6.0 min, 70%; 7.0 min, 100%; 7.5–8.0 min, 5%.

**MS/MS data acquisition using Q-ToF.** MS data were collected using a Bruker MaXis Q-ToF with Compass Hystar. For positive and negative ionization, the following settings were used: capillary voltage of 4,500 V; nebulizer gas pressure of 2 bar; ion source temperature of 200 °C; and dry gas flow of 9 l min$^{-1}$. All spectra were collected using data-dependent acquisition with a scan range of 100–2,000 $m/z$, for which the spectral rate was set to 3 Hz and 10 Hz for MS1 and MS2, respectively. The MS/MS collision energies that were used are available in Supplementary Table 4. The five most intense ions per MS1 were selected for MS/MS acquisition, and an active exclusion was enabled and set to release after 30 s and only allow 2 spectra, after which the precursor ions were reconsidered only if the current intensity/previous intensity was >2. Hexakis(1*H*,1*H*,2*H*-perfluoroethoxy)phosphazene (CAS 186817-57-2) was used as an internal calibrant, and lock mass correction was applied on raw.d files in Bruker DataAnalysis. LC–MS/MS data for the synthetic mixtures from the Q-ToF are publicly available and deposited into the MassIVE data repository under accession numbers MSV000089491 (acyl amides), MSV000089493 (acyl esters) and MSV000087522 (conjugated bile acids).

**MS/MS data acquisition using Orbitrap.** MS data were collected using a Thermo Q-Exactive Orbitrap in positive mode using Thermo XCalibur software. Electrospray ionization parameters were set to 53 l min$^{-1}$ for sheath gas, 14 l min$^{-1}$ for auxiliary gas, 0 l min$^{-1}$ for spare gas and 400 °C for auxiliary gas temperature. The spray voltage was set to 3,500 V, the capillary temperature to 320 °C and the S-Lens radio frequency level to 50 V. MS1 data were collected from 150 to 1,500 $m/z$ with a resolution of 35,000 at $m/z$ 200 with 1 micro scan. The maximum ion injection time was set to 100 ms with an automatic gain control (AGC) target of $1.0 \times 10^6$. MS/MS spectra were collected using data-dependent acquisition, for which the top five most abundant ions in the MS1 scan were selected for fragmentation. Normalized collision energies were increased stepwise from 20, 30 to 40. MS2 data were collected with a resolution of 17,500 at $m/z$ 200 with 1 micro scan and an AGC of $5.0 \times 10^5$. All LC–MS/MS data collected using the Orbitrap for our bile acid mixtures are publicly available from the MassIVE data repository under accession number MSV000087523.

## Searching for MS/MS spectra of conjugated bile acids in GNPS public data

Following data acquisition, the raw data were converted into.mzXML or.mzML format using Bruker DataAnalysis or GNPS data converter (https://gnps-quickstart.ucsd.edu/conversion) for Q-ToF and Orbitrap data, respectively. The data were then run through the MSMS-Chooser workflow on GNPS[17] (https://ccms-ucsd.github.io/GNPSDocumentation/msmschooser/) for automatic selection of MS/MS spectra for each conjugated bile acid in the mixture. MSMS-Chooser uses the InChI or SMILES provided by the user to calculate the monoisotopic mass [M] for each compound and then calculate the monoisotopic $m/z$ values of adducts. The algorithm then searches for the calculated $m/z$ values within a 10 ppm mass tolerance window and returns the corresponding MS/MS scan if detected. For some reactions, there

were two possible regioisomeric products. In most cases, the MS/MS spectra for isomers were nearly identical. In these cases, the MS/MS spectra used in the MASST searches were those automatically chosen by MSMS-Chooser, for which the algorithm selects the one with the most abundant precursor ion. The universal spectrum identifier (USI) links for MS/MS spectra of the [M+H]+ adducts were then recorded from the MSMS-Chooser outputs for subsequent analysis. The MSMS-Chooser links to access data for the synthetic acyl amide and acyl ester mixtures can be accessed at https://gnps.ucsd.edu/ProteoSAFe/status.jsp?task=1eb0a498034d46adbca6048b4d2d3b7e and https://gnps.ucsd.edu/ProteoSAFe/status.jsp?task=4b3ecc36162e41e2bf4901c3e8c2f44b, respectively. The MSMS-Chooser links for Q-ToF data acquired in positive and negative ionization mode for the synthetic bile acid mixtures can be accessed at https://gnps.ucsd.edu/ProteoSAFe/status.jsp?task=83b66a72c39040c392a449a5ed1ef4cd and https://gnps.ucsd.edu/ProteoSAFe/status.jsp?task=1ebcf6db4b414c849eaf339699bf0d60, respectively. The link for QE Orbitrap data acquired in positive mode for the synthetic bile acid mixtures can accessed at https://gnps.ucsd.edu/ProteoSAFe/status.jsp?task=988893c4832544c4ab216fa9c0a87d11.

USI links[51] for all MS/MS spectra of [M+H]+ adducts were then used to query against GNPS public data using MASST (https://ccms-ucsd.github.io/GNPSDocumentation/masst/). Links for the USI links and MASST results are provided in Supplementary Table 5. Links for MASST searches of the synthetic amides and esters can be accessed at https://gnps.ucsd.edu/ProteoSAFe/status.jsp?task=4ef6467b2c834e60ad3c332d48c0c460 and https://gnps.ucsd.edu/ProteoSAFe/status.jsp?task=ea86966cf51c4762945ff87e2cdff63c, respectively. Links for MASST searches of Q-ToF data in positive and negative ionization modes for the synthetic bile acid mixtures can accessed at https://gnps.ucsd.edu/ProteoSAFe/status.jsp?task=bfdf8b9aeaea461fab10329b8fb458bf and https://gnps.ucsd.edu/ProteoSAFe/status.jsp?task=74c2ecd7f2b44c13a358a168403648a8, respectively. The link for MASST searches of Orbitrap data for the synthetic bile acid mixtures can be accessed at https://gnps.ucsd.edu/ProteoSAFe/status.jsp?task=5b9b3332c61743bb9944040d3d373415. Results from these public data searches were concatenated and analysed using Excel 16.61.1 and R studio.

MASST searches were performed using the following requirements. First, the precursor mass was filtered with 0.01 or 0.02 *m/z* for QE Orbitrap and Q-ToF data, respectively. To minimize false discoveries, the matches required a minimum of 6 fragment ions and a cosine score of 0.7 to match.

MS/MS matches that met these specified criteria give rise to a false discovery rate of <1% (ref. 52), and matches were further confirmed by manual inspection of mirror plots. MS/MS spectra were also inspected for specific fragment ions diagnostic of their structures. Although this protocol does not guarantee against false discovery of closely related isomers or compounds, we found that it does prevent completely inaccurate identifications. For example, in the case of our ester searches, although we obtained hundreds of MS/MS spectra from the synthetic mixtures, there were only 13 that had matches to public data.

## Targeted LC–MS/MS quantification of conjugated bile acids in healthy human fecal samples

Fecal samples from 15 healthy individual donors (aged 23–65 years, 5 female, 10 male) were purchased from BioreclamationIVT. Approximately 25 mg of fecal sample was extracted with 1 ml ice-cold methanol containing 0.5 μM deuterated internal standards. The following deuterated standards were obtained from Medical Isotopes: glycochenodeoxycholic acid-d4, glycodeoxycholic acid-d4 and glycocholic acid-d4. Samples were homogenized with 0.1 mm silica beads in a Precellys 24 tissue homogenizer for 2 × 20 s at 6,500 r.p.m., cooling on ice between cycles. Additionally, cells were lysed using three freeze–thaw cycles in liquid nitrogen. Samples were centrifuged at 4 °C for 15 min at 14,000*g*, and 200 μl supernatant was transferred to a 11 mm polypropylene autosampler vial.

LC–MS/MS analyses were performed using a Waters Acquity UPLC system coupled with a Waters Xevo TQ-S triple quadrupole mass spectrometer. Chromatographic separation was achieved using an Acquity BEH C8 (2.1 × 100 mm, 1.7 μm) UPLC column (Waters) heated to 60 °C. Samples (1 μl) were injected into the column and eluted with 80% mobile phase A (1 mM acetic acid in 9% acetonitrile, pH 4.15) for 0.1 min followed by a linear gradient to 35% mobile phase B (1:1 acetonitrile to isopropanol) over 9.15 min, 85% B over 2.25 min, and 100% B over 0.3 min. 100% B was held for 0.6 min followed by a linear gradient to 10% B over 0.1 min and held for 2.5 min, for a total of 15 min. The flow rate was modified throughout the run starting with 300 μl min$^{-1}$ for 9.25 min, then increasing to 325 μl min$^{-1}$ over 2.25 min and 500 μl min$^{-1}$ over 0.6 min. The flow rate was held at 500 μl min$^{-1}$ for 0.3 min, then reduced to 300 μl min$^{-1}$ over 0.4 min and held for 2.2 min. MS analyses were carried out using electrospray ionization in separate runs of negative ion and positive ion mode. The following negative ion mode settings were used: capillary voltage of 1.5 kV; cone voltage of 60 V; source temperature of 150 °C; desolvation temperature of 600 °C, desolvation gas flow of 1,000 l h$^{-1}$; and cone gas flow of 150 l h$^{-1}$. The following positive ion mode settings were used: capillary voltage of 3.5 kV; cone voltage of 60 V; source temperature of 150 °C; desolvation temperature of 300 °C; desolvation gas flow of 540 l h$^{-1}$; and cone gas flow of 150 l h$^{-1}$. Conjugated bile acids were identified by MRM transitions from parent [M-H]− ions to daughter [M-H]− ions corresponding to the amino acid fragment. In negative ion mode, collision energies for each bile acid were 35 eV (Glu-CA/CDCA/DCA, Met-CDCA/DCA, Phe-CDCA/DCA, Trp-CA/CDCA), 40 eV (Gly-CA/CDCA/DCA, Ile-CA, Leu-CA, Phe-CA, Tyr-CA/CDCA) or 60 eV (Tau-CA/CDCA/DCA). In positive ion mode, collision energies for each bile acid were 30 eV for Thr-CA. Peak areas for each conjugated bile acid were normalized to peak areas from deuterated internal standards of Gly-CA-d4, Gly-CDCA-d4 or Gly-DCA-d4, depending on the base bile acid structure. Quantification was performed against a standard curve of synthesized standards at 10 concentrations ranging from 0.01 to 5 μM within the same experiment.

## Detection of conjugated bile acids in pooled PRISM and iHMP2 samples

To test the presence of new conjugated bile acids in the PRISM and iHMP2 cohorts we used the C18-neg (negative ion mode analysis of metabolites of intermediate polarity; for example, bile acids and free fatty acids) method originally used in the acquisition process for both studies[38]. We analysed iHMP2 stool QC pool homogenates in tandem with the new bile acid mixtures. In brief, samples and standards were injected onto a 150 × 2.1-mm Acquity BEH C18 column (Waters). The column was eluted isocratically at a flow rate of 450 μl min$^{-1}$ with 20% mobile phase A (0.01% FA in water) for 3 min followed by a linear gradient to 100% mobile phase B (0.01% acetic acid in acetonitrile) over 12 min. MS analyses were carried out using electrospray ionization in the negative ion mode using a Thermo IDX mass spectrometer (Thermo Fisher Scientific) in the full scan analysis over 70–850 *m/z* at 120,000 resolution. Additional MS settings were as follows: sheath gas of 45; aux gas of 5; sweep gas of 1; spray voltage of 3.5 kV; capillary temperature of 320 °C; vaporizer temperature of 300 °C; RF lens of 60; microscan of 1; normalized AGC target of 250; and maximum ion time of 250 ms. MS/MS spectra were generated for all bile acid conjugates and any features matching their retention time and *m/z* in the HMP2 stool QC pool at five different collision energies (10, 20, 30, 40 and 50 V). Data are publicly available from the Metabolomics Workbench repository (www.metabolomicsworkbench.org), where it has been assigned project identifiers PR000639 (iHMP2) and PR000677 (PRISM). The standards were run within the same experiment. Table of masses, ppm errors and retention times of standards and sample and representative data can be found in Supplementary Table 6.

## Processing of the relative quantitative data for PRISM and iHMP2

Relative abundance quantification of stool metabolomics features from two IBD cohorts with 546 samples in iHMP2 (ref. 38) and 220 samples in PRISM[39] are publicly available as supplementary tables within their publications. We extracted values for the peaks from the C18-neg method with compound identifiers that were confirmed to match MS/MS spectra of the new bile acids as described above. Of the 63 detected by MS/MS in these two cohorts, we were able to obtain relative peak area information for 45 conjugated bile acids in iHMP2 and 19 in PRISM. Using the associated metadata that describes the samples as originating from non-IBD, UC or CD individuals, we compared in R studio the relative abundance of each peak in a pairwise fashion between each group using Wilcoxon test and $P$ value adjustment with Benjamini–Hochberg.

## Anaerobic culturing of HMP isolates

In brief, 200 µl of reinforced clostridial medium (BD Difco) was added to each well within a shallow 96-well plate. Then 5 µl of inoculum containing different HMP strains was carefully transferred to the plates using a multichannel pipette, with very gentle mixing. The culture plate was sealed with a breathable film cover (Breathe-Easy) and incubated anaerobically for 48 h at 37 °C taking the optical density every 30 min. After 48 h of incubation, 5 µl was taken out of each well and used to inoculate 200 µl of fecal culture medium made exactly according to a previously published method[53]. Bile salts suitable for microbiology, purchased from Millipore Sigma (48305), were components of this medium. The product lists the composition as about 50% DCA sodium salt and about 50% CA sodium salt. Analysis of the medium controls shown in Extended Data Fig. 8a,b reveal that there are also Tau and Gly conjugates. It was added at a concentration of 500 mg l[-1]. Three plates were prepared in parallel using the above methods. One plate ($t = 0$) was covered with an aluminium sealing film and immediately frozen after inoculation to serve as a baseline sample. The other two plates were sealed with a breathable film cover and incubated anaerobically for 72 h at 33 °C, with the optical density recorded every 30 min. Following incubation, samples were sealed with an aluminium sealing film (AlumaSeal), frozen and stored at −80 °C until extraction. In each of these plates, there were 15 wells that were not inoculated with bacteria, which served as medium controls.

## Extraction of bacterial cultures

Samples were subjected to freeze–thaw cycles three times to lyse the cells. The liquid from each sample (about 180 µl) was then transferred to a deep 96-well plate, and 600 µl of LC-MS-grade methanol was added to each well. The samples were covered and sonicated for 10 min, then centrifuged for 15 min at 2,000 r.p.m. A 200 µl aliquot of each supernatant was transferred to a shallow 96-well plate and the liquid was evaporated in vacuo under centrifugation. Samples were stored dry at −80 °C until LC–MS/MS analysis. The dried extracts were resuspended in a 50% methanol and $H_2O$ solution containing 1 µM sulfadimethoxine as an internal standard, mixed thoroughly using a pipette and sonicated for 10 min. Samples were diluted fivefold for LC–MS/MS analysis.

## LC–MS/MS analysis of culture extracts

LC–MS/MS analysis was performed using a Thermo UltiMate 3000 UPLC system coupled to a high-resolution Q-ToF mass spectrometer (Bruker Daltonics MaXis HD). A polar C18 column (Kinetex polar C18, 100 × 2.1 mm, 2.6 µm particle size, 100 Å pore size; Phenomenex) was used as the stationary phase, and a high-pressure binary gradient pump was used to deliver the mobile phase, which consisted of solvent A (100% $H_2O$ + 0.1% FA) and solvent B (100% acetonitrile + 0.1% FA). The flow rate was set to 0.5 ml min[-1] and the injection volume for each sample was 5 µl. Following injection, samples were eluted with the following gradient: 0.0–1.0 min, 5.0% B; 1.1 min, 25.0% B; 5.0 min, 60.0% B; 5.75–6.5 min, 100.0%; 6.6–7.0 min, 5.0%.

All LC–MS/MS data were obtained in positive mode using the same parameters as specified in the methods for LC–MS/MS analysis of bile acid mixtures. Lock mass calibration was applied for the internal calibrant hexakis(1H,1H,2H-perfluoroethoxy)phosphazene (186817-57-2) and the raw data (.d) was converted to .mzXML format using Bruker DataAnalysis software. The data were pre-processed using MZMine2 (refs. 54,55) and run through the GNPS feature-based molecular networking workflow[18,56]. Details of analysis are provided in the Supplementary Methods. All LC–MS/MS data are publicly available and deposited in the MassIVE data repository (http://massive.ucsd.edu) under accession number MSV000084475. For a pooled sample, the standards were run within the same experiment. The feature-based molecular networking job can be accessed at https://gnps.ucsd.edu/ProteoSAFe/status.jsp?task=1e07d6df47924aec951739fb3695b44a. Supplementary Table 7 compares the relative abundance data across bacterial strains. The ppm values of calculated versus detected and retention time comparisons of standard versus observed and representative traces are provided in Supplementary Table 8. Relative abundance of bile acids were visualized using R studio.

## LC–IMS-MS analysis of culture extracts

LC–IMS-MS analyses of the HMP culture extracts were performed using an Agilent 1290 Infinity UPLC system (Agilent Technologies) coupled to an Agilent 6560 IM-Q-ToF MS instrument (Agilent Technologies). Chromatographic separation was achieved using a Restek Raptor C18 column (1.7 µm, 2.1 × 50 mm) heated to a temperature of 60 °C. Mobile phase A comprised 5 mM ammonium acetate, and mobile phase B was 1:1 methanol to acetonitrile. The LC initially started at 0.5 ml min[-1] and 15% B with isocratic elution for 2 min followed by a stepped gradient going to 80% B over the next 7.7 min (15–35% B over 2 min, 35–40% B over 2 min, 40–50% B over 1.5 min, 50–55% B over 1.1 min, 55–80% B over 1.1 min). The flow rate and mobile phase composition were then increased to 0.8 ml min[-1] and 85% B for an additional 0.8 min followed by re-equilibration at initial conditions for 2 min, resulting in a total run time of 12.5 min. For the bile acid analyses, the dried culture extracts were resuspended in 75 µl of 1:1 methanol–$H_2O$ and 5 µl was injected into the electrospray ionization source and analysed in negative ion mode (capillary voltage of 4,000 V, nebulizer gas pressure of 40 psi, ion source temperature of 325 °C, dry gas flow of 10 l min[-1]). Data were collected from 50 to 1,700 $m/z$ with a IMS drift potential of 17.2 V cm[-1], frame rate of 0.09 frames per s, IMS transient rate of 16 IMS transients per frame, maximum IMS drift time of 60 ms and TOF transient rate of 600 transients per IMS transient. IMS-MS spectra were acquired using MassHunter acquisition software and raw files (.d) were uploaded to Skyline[57] for peak picking and molecular annotation using a library built from the synthesized bile acid mixtures. To consider a microbial conjugated bile acid from the HMP samples a match to the synthetic library, mass accuracy had to be within 10 ppm (although looking at the results, most are within 5 ppm), idotp had to be greater than 0.8 (isotopic distribution match to the MF), RT had to be within ±0.1 min and the drift time in the experimental data had to fall within the window set in Skyline for an IMS resolving power of 40. All LC–IMS-MS data are publicly available and deposited in the MassIVE data repository (http://massive.ucsd.edu) under accession number MSV000087889. The standards were run within the same experiment. The observed drift time values, observed masses for the reference bile acids and reports using Skyline for each sample can be found in Supplementary Table 9. Relative abundance of conjugated bile acids were visualized using R studio.

## Synthesis of the top 15 most abundant bile acids in the iHMP2 IBD cohort

Synthesis of the conjugated bile acids was adapted from a previously described method[58]. Bile acid (0.25 mmol, 1 eq.) was dissolved in anhydrous THF (4.9 ml, 0.05 M) and cooled to 0 °C with stirring.

Ethyl chloroformate (0.3 mmol, 1.2 eq.) was added followed by triethylamine (0.3 mmol, 1.2 eq.) and the reaction was stirred for 1.5 h at 0 °C. After complete conversion of starting material by TLC, a cold solution of amino acid (0.375 mmol, 1.5 eq.) and inorganic base (NaHCO₃ or NaOH, 0.375 eq. or 1.5 eq., respectively) in H₂O (4.9 ml, 0.05 M) was added in one portion. Then, the reaction was stirred for 2 h, gradually warming to room temperature. After this time, THF was removed in vacuo, then 2 M HCl was added to acidify the reaction mixture to pH < 2, producing a white precipitate. The mixture was extracted in dichloromethane (3 × 20 ml), then the combined organic layers were washed with brine (50 ml), dried over Na₂SO₄ and concentrated in vacuo. The crude residue was purified over silica gel using methanol and dichloromethane with 1% acetic acid. Further details are provided in the Supplementary Methods.

## Quantification of conjugated bile acids in IBD fecal samples

Human fecal samples from a clinical IBD cohort ($n = 168$) were collected under University of California San Diego IRB approval number 131487. Informed consent was obtained from all study participants. Fecal samples were weighed into microcentrifuge tubes and homogenized in 50% methanol–H₂O solution with a 1:10 w/v ratio, for 5 min at 5 Hz. The samples were centrifuged at 14,000 r.p.m. for 15 min, then a 200 μl aliquot of each supernatant was transferred to a 96-well plate and dried under centrifugal vacuum. The dried extracts were covered and stored at −80 °C until analysis, at which time the samples were resuspended in 200 μl of 50% methanol–H₂O solution with 1 μM sulfadimethoxine as internal standard.

LC–MS analysis was performed using a Thermo UltiMate 3000 UPLC system coupled to a high-resolution Q-ToF mass spectrometer (Bruker Daltonics MaXis HD). Chromatography conditions were identical to those detailed in the methods for 'LC–MS/MS analysis of culture extracts'. MS1 data were collected in positive ionization mode, and the following settings were used: capillary voltage of 4,500 V; nebulizer gas pressure of 2 bar; ion source temperature of 200 °C; and dry gas flow of 9 l min⁻¹.

Matrix effects were evaluated for the quantification of these bile acids in fecal samples. Commercially available bile acids were purchased from Cayman Chemical and all others were synthesized using the above protocols. Fecal extracts from six patients without conjugated bile acids detected were pooled and mixed, then aliquoted into tubes to make solutions with final concentrations ranging from 0.06 to 1 μM bile acid. Calibration curves were built using linear fit to determine the slope and $y$ intercept. These parameters were used to determine the concentration of each bile acid in the fecal samples, which was then expressed in mg kg⁻¹ using masses of the original fecal samples. Concentrations of these bile acids were visualized using R studio. Exact masses and retention times of bile acids detected in this analysis are provided in Supplementary Table 10.

## Immunomodulatory activity assays

Co-culture of bone-marrow-derived dendritic cells (BMDCs) and naive CD4⁺ T cells was performed. In brief, BMDCs were generated from bone marrow progenitor cells isolated from femurs of wild-type C57BL6/J mice in the presence of 20 ng ml⁻¹ GM-CSF (Miltenyi) in complete RPMI 1640 (10% FBS, 50 U ml⁻¹ penicillin, 50 μg ml⁻¹ streptomycin, 2 mM L-glutamine, 1 mM sodium pyruvate, 1 mM HEPES, non-essential amino acids and β-mercaptoethanol). BMDCs co-cultured with splenic naive CD4⁺ T cells at a ratio of 1:10 (DC:CD4⁺ T cells) and treated with 100 μM of bile acids in the presence of anti-mouse CD3 (2 μg ml⁻¹, eBiosciences), mouse IL-2 (20 μg ml⁻¹, Peprotech), mouse IL-6 (20 ng ml⁻¹, Peprotech), mouse IL-1b (20 ng ml⁻¹, Peprotech), mouse IL-23 (10 ng ml⁻¹, Life Technologies) and human TGFβ (2 ng ml⁻¹, Peprotech). After 3 days of co-culture, cells were stimulated with a cell activation cocktail with brefeldin A (2 μl ml⁻¹, BioLegend) for 5 h and stained for 30 min at 4 °C with either Live/Dead fixable aqua dead stain kit (Life Technologies) at 1:1,000 dilution or with empirically titrated concentrations of the following antibodies:

APC-eF780-conjugated anti-mouse CD4 (clone: RM4-5) at 1:400 dilution, PE-conjugated anti-human CD2 (clone: RPA-2.10) at 1:400 dilution. For intracellular staining, cells were fixed and permeabilized using a Foxp3/Transcription Factor Staining buffer set (Life Technologies) according to the manufacturer's protocol. Intracellular staining was performed using the following antibodies at the specified dilutions: PE-Cy7-conjugated anti-mouse IL-17A (clone: eBio17B7) at 1:200 dilution; BrilliantViolet785-conjugated anti-mouse IFNγ (clone: XMG1.2) at 1:400 dilution; APC-conjugated anti-mouse RORgt (clone: AFKJS-9) at 1:200 dilution; and PE-Dazzle594-conjugated anti-mouse Tbet (clone: 4B10) at 1:200 dilution for 4 h. All antibodies were purchased from Thermo Scientific/eBiosciences, BD or BioLegend. Cell acquisition was performed using a LSR II (BD), and data were analysed using FlowJo software suite (TreeStar). Data were visualized and analysed using Prism v.7.0a.

## Reporter screen for PXR activity

Huh7 cells were seeded into 24-well plates at a density of $2 \times 10^4$ cells per well and cultured in DMEM (Gibco; Thermo Fisher Scientific) supplemented with 10% FBS (Gibco) and 1× penicillin–streptomycin (Gibco) at 37 °C, 5% CO₂. Once cells reached a confluence of about 80%, Lipofectamine 3000 reagent (Invitrogen) was used to co-transfect each well with 100 ng pSG5-hPXR plasmid and 400 ng of pGL4-TK-(DR3)₃ reporter plasmid, containing three copies of the DR3 PXR response element (AGGTCAnnnAGGTCA) (both provided by G.G., University of Kansas Medical Center, Kansas, USA). After 48 h of transfection, cell medium was replaced with William's modified medium E (Gibco) and incubated for 3 h. Cells were then treated with 1% methanol per well (Fisher Scientific) containing conjugated bile acids or rifampicin (Sigma-Aldrich). After 16 h of treatment, the cells were lysed with a freeze–thaw at −80 °C in 100 μl lysis buffer, containing 25 mM Tris-phosphate (pH 7.8), 2 mM dithiothreitol, 2 mM CDTA (1,2-diaminocyclohexane-*N*,*N*,*N'*,*N'*-tetraacetic acid), 10% glycerol and 1% Triton X-100. The luciferase assay was performed with 20 μl of each cell lysate mixed with 80 μl Promega Luciferase Assay system (Promega) in a GloMax 20/20 Luminometer (Promega).

Organoid cultures for *PXR* and *Cyp3a11* expression and isolation of primary intestinal organoids were performed using a modified protocol as previously described[59,60]. In brief, the lower half of the small intestine was removed from PXR-humanized mice in C57Bl6/N background, opened longitudinally and washed with ice-cold PBS. The intestines were diced into 2 mm segments followed by extensive washing to remove contaminants. Then, the intestinal pieces were incubated in Gentle Dissociation reagent (7174, Stemcell Technologies) on an ice platform with gentle rotation for 15 min. The dissociation reagent was removed and intestinal cells were washed with PBS containing 0.1% BSA by pipetting. The washing step was repeated four times and the supernatant collected from each wash and labelled as fractions 1–4. Fractions 3 and 4 containing the intestinal crypts were pooled for use. Crypts were counted, embedded in Matrigel (354230, Corning) and cultured in IntestiCult Organoid Growth medium (6005, Stemcell Technologies). Medium was changed every 3 days. The cells were used for experiments at 7 days after seeding or further passages (no more than three generations). The intestinal organoids were treated with 0.1% DMSO, Glu-CDCA, Glu-CA and rifampicin at 10, 50, 100 and 500 μM prepared in fresh culture medium. Intestinal organoids were collected 18 h after treatment for analysis of *Cyp3a11* mRNA expression. All data were visualized and analysed using Prism v.7.0a.

## Reporting summary

Further information on research design is available in the Nature Portfolio Reporting Summary linked to this article.

## Data availability

All untargeted LC–MS/MS data used in this study are publicly available at MassIVE (https://massive.ucsd.edu/) under the following

accession numbers: MSV000089491 (*N*-acyl amide synthetic mixtures); MSV000089493 (fatty ester synthetic mixtures); MSV000087522 (conjugated bile acid synthetic mixtures acquired on Q-ToF); MSV000087523 (conjugated bile acid synthetic mixtures acquired on Orbitrap); MSV000084908 (IBD dataset used to create Fig. 2e); MSV000088735 (IBD dataset used to create Fig. 2f); MSV000087562 (pooled fecal samples from iHMP2 cohort; Fig. 3a); MSV000092337 (IBD fecal samples used for quantitation; Fig. 3b); MSV000084475 (LC–MS/MS data for bacterial culture extracts; Fig. 4a,b); and MSV000087889 (LC-IMS-MS data for bacterial culture extracts; Fig. 4c). Source data for iHMP2 and PRISM datasets are available through Metabolomics Workbench (www.metabolomicsworkbench.org) under project identifiers PR000639 and PR000677, respectively. All spectra used in MASST searches are provided as a resource and can be directly downloaded from GNPS (https://gnps.ucsd.edu/ProteoSAFe/gnpslibrary. jsp?library=BILELIB19, https://gnps.ucsd.edu/ProteoSAFe/gnpslibrary. jsp?library=ECG-ACYL-AMIDES-C4-C24-LIBRARY, https://gnps.ucsd. edu/ProteoSAFe/gnpslibrary.jsp?library=ECG-ACYL-ESTERS-C4-C24-LIBRARY) and their USI and MASST job links are provided in Supplementary Table 8. Source data are provided with this paper.

## Code availability

Code used to produce figures can be found along with the source files at GitHub: https://github.com/emgentry/Synthesis-based-reverse-metabolomics.

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

**Acknowledgements** This research was supported by R01 DK13611701 and the Collaborative Microbial Metabolite Center supported under U24 DK133658, Crohn's and Colitis foundation number 675191, the US National Institutes of Health for the tools for rapid and accurate structure elucidation of natural products (R01 GM107550), the National Institutes of Health (P30 ES025128, P42 ES027704 and P42 ES031009 to E.S.B.) to (P42 ES027704, P42 ES031009, R01 GM141277 and RM1 GM145416 to E.S.B.), a Center for Computational Mass Spectrometry grant (P41 GM103484), a cooperative agreement with the United States Environmental Protection Agency (STAR RD 84003201 to E.S.B.) and the reuse of metabolomics data (R03 CA211211), the National Institute of Diabetes and Digestive and Kidney Diseases (U01 DK119702 to A.D.P.), the Pennsylvania Department of Health using Tobacco CURE funds (A.D.P.), the Tombros Foundation (A.D.P.), the National Institute of Diabetes and Digestive and Kidney Diseases (R00 DK110534 to H.C.; T32 DK007202 to M.C.T.), the National Institute of Allergy and Infectious Diseases (R01 AI67860 to H.C.), the National Academies of Sciences, Engineering and Medicine through the Predoctoral Fellowship of the Ford Foundation (M.C.T.), the Gordon and Betty Moore Foundation (A.T.A.), and the Howard Hughes Medical Institute Graduate Fellowships grant (GT15123 to M.C.T.). This research was supported in part by the Intramural Research Program of the NIH, National Institute of Environmental Health Sciences (ES103363-01). The views expressed in this manuscript do not reflect those of the funding agencies. The use of specific commercial products in this work does not constitute endorsement by the authors or the funding agencies.

**Author contributions** E.C.G. and P.C.D. conceptualized the idea. E.C.G. and M.P. performed all synthesis experiments. E.C.G., S.L.C., A.K.S., J.A.-P., A.T.A., F.H. and D.R.P. acquired data. E.C.G., S.L.C., M.P., D.R.P., S.Z., F.H. and M.S.-N. analysed data. H.V., C.B.C., E.S.B. and R.J.X. guided analyses. E.C.G. and P.B.-F. performed microbial culturing experiments. M.W. and A.K.J. developed the GNPS and MASST infrastructure used for data analyses. A.N.A., B.B., A.H. and N.V.C. coordinated clinical sample collection. D.S. managed, maintained and advised synthetic chemistry. S.L.C. and A.D.P. performed PXR activity assays. T.Y. and F.J.G. performed organoid culturing experiments. M.C.T. and H.h.L. performed immunomodulatory assays. H.C. supervised experimental design and data analyses. E.C.G. and P.D. wrote the manuscript and all authors contributed to editing.

**Competing interests** P.C.D. is on the scientific advisory board of Sirenas and Cybele Microbiome, and is founder and scientific advisor and has equity in Ometa Labs, Arome and Enveda (with approval by UC San Diego). M.W. is a co-founder and had equity in Ometa Labs. R.K. Gencirq (stock and SAB member), DayTwo (consultant and SAB member), Cybele (stock and consultant), Biomesense (stock, consultant, SAB member), Micronoma (stock, SAB member, co-founder) and Biota (stock, co-founder). No competing interests exist for E.C.G., S.L.C., M.P., P.B.-F., A.K.S., M.C.T., H.L., S.Z., T.Y., J.A.-P., D.R.P., A.T.A., A.K.J., F.H., M.S.-N., H.V., A.N.A., B.B., A.H., N.V.C., F.J.G., C.B.C., R.J.X., H.C., E.S.B., A.D.P. and D.S.

**Additional information**
**Correspondence and requests for materials** should be addressed to Pieter C. Dorrestein.

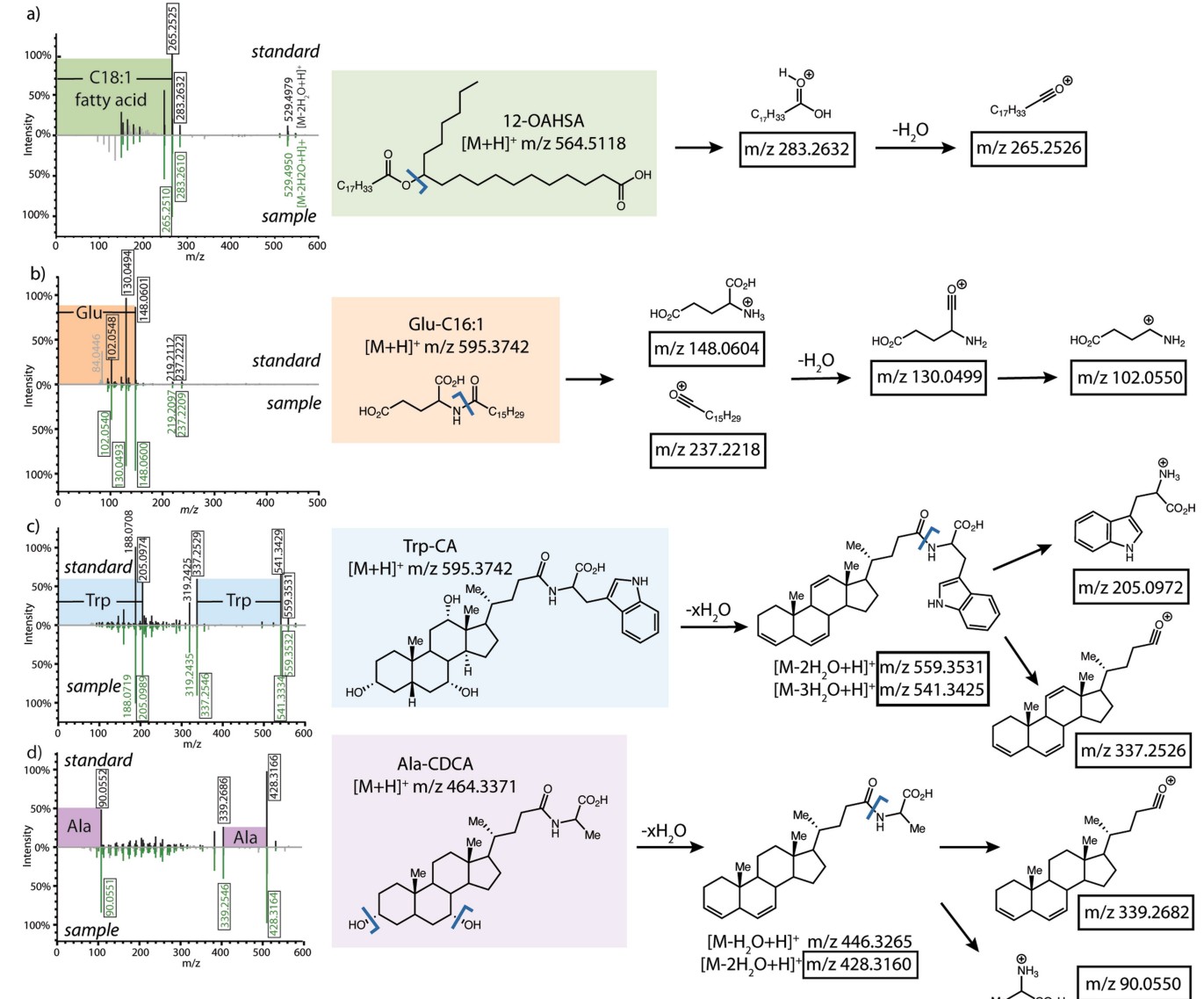

**Extended Data Fig. 1 | Representative fragmentation of standard vs observed MS/MS in public data with key fragment ions shown.** a) Fatty acid esters of hydroxy fatty acids (FAHFAs) b) N-acyl amides c) trihydroxylated bile acids d) dihydroxylated bile acids.

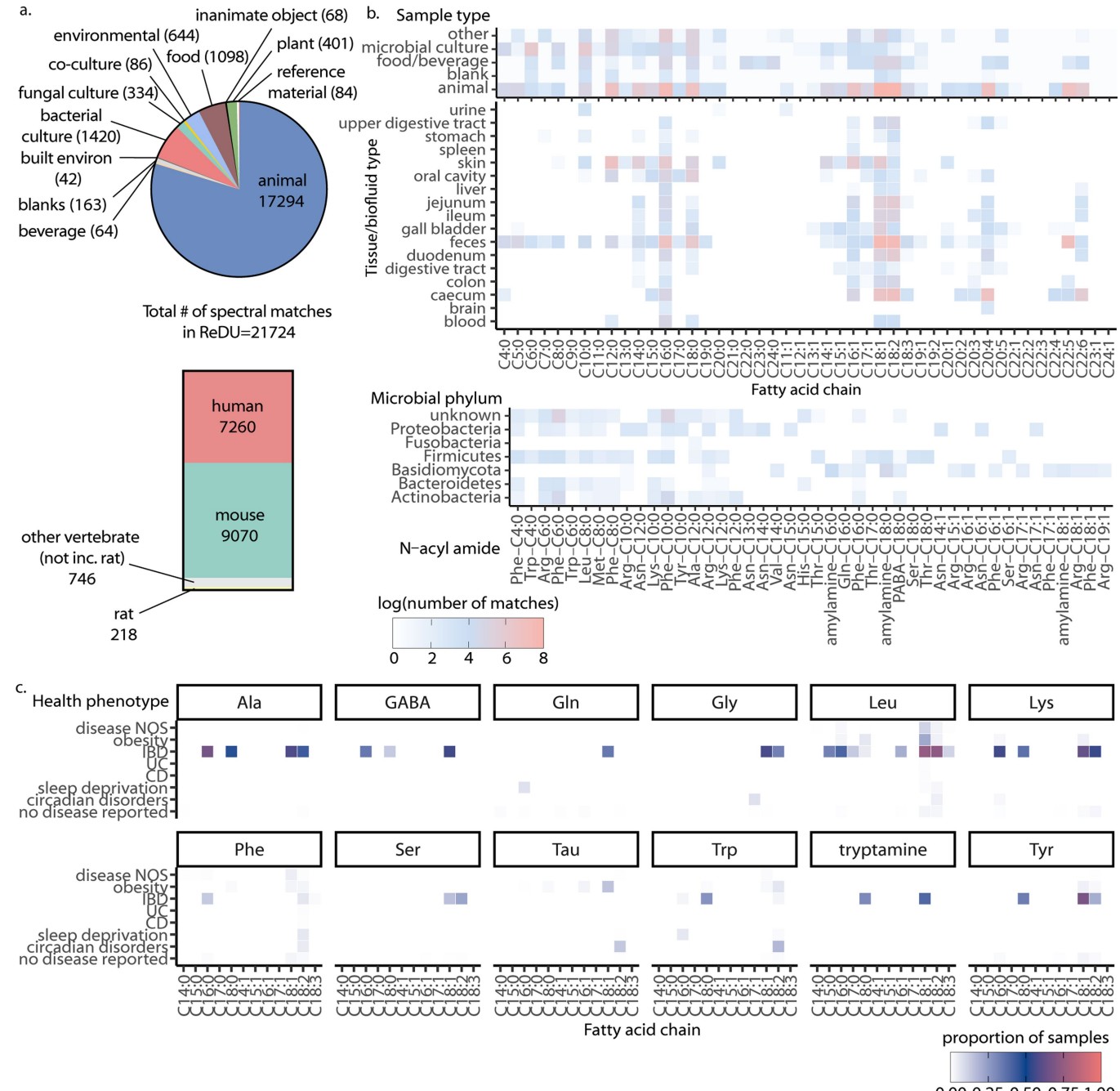

**Extended Data Fig. 2 | Results of MASST searches for *N*-acyl amides.** a) Pie chart representing sample types in which the spectra of synthesized amides were detected in ReDU. b) Heatmap representing log(number of spectral matches) across different sample types, organized by fatty acid chain identity.

c) Heatmap showing the proportion of samples where N-acyl amides were detected in different health-related phenotypes (all the matches to the IBD was from a longitudinal study of a single person).

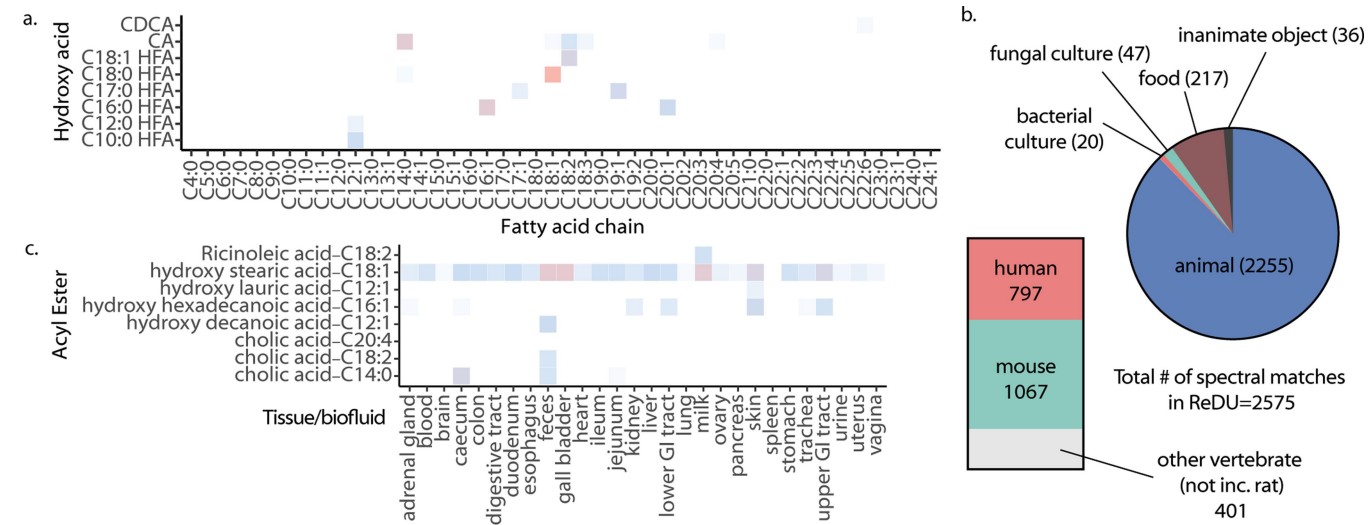

**Extended Data Fig. 3 | Results of MASST searches for acyl esters.**
a) Representative heatmap showing log(number of spectral matches) across unique synthesized acyl esters. b) Pie chart representing the number of spectra for synthesized esters detected in different sample types from ReDU. c) Heatmap representing log(number of spectral matches) for acyl esters across different tissue types.

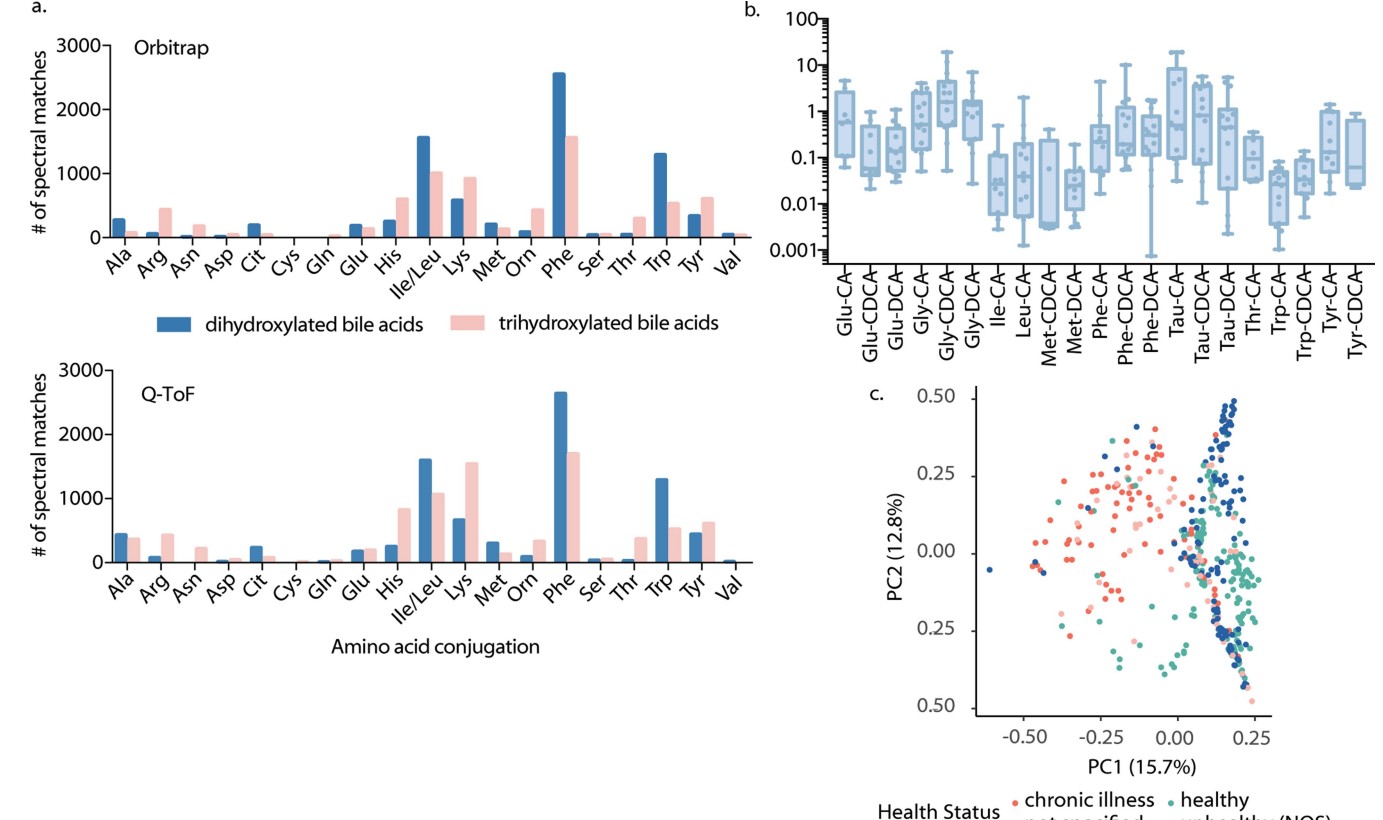

**Extended Data Fig. 4 | Analysis of synthetic conjugated bile acid mixtures.**
a) Comparison of amino acid distributions for MASST searches of Orbitrap (Thermo QE) vs. Q-ToF (Bruker MaXis) data in positive ionization mode. b) Quantification of conjugated bile acids in healthy human fecal samples (n = 15). Boxplots show first (lower) quartile, median, and third (upper) quartile, whiskers extend from the minimum to maximum values and the centre line indicates the median. c) Principal coordinates analysis (PCoA) plot using binary Jaccard distances for sample compositions of synthesized conjugated bile acids, which excludes Gly and Tau amides.

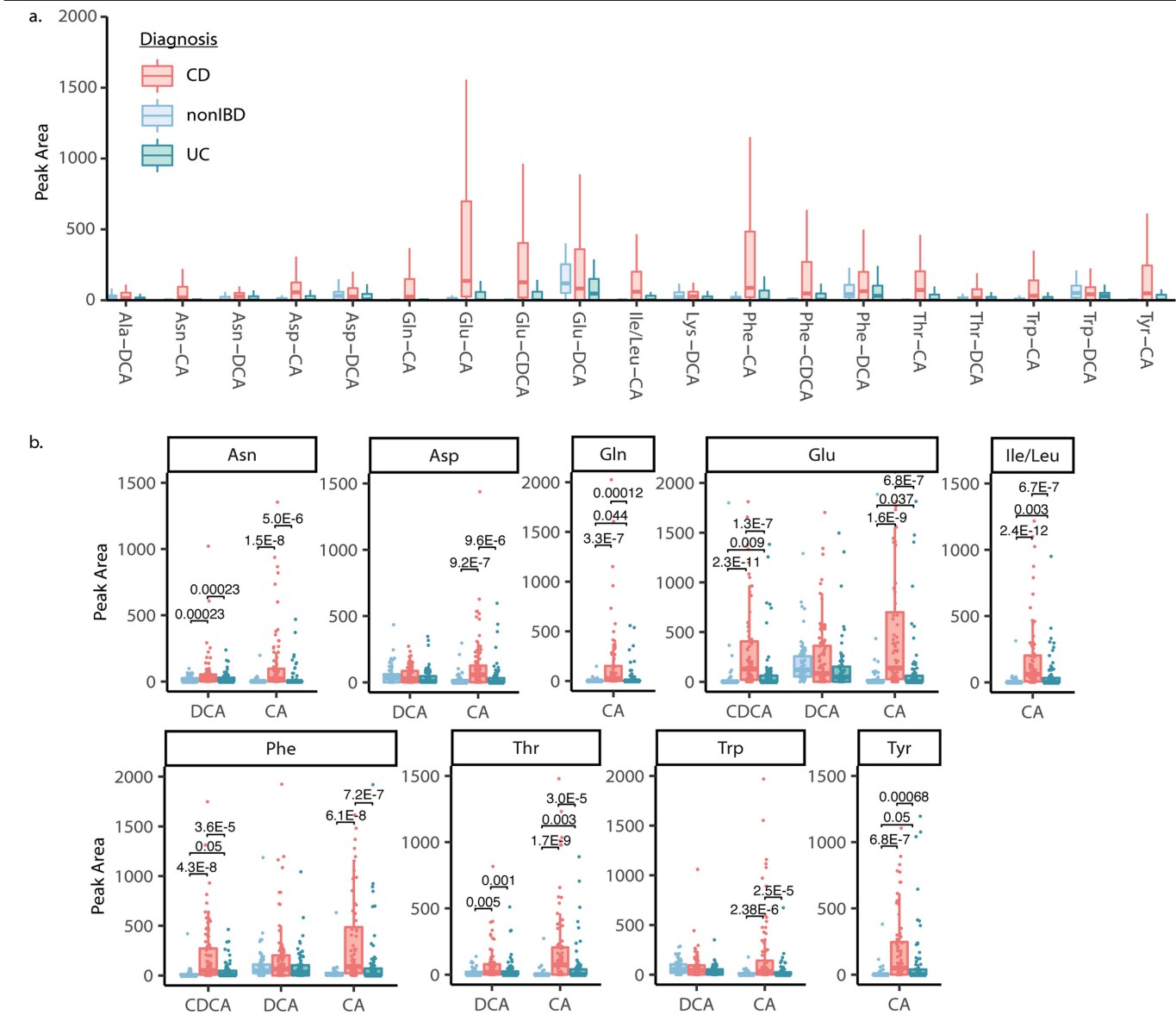

**Extended Data Fig. 5 | Independent validation of new conjugated bile acids in PRISM human IBD cohort.** a) Peak area abundances of conjugated bile acids detected in longitudinal PRISM dataset[10]. Boxplots show first (lower) quartile, median, and third (upper) quartile and whiskers are 1.5 times the interquartile range. n for Crohn's disease (CD) = 68, n for non-IBD = 34 and n for ulcerative colitis (UC) = 53. b) Peak area abundances of selected bile acids that were higher in Crohn's and/or Ulcerative Colitis patients relative to non-IBD individuals, as determined by pairwise two-sided Wilcoxon tests. Only significant values are shown, which were calculated using a Benjamini-Hochberg correction. Boxplots show first (lower) quartile, median, and third (upper) quartile and whiskers are 1.5 times the interquartile range. n for CD = 68, n for non-IBD = 34 and n for UC = 53.

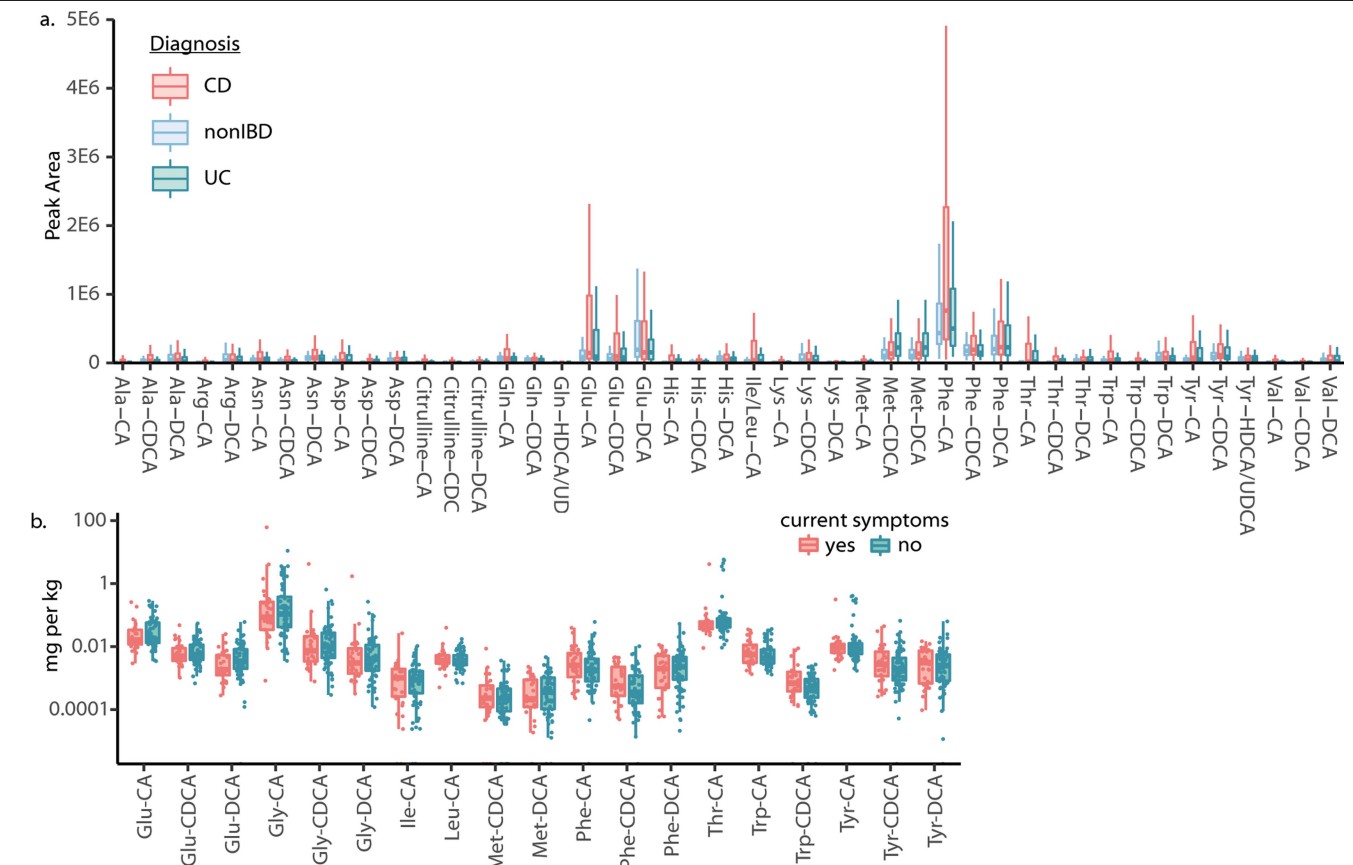

**Extended Data Fig. 6 | Overview of conjugated bile acids in iHMP2 human IBD cohort.** a) Peak area abundances of conjugated bile acids in the iHMP2 dataset. Boxplots show first (lower) quartile, median, and third (upper) quartile and whiskers are 1.5 times the interquartile range. n for CD = 265, n for non-IBD = 135 and n for UC = 146. b) Concentrations of conjugated bile acids in fecal samples from ulcerative colitis patients whose symptoms are active or inactive. Pairwise two-sided Wilcoxon tests were performed and no significant p-values were found. Boxplots show first (lower) quartile, median, and third (upper) quartile and whiskers are 1.5 times the interquartile range. n for CD = 265, n for non-IBD = 135 and n for UC = 146.

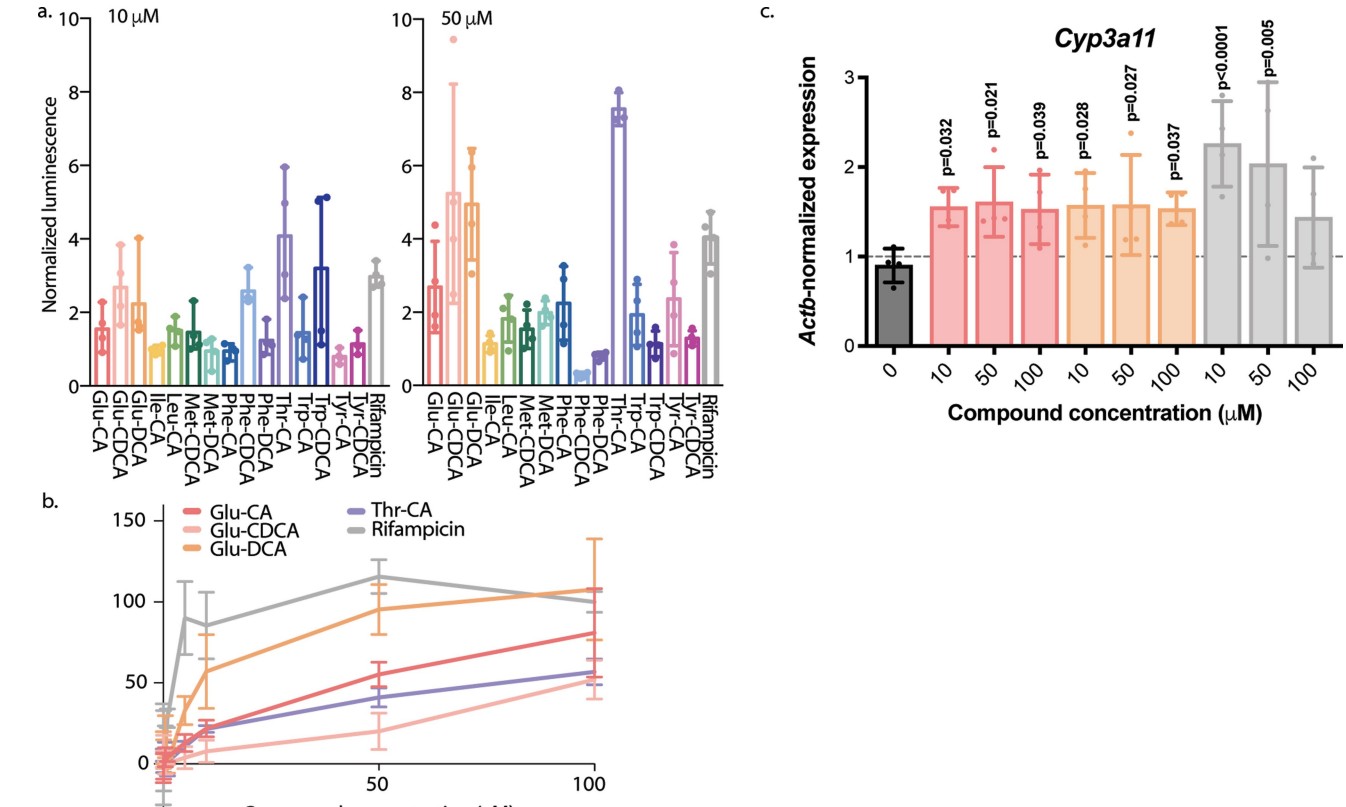

**Extended Data Fig. 7 | PXR activity of conjugated bile acids.** a) PXR agonism at 10 (left) and 50 μM (right) concentrations of the 15 new conjugated bile acids in the iHMP2 dataset, reported as normalized luciferase luminescence (a.u.). Top 15 bile acids were chosen based on the upper value of their interquartile range. Each concentration was run in biological quadruplicates for every substrate tested. Rifampicin, a known PXR agonist, was used as a positive control for comparison. Thr-CA significantly increases PXR agonism versus rifampicin (p = 0.0019) using a one-way ANOVA test. Data is represented as mean values +/− SD. b) Concentration dependence of PXR activity for top candidate agonists compared to Rifampicin, n = 5 for control and n = 4 for each bile acid concentration tested. Data represented as mean values +/− SD. c) Gene expression analysis of *Cyp3a11* in small intestinal organoids after exposure to conjugated bile acids at varying concentrations. Compounds are colored as Glu-CA (red), Glu-CDCA (orange), rifampicin (gray). Significance was calculated using one-way ANOVA, n = 4 biological replicates for each concentration tested, data represented as mean values +/− SD.

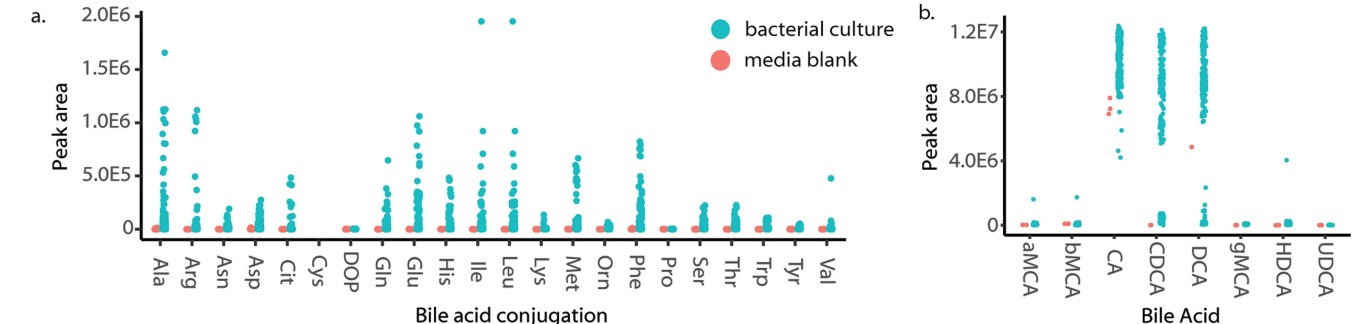

**Extended Data Fig. 8 | LC-IMS-MS Analysis of Bacterial Cultures.** a) Peak area abundances of conjugated bile acids in bacterial cultures vs. media blanks. b) Peak area abundances for unconjugated bile acids in bacterial cultures vs. media blanks.

# Reporting Summary

## Statistics

For all statistical analyses, confirm that the following items are present in the figure legend, table legend, main text, or Methods section.

| n/a | Confirmed | |
|---|---|---|
| ☐ | ☒ | The exact sample size ($n$) for each experimental group/condition, given as a discrete number and unit of measurement |
| ☐ | ☒ | A statement on whether measurements were taken from distinct samples or whether the same sample was measured repeatedly |
| ☐ | ☒ | The statistical test(s) used AND whether they are one- or two-sided<br>*Only common tests should be described solely by name; describe more complex techniques in the Methods section.* |
| ☐ | ☒ | A description of all covariates tested |
| ☐ | ☒ | A description of any assumptions or corrections, such as tests of normality and adjustment for multiple comparisons |
| ☐ | ☒ | A full description of the statistical parameters including central tendency (e.g. means) or other basic estimates (e.g. regression coefficient) AND variation (e.g. standard deviation) or associated estimates of uncertainty (e.g. confidence intervals) |
| ☐ | ☒ | For null hypothesis testing, the test statistic (e.g. $F$, $t$, $r$) with confidence intervals, effect sizes, degrees of freedom and $P$ value noted<br>*Give P values as exact values whenever suitable.* |
| ☒ | ☐ | For Bayesian analysis, information on the choice of priors and Markov chain Monte Carlo settings |
| ☒ | ☐ | For hierarchical and complex designs, identification of the appropriate level for tests and full reporting of outcomes |
| ☒ | ☐ | Estimates of effect sizes (e.g. Cohen's $d$, Pearson's $r$), indicating how they were calculated |

*Our web collection on statistics for biologists contains articles on many of the points above.*

## Software and code

Policy information about availability of computer code

| Data collection | Thermo XCalibur and Bruker Compass were used for LC-MS/MS data acquision. LC-IMS-MS data was acquired using Agilent MassHunter Acquisition Software. BD FACS DIVA Software (v8.0.2) were used to collect flow cytometry data. |
|---|---|
| Data analysis | Bruker DataAnalysis and GNPS Mass Spectrometry File Conversion (https://ccms-ucsd.github.io/GNPSDocumentation/fileconversion/) software were used to convert raw LC-MS/MS data to .mzXML or .mzML format. MS/MS-Chooser (https://ccms-ucsd.github.io/GNPSDocumentation/msmschooser/) and MASST (https://masst.ucsd.edu/) were used to analyze the LC-MS/MS data for the synthetic bile acid mixtures. Skyline was used for peak peaking and compound annotation of LC-IMS-MS data. FlowJo software suite (v10.8.0) was used for single-cell flow cytometry analysis. PRISM v7.0a, Excel 16.61.1, and RStudio 2022.02.3  were used to analyze data and generate graphs. No custom packages or software were developed for this analysis. Code used in this study are available at: https://github.com/emgentry/Synthesis-based-reverse-metabolomics. |

For manuscripts utilizing custom algorithms or software that are central to the research but not yet described in published literature, software must be made available to editors and reviewers. We strongly encourage code deposition in a community repository (e.g. GitHub). See the Nature Portfolio guidelines for submitting code & software for further information.

## Data

Policy information about availability of data

All manuscripts must include a data availability statement. This statement should provide the following information, where applicable:

- Accession codes, unique identifiers, or web links for publicly available datasets
- A description of any restrictions on data availability
- For clinical datasets or third party data, please ensure that the statement adheres to our policy

All untargeted LC-MS/MS data used in this study are publicly available at MassIVE (https://massive.ucsd.edu/) under the following accession numbers: MSV000089491 (N-acyl amide synthetic mixtures), MSV000089493 (fatty ester synthetic mixtures), MSV000087522 (conjugated bile acid synthetic mixtures acquired on Q-ToF), MSV000087523 (conjugated bile acid synthetic mixtures acquired on Orbitrap), MSV000084908 (IBD dataset used to create Figure 2e), MSV000088735 (IBD dataset used to create Figure 2f), MSV000087562 (pooled fecal samples from iHMP2 cohort - Figure 3a), MSV000092337 (IBD fecal samples used for quantitation - Figure 3b), MSV000084475 (LC-MS/MS data for bacterial culture extracts - Figure 4a and 4b), and MSV000087889 (LC-IMS-MS data for bacterial culture extracts - Figure 4c). Source data for iHMP2 and PRISM datasets are available through Metabolomics Workbench (www.metabolomicsworkbench.org) under project IDs PR000639 and PR000677, respectively. All spectra used in MASST searches are provided as a resource and can be directly downloaded from GNPS (https://gnps.ucsd.edu/ProteoSAFe/gnpslibrary.jsp?library=BILELIB19, https://gnps.ucsd.edu/ProteoSAFe/gnpslibrary.jsp?library=ECG-ACYL-AMIDES-C4-C24-LIBRARY, https://gnps.ucsd.edu/ProteoSAFe/gnpslibrary.jsp?library=ECG-ACYL-ESTERS-C4-C24-LIBRARY) and their universal spectral identifiers (USIs) and MASST job links are provided in Supplementary Table 8.

## Human research participants

Policy information about studies involving human research participants and Sex and Gender in Research.

| Reporting on sex and gender | This study includes information about the biological sex of patients assigned at birth, but does not include information about the gender of individuals. Here, there were an equal distribution of females (n=92) and males (n=90), as defined by sex, included in the study. Sex-based analyses are not included in this manuscript as there were no significant findings with relation to the abundance of these bile acids in different sexes. |
|---|---|
| Population characteristics | In this study, diagnosis and symptom activity were population characteristics used for the analysis. Clinical diagnosis was defined as either Crohn's disease or ulcerative colitis, based on histological and clinical features of inflammatory bowel disease. Symptom activity was self-reported as either yes or no by patients. Patients were 18-79 years old with an average age of 45. |
| Recruitment | Patients aged 18 years and older with a confirmed diagnosis of IBD were recruited during routine visits with no bias toward any particular group. Informed consent was obtained. |
| Ethics oversight | Human fecal samples were collected under UCSD IRB approval #131487. |

Note that full information on the approval of the study protocol must also be provided in the manuscript.

# Field-specific reporting

Please select the one below that is the best fit for your research. If you are not sure, read the appropriate sections before making your selection.

☒ Life sciences        ☐ Behavioural & social sciences        ☐ Ecological, evolutionary & environmental sciences

For a reference copy of the document with all sections, see nature.com/documents/nr-reporting-summary-flat.pdf

# Life sciences study design

All studies must disclose on these points even when the disclosure is negative.

| Sample size | For repository-scale analysis, we did not use statistical methods to determine sample size, but rather the sample size was selected by the number of files publicly available in the GNPS database and ReDU. We found an association of some conjugated bile acids with IBD which was validated in independent clinical cohorts. Therefore, the study was sufficiently powered to produce useful results. Sample size for iHMP2 (https://pubmed.ncbi.nlm.nih.gov/31142855/) and PRISM (https://www.ncbi.nlm.nih.gov/pmc/articles/PMC6342642/) cohorts were not pre-determined, but rationale for their respective sample sizes were included in reporting summaries of their associated publications. To quantify conjugated bile acids in fecal samples from IBD patients and healthy controls, 162 (72 Crohn's patients and 90 UC patients) and 15 subjects were included, respectively. For PXR and immunological assays, at least three biological replicates were used to determine fold induction of experimental samples vs. untreated control. For bacterial cultures, biological duplicates of individual strains were deemed sufficient to characterize bile acid conjugating abilities on the genus and class level.<br>For public datasets used in this manuscript,<br>MSV000084908 IBD 200 study: 203<br>MSV000088735 Pediatric IBD study: 27 |
|---|---|

| Data exclusions | No data were excluded from the manuscript, but some figure panels include representative examples (e.g. conjugated bile acids elevated in IBD) rather than all of the data. |
|---|---|
| Replication | Reproducibility was verified by performing independent biological replicates and attempts at replication were successful. For PXR reporter assays and organoid experiments, each concentration of each bile acid was performed four times. For T cell differentiation assay, each experiment was performed at least three times. For culturing bacterial isolates, experiments were performed in duplicate. |
| Randomization | Not applicable to human IBD cohorts as data was from a cross-sectional study in which multiple pre-defined cohorts were determined by diagnosis prior to the study. For all LCMS analyses, the order in which samples were placed in 96-well plates was randomized. The order in which samples were run was also randomized per 96-well plate with the exception of the bacterial cultures. In this case, blanks were run after every six sample injections and QCs were checked to ensure no carry over or instrument drift occurred during analysis. |
| Blinding | For human IBD cohorts, blinding was not possible because study subjects were diagnosed prior to this study.  For all other experiments, researchers were not blinded, but data was analyzed in an objective fashion. |

# Reporting for specific materials, systems and methods

We require information from authors about some types of materials, experimental systems and methods used in many studies. Here, indicate whether each material, system or method listed is relevant to your study. If you are not sure if a list item applies to your research, read the appropriate section before selecting a response.

## Materials & experimental systems

| n/a | Involved in the study |
|---|---|
| ☐ | ☒ Antibodies |
| ☐ | ☒ Eukaryotic cell lines |
| ☒ | ☐ Palaeontology and archaeology |
| ☒ | ☐ Animals and other organisms |
| ☒ | ☐ Clinical data |
| ☒ | ☐ Dual use research of concern |

## Methods

| n/a | Involved in the study |
|---|---|
| ☒ | ☐ ChIP-seq |
| ☐ | ☒ Flow cytometry |
| ☒ | ☐ MRI-based neuroimaging |

## Antibodies

| Antibodies used | Rat anti-CD4 [RM4-5] APC-ef780 (Invitrogen #47-0042-82) (1:400 for flow cytometry)<br>Mouse anti-CD2 [RPA-2.10] PE (Invitrogen #12-0029-41) (1:400 for flow cytometry)<br>Rat anti-IL-17A [eBio17B7] PEcy7 (Invitrogen #25-7177-80) (1:200 for flow cytometry)<br>Rat anti-IFNg [XMG1.2] BV785 (Biolegend #505838) (1:400 for flow cytometry)<br>Rat anti-RORgt [AFKJS-9] APC (Invitrogen #17-6988-82) (1:200 for flow cytometry)<br>Rat anti-T-bet [4B102] PE/Dazzle 594 (Biolegend #644828) (1:200 for flow cytometry)<br>Live/Dead fixable aqua dead cell stain kit (Invitrogen #L34966) (1:1000 for flow cytometry)<br>anti-mouse CD3 [17A2]  (Invitrogen # 14-0032-82) (1:500 for flow cytometry) |
|---|---|
| Validation | Rat anti-CD4 [RM4-5] APC-ef780 (Invitrogen #47-0042-82) (1:400 for flow cytometry)<br>-https://www.thermofisher.com/antibody/product/CD4-Antibody-clone-RM4-5-Monoclonal/47-0042-82<br><br>Mouse anti-CD2 [RPA-2.10] PE (Invitrogen #12-0029-41) (1:400 for flow cytometry)<br>-https://www.thermofisher.com/antibody/product/CD2-Antibody-clone-RPA-2-10-Monoclonal/12-0029-41<br><br>Rat anti-IL-17A [eBio17B7] PEcy7 (Invitrogen #25-7177-80) (1:200 for flow cytometry)<br>-https://www.thermofisher.com/antibody/product/IL-17A-Antibody-clone-eBio17B7-Monoclonal/25-7177-80<br><br>Rat anti-IFNg [XMG1.2] BV785 (Biolegend #505838) (1:400 for flow cytometry)<br>-https://www.biolegend.com/en-us/products/brilliant-violet-785-anti-mouse-ifn-gamma-antibody-7987?GroupID=GROUP24<br><br>Rat anti-RORgt [AFKJS-9] APC (Invitrogen #17-6988-82) (1:200 for flow cytometry)<br>-https://www.thermofisher.com/antibody/product/ROR-gamma-t-Antibody-clone-AFKJS-9-Monoclonal/17-6988-82<br><br>Rat anti-T-bet [4B102] PE/Dazzle 594 (Biolegend #644828) (1:200 for flow cytometry)<br>-https://www.biolegend.com/en-us/products/pe-dazzle-594-anti-t-bet-antibody-11891<br><br>Live/Dead fixable aqua dead cell stain kit (Invitrogen #L34966) (1:1000 for flow cytometry)<br>-https://www.thermofisher.com/order/catalog/product/L34965<br><br>anti-mouse CD3 [17A2]  (Invitrogen # 14-0032-82) (1:500 for flow cytometry)<br>https://www.thermofisher.com/antibody/product/CD3-Antibody-clone-17A2-Monoclonal/14-0032-82 |

# Eukaryotic cell lines

Policy information about cell lines and Sex and Gender in Research

| | |
|---|---|
| Cell line source(s) | Bone marrow-derived dendritic cells were derived from WT C57BL6/J mice and CD4+ T cells were derived from Foxp3-hCD2 reporter mice. Huh7 cells were obtained from Xenotech. Organoids were derived from the small intestine of PXR-humanized mice in C57Bl6/N background. |
| Authentication | Cell lines used were not authenticated. |
| Mycoplasma contamination | Cell lines were not tested for mycoplasma contamination. |
| Commonly misidentified lines (See ICLAC register) | No commonly misidentified lines were used. |

# Flow Cytometry

## Plots

Confirm that:

☒ The axis labels state the marker and fluorochrome used (e.g. CD4-FITC).

☒ The axis scales are clearly visible. Include numbers along axes only for bottom left plot of group (a 'group' is an analysis of identical markers).

☒ All plots are contour plots with outliers or pseudocolor plots.

☒ A numerical value for number of cells or percentage (with statistics) is provided.

## Methodology

| | |
|---|---|
| Sample preparation | Naïve CD4+ T cells were isolated from Foxp3-hCD2 reporter mice and cultured with bone marrow derived dendritic cells from WT C57BL6/J mice under Th17 polarizing conditions. Cells were treated with 100 μM of bile acids on day 0 and collected on day 3 for flow cytometric analysis. Cells were first stained with Live/Dead dye and surface markers and then fix/permed with eBioscience™ Foxp3 / Transcription Factor Staining Buffer Set. RORgt were stained for overnight and other intracellular markers and cytokines were stained for 3h. |
| Instrument | BD LSR II |
| Software | BD FACS DIVA Software (v8.0.2) were used to collect data. FlowJo (v10.8.0) software were used to analyze data. No custom code has been deposited into a community repository. |
| Cell population abundance | The purity of Th17 cells were determined by the labeling of specific markers. Th17 cells were defined as CD4+ hCD2 (Foxp3)- Tbet- RORgt+ cells. The abundance of Th17 cells were 46.1 ± 6.83%. |
| Gating strategy | Samples were first gated to distinguish population of cells based on the side and forward scatter properties (SSC-A vs FSC-A), followed by singlets (FSC-H vs FSC-A and SSC-H vs SSC-A). Live cells were identified by using live/dead staining. Th17 cells were defined as CD4+ hCD2 (Foxp3)- Tbet- RORgt+ cells. IL-17A and IFNg expression were examined on Th17 populations. |

☒ Tick this box to confirm that a figure exemplifying the gating strategy is provided in the Supplementary Information.

