## [Peer Review File · Nature]

Manuscript Title: Reverse Metabolomics for the Discovery of Chemical Structures from Humans

Reviewer Comments & Author Rebuttals

Reviewer Reports on the Initial Version:

Referees' comments:

Referee #1 (Remarks to the Author):

In this paper, Gentry and colleagues described a novel “synthesis-based reverse metabolomics approach” to identify new metabolites and their structure.

Instead of the classical method consisting of the isolation of a compound of interest, followed by its structural characterization (which is very fastidious and not amenable to high throughput), the authors propose to reverse the process. Metabolites with a known structure are synthesized first using synthetic combinatorial chemistry, and their spectral data are used to search for matches in public untargeted metabolomics data.

As a proof of concept, the authors applied this strategy to discover new bile acids.

They synthesized an array of amino acid conjugated bile acids using combinatorial amide coupling reactions between 8 unique di- and tri-hydroxylated bile acids and 22 amino acids.

They identified 145 different bile acid amidates that have MS/MS matches in public metabolomics data, including 139 that had never been described before. The authors observed that some of the newly identified bile acids were associated with inflammatory bowel disease phenotype. A confirmation of these associations was performed using data from an independent cohort.

The authors then focused on the ability of some bile acids to activate the nuclear receptor PXR, which is involved in many physiologic and pathogenic mechanisms. The ability of the 12 most abundant bile acid conjugates observed in the iHMP2 cohort to activate PXR was evaluated in a cell reporter assay. The authors report that some of them indeed activate PXR.

The authors then screened 202 isolates from the first Human Microbiome Project for bile acid conjugation capabilities and observed that many bacteria were able to make different amino-acid conjugation.

The described method is very innovative and has a huge potential to reduce the dark (unidentified) matter from metabolomics data, particularly in the microbiome field.

I cannot comment on chemistry aspects as they are not in my field of expertise.

The “biological” results are relatively limited and it seems that most of the novelty of the paper is in the “synthesis-based reverse metabolomics approach” to identify new metabolites.

Major comments:

- The PXR results seem a little bit “dry”. For example, it would be interesting to connect these results with the alteration of BA composition observed in patients with IBD.
- Similarly, the results on the ability of bacterial isolates to conjugate bile acids are very limited.

These two biological results are only observational and, currently, they do not bring so much to the paper. I would suggest either expanding the functional consequences of these results (but it might be out of the scope of this paper), or reducing it.

Minor comments:

- The limitation of the newly described approach should be mentioned. Particularly, this method supposes that the investigator determines a priori the type of molecules he/she wants to focus on. It will thus not apply to the characterization of a metabolite with potential biological interest and identified with a non a priori method (for example, a differential analysis between healthy subjects and patients).
- The sentence “ drugs such as rifamycins that target the pregnane X receptor (PXR), a bile acid receptor, are successfully used to treat IBD” (lines 243-245) does not seem right. Rifamycins are not used to treat IBD, and the cited references do not refer to it.

Referee #2 (Remarks to the Author):

The letter by Gentry et al describe a “synthesis-based reverse metabolomics” approach whereby MS/MS spectra acquired from synthesized compounds are then searched against public metabolomics data to identify these compounds and associate them with disease. The authors generated 145 amidated bile acids through amidation and confirmed the structures. Searching public databases by MASST and ReDU, as well as separate IBD cohorts, the authors report increased levels of these novel amidated bile acids in IBD. PXR is targeted for IBD treatment by rifampicin, and the authors show that these amino acid-bile acid conjugates activate PXR with similar affinity as rifampicin, although it is not clear what this might mean. Identifying bacteria that can generate these conjugates is also important in terms of linking this to the microbiome and future attempts to identify mechanism of conjugation. Overall, this is a useful approach that can be widely adapted to the study of other metabolites. There are some concerns about novelty given the previous report by the corresponding author of novel bile acid-amino acid conjugates in Nature recently (PMID: 32103176).

Major criticisms

Line 207-209: It is stated that the novel amino acid conjugates are observed at higher frequency in IBD patients, and refers to Fig 2C. The heat map intensity is based on “% of matches”, but the intensity in IBD, UC, and CD are very low compared to “no disease reported”. However, in figure 3b, which shows “peak area”, there is consistently higher levels relative to “nonIBD”, which presumably is the same group as “no disease reported”. This is confusing, because in figure 3C and 3D it is suggested that these bile acid conjugates are just as effective, if not more, in several cases as rifampicin (treatment) at activating PXR. Then we are told that more studies are needed to

determine what is going on with these conjugates and IBD. So it is unclear if these conjugates are low in IBD relative to health (Figure 2C), or significantly higher (Figure 3b), but serve as a potential treatment for the disease state in which they are abundant in (Figure 3C, 3D).

Line 297-299: It is suggested that 12 α -dehydroxylation has not been reported. However, there have been reports of this, including recently (PMID:34463573; PMID:6492790). However, the medium that is used, reinforced clostridium medium (Line 567), has peptone. It has been reported that peptone from animal sources contains 9.53 mg/g bile acids (PMID:16347619). Our lab has confirmed this with peptone sources we have obtained by LC/MS. It will be important to know if CDCA is detected and at what concentration in the RCM blank.

Minor criticisms

-Line 55 and 300: bacterial genera should either be described as Bifidobacterium or bifidobacteria, for instance.

-Are any of the bile acid conjugates in Figure 3 c and d significant relative to rifampicin treatment?

-with rifampicin treatment, the text does not establish the dose being tested is physiologically relevant.

Referee #3 (Remarks to the Author):

Summary:

The workflow described in this paper is to synthesize putative biological compounds, obtain MS/MS on both the orbitrap and QTOF, run MS/MS searches in publicly deposited mass spec data to find matches and then link with phenotype through searching of public metabolomics datasets.

Comments:

The authors recently published on a novel class of amino acid conjugated bile acids (Nature, PMID: 32103176). It is reasonable to predict that other amino acid conjugations might also exist. In this work the authors made the conjugations, collected MS/MS data from the standards and then looked for evidence that they were present in public datasets. This is a sensible and compelling set of experiments to pursue. From this, however, the authors try to abstract a workflow that they call "reverse metabolomics" that they claim is broadly applicable to the discovery of many new classes of metabolites. This reviewer questions whether the presented method will be generally useful for other metabolites. The bile acids are an exceptional group of compounds for a few reasons. First the authors already had strong reason to believe that additional conjugates would exist based on their prior work. Second the chloroformate mediated amide bond chemistry is unique and it was straightforward for the authors to produce many different conjugates with high yield. It is not clear

that the same strategy will work with other classes of compounds with sufficient throughput, especially without a pre-defined hypothesis and modular chemical functional groups. To demonstrate that "reverse metabolomics" is an effective and efficient strategy to discover unknown molecules the authors need to demonstrate its utility beyond one unique class of compounds.

Another issue with "reverse metabolomics" as a broad approach is that no assessment is provided for how well it works. MS/MS for metabolites is not very specific and the problem with public databases is that there is no quality control, so the data might not be reliable. When searching tens of thousands of public MS/MS patterns it is likely that there will be artificial matches. How frequently does this occur? How do the authors control for false positive hits? The authors should test molecules that do not occur in samples and see what hits are returned.

A related question is in how many public datasets does the hit have to occur to be reliable? This should be assessed. The fact that some bile acids occur in one matrix for a few rare samples (for example blood or liver) demonstrates the complication. Why do bile acids occur in blood, liver and urine in these samples? Is that real? Is there a biological explanation? Most likely it represents false positives and highlights the need to implement some method to control for nonspecific hits. These questions show why it is important to look at more than one example for quantitative testing of false discovery and true positive statistics.

The major finding of this work is the report of new bile acids in human samples. To the best of my knowledge this is the first time that these compounds are being reported in human subjects. The burden of proof to show that these are indeed the compounds being measured is high. I am concerned by how poorly some of the MS/MS from standards match the MS/MS from samples. It could easily be that the compounds in the samples are some structural variation of the standard. To name a few--- Ala-aMCA has a fragment at 350 that is not in the sample. Fragments from 200-220 are much larger in the sample than in the standard. Fragment 426 is twice as large in the sample than the standard. Ala-bMCA has a major (almost most dominant fragment) at 126 in the sample that is not in the standard. Ala-CA also has a major non-matching fragment at 126. The patterns for most Arg-species look totally different between the sample and standard. The ratios between fragments in the citrate gMCA case does not match between standards and sample. Glu-CDCA His-CA are other examples in which the intensities of the fragments actually flip. The low intensity fragment from the sample is the highest in the standard and vice versa. This data alone falls short of the evidence that is needed to convincingly support the publication of new compounds.

It is surprising that the authors didn't confirm that the MS/MS from their synthesized standards matched MS/MS from human samples collected in their own lab. I understand that it is intriguing to use public data but there are too many questions about its quality and provenance. To confirm identification of new compounds, confirmation is needed whereby the authors collect data on samples and standards using the same methods, instruments, voltages etc. The authors should ensure that the MS/MS, retention time and NMR spectra from the standards match human samples analyzed in their own lab. This is the longstanding convention when identifying new natural products.

The authors make a weak claim that the bile acids are associated with inflammatory bowel disease.

Author Rebuttals to Initial Comments:

Referees' comments:

Referee #1 (Remarks to the Author):

In this paper, Gentry and colleagues described a novel “synthesis-based reverse metabolomics approach” to identify new metabolites and their structure.

Instead of the classical method consisting of the isolation of a compound of interest, followed by its structural characterization (which is very fastidious and not amenable to high throughput), the authors propose to reverse the process. Metabolites with a known structure are synthesized first using synthetic combinatorial chemistry, and their spectral data are used to search for matches in public untargeted metabolomics data.

As a proof of concept, the authors applied this strategy to discover new bile acids.

They synthesized an array of amino acid conjugated bile acids using combinatorial amide coupling reactions between 8 unique di- and tri-hydroxylated bile acids and 22 amino acids.

They identified 145 different bile acid amidates that have MS/MS matches in public metabolomics data, including 139 that had never been described before. The authors observed that some of the newly identified bile acids were associated with inflammatory bowel disease phenotype. A confirmation of these associations was performed using data from an independent cohort.

The authors then focused on the ability of some bile acids to activate the nuclear receptor PXR, which is involved in many physiologic and pathogenic mechanisms. The ability of the 12 most abundant bile acid conjugates observed in the iHMP2 cohort to activate PXR was evaluated in a cell reporter assay. The authors report that some of them indeed activate PXR. The authors then screened 202 isolates from the first Human Microbiome Project for bile acid conjugation capabilities and observed that many bacteria were able to make different amino-acid conjugation.

Excellent summary.

The described method is very innovative and has a huge potential to reduce the dark (unidentified) matter from metabolomics data, particularly in the microbiome field.

Thanks.

I cannot comment on chemistry aspects as they are not in my field of expertise.

The “biological” results are relatively limited and it seems that most of the novelty of the paper is in the “synthesis-based reverse metabolomics approach” to identify new metabolites.

The reviewer is exactly correct about the specifics of the advance we claim in this paper. A large part of the novelty is that we can connect the result to biological and/or clinical information, also called metadata (taxonomy, disease, tissue type etc), so quite a lot of information can be learned very quickly with respect to the new molecules, and this ability will grow as more LC-MS/MS based untargeted metabolomics projects are made public (currently, ~2,300 public projects, 1.2 billion MS/MS spectra and doubling every 2-3 years).

Major comments:

- The PXR results seem a little bit “dry”. For example, it would be interesting to connect these results with the alteration of BA composition observed in patients with IBD.

We appreciate the comment. Our question was specifically to address whether they had any potential bioactivity and how they related to IBD. We do not yet have a complete mechanistic link, but many labs, including dedicated experts in IBD (our lab is expert in metabolite annotation from untargeted metabolomics), are beginning to work on a complete mechanistic understanding of the molecules. However, based on the comment as well as that of other reviewers, we aimed to further solidify that these bile acids are associated with IBD and that pathways known to be involved in IBD can be influenced by the bile acids. We now show that antibiotic treatment in pediatric IBD patients alters the levels of the bile acids (Figure 2f). We also show that these bile acids are increased in Crohn’s disease patients with active symptoms (Figure 3b). We now also show they affect production of pro-inflammatory cytokines involved in IBD such as IL-17 and IFN- γ (Figure 3c) and expression of the known downstream PXR target gene Cyp3a11 (Extended figure 6c).

Figure 2: Repository-scale analysis of public mass spectrometry data. a) Heatmap representing the log value of spectral matches for 1472 *N*-acyl amides in the entire GNPS metabolomics repository. b) Heatmap showing the log value of unique MS/MS spectral matches for each amine conjugation in different tissues and biofluids using GNPS public data with metadata available in ReDU. c) Heatmap showing the log value of unique spectral matches for conjugated bile acids across different tissues and biofluids using GNPS public data with metadata

available in ReDU. d) Heatmap showing the proportion of samples that each MS/MS synthesized bile acid was detected for health phenotypes across the public metabolomics repositories that have metadata available in ReDU. CD, Crohn's disease; UC, ulcerative colitis; IBD, inflammatory bowel disease; disease NOS, disease not otherwise specified. From top to bottom for health phenotype, n=1556, n=679, n=84, n=144, n=207, n=46, n=713, n=317, n=195, n=56, n=59, n=13108 and n=6092. e) Relative MS1 abundances of conjugated bile acids across clinical groups in the single project MSV000084908. CD, Crohn's disease, n=103; UC, Ulcerative colitis, n=60. f) Relative MS1 abundances of conjugated bile acids in relation to antibiotic usage in data from a pediatric IBD cohort deposited as MSV000088735. Antibiotic use, n=72; no antibiotic use, n=45. Boxplots show first (lower) quartile, median, and third (upper) quartile and whiskers are 1.5 times the interquartile range. Statistical significance was tested using a pairwise two-sided Wilcoxon rank sum test. Only significant p-values are shown, which were adjusted using Benjamini-Hochberg correction.

Figure 3: IBD association of new conjugated bile acids. a) Relative peak area abundances of selected bile acids that were higher in Crohn's (red) and/or Ulcerative Colitis (green) patients relative to non-IBD individuals (blue) in iHMP2 study, as determined by pairwise two-sided Wilcoxon tests. Boxplots show first (lower) quartile, median, and third (upper) quartile and whiskers are 1.5 times the interquartile range. Significance is shown as Benjamini-Hochberg corrected p-values. CD, Crohn's disease, n = 265; non-IBD, n = 135; UC, Ulcerative colitis, n = 146 b) Concentrations of conjugated bile acids in fecal samples from individuals with active (n=23) or inactive (n=72) Crohn's disease. Y-values represent mg of conjugated bile acid per kg of fecal matter. Boxes represent the interquartile range, center line is the median and whiskers are 1.5

times the interquartile range. Statistical significance was tested using a one-sided Wilcoxon test and only significant p-values are shown. c) Flow cytometric quantification of IL-17 and IFN- γ production in naive CD4+ T cells from Foxp3-hCD2 reporter mice. Cells were treated with 100 μ M of bile acids on day 0 and CD4+ T cells were gated for analyses on day 3. Assays were performed with three or more biological replicates for every substrate tested. Bar plot shows the mean value and error bars represent standard deviation. Statistical significance calculations were performed using a one way Kruskal-Wallis test.

Extended Data Fig. 6: PXR activity of conjugated bile acids. a) PXR agonism at 10 (left) and 50 μ M (right) concentrations of the 15 new conjugated bile acids in the iHMP2 dataset, reported as normalized luciferase luminescence (a.u.). Top 15 bile acids were chosen based on the upper value of their interquartile range. Each concentration was run in biological quadruplicates for every substrate tested. Rifampicin, a known PXR agonist, was used as a positive control for comparison. Thr-CA significantly increases PXR agonism versus rifampicin ($p=0.0019$) using a one-way ANOVA test. b) Concentration dependence of PXR activity for top candidate agonists compared to Rifampicin. c) Gene expression analysis of *Cyp3a11* in small intestinal organoids after exposure to conjugated bile acids at varying concentrations. Compounds are colored as Glu-CA (red), Glu-CDCA (orange), rifampicin (gray). Significance was calculated using one-way ANOVA and is denoted by asterisks: *, $P < 0.05$; **, $P < 0.01$; ***, $P < 0.001$; ****, $P < 0.0001$.

- Similarly, the results on the ability of bacterial isolates to conjugate bile acids are very limited. These two biological results are only observational and, currently, they do not bring so much to the paper. I would suggest either expanding the functional consequences of these results (but it might be out of the scope of this paper), or reducing it.

We agree with the reviewer that a full mechanistic interpretation is beyond the scope of this paper. Reverse metabolomics, by design, is observational so the interpretation by the reviewer is absolutely correct. Without establishing the existence of molecules, it becomes very hard to begin to understand who and what makes them and propose their function. Here we not only have one of the most effective approaches for structure elucidation of new molecules but also to learn about their biological observations and associations. Using this approach we have uncovered that not only are they found in animals, but also in people and that they are disease associated and human microbiota are able to make them. We know there are now a growing number of labs working on the full mechanistic interpretation of the new bile acids, but we think it is important to show here that they are indeed associated with IBD and that they have biological effects at concentrations comparable to that we see in stool, as described with the figures above. I personally cannot wait to see how the mechanistic puzzle will be solved by the community.

This is how we rephrased it in the text for PXR, IBD, immune marker links that are known but we state that the role of these bile acids is at a hypothesis stage.

“Since bile acids can be immunomodulatory, one pathway of interest is bile acid-mediated immune dysfunction. A series of recent studies discovered that cholic acid, chenodeoxycholic acid and two secondary lithocholic acid metabolites can affect T cell homeostasis^{49–51} and we postulated that some of our lead bile acids may also dysregulate host immune response. Therefore, we tested our synthesized bile acids for immunomodulatory activities and found that five of the conjugated bile acids increased production of interleukin 17 (IL-17) in CD4+ T cells and three of these (Phe-CDCA, Trp-CDCA and Tyr-CDCA) also increased levels of interferon gamma (IFN- γ) (**Figure 3c**). Most notably, Trp-CDCA resulted in about a six-fold increase of both IL-17 and IFN- γ , two cytokines involved in the pathogenesis of Crohn’s disease⁵².”

“Specifically, PXR agonizing drugs that are semisynthetic derivatives of rifamycin (e.g. rifaximin, rifampicin) have been used to treat IBD in clinical trials and decreased PXR gene expression has been associated with IBD^{46,47,51–55}.”

“Taken together, these findings support the hypothesis that some of the newly discovered bile acids could play an important role in IBD through PXR and/or immune-mediated processes, more work is needed to fully understand these finding but it shows that reverse metabolomics was able to discover new biologically relevant molecules.”

Minor comments:

- The limitation of the newly described approach should be mentioned. Particularly, this method supposes that the investigator determines a priori the type of molecules he/she wants to focus on. It will thus not apply to the characterization of a metabolite with potential biological interest and identified with a non a priori method (for example, a differential analysis between healthy subjects and patients).

Thanks for this comment, as we should be more specific about this issue. Solving the data non a-priori (for signals between disease and healthy) is not the goal of this work, although we clearly show that it does provide insight into such states for IBD for some molecules. Thus our approach is hypothesis-generating, akin to most omics technologies. There are no other papers that synthesize hundreds (and now thousands of compounds) to assess whether they are detected in public data, to give insight into how they distribute taxonomically, and to test how they assort among disease and tissue types-it simply was not possible. Our new method will not solve the structures of all unknown compounds, but it certainly complements existing approaches. A very important part of this work, that we have now emphasized more, is that now that this reference data is available it can be re-used by the community.

To address the point by the reviewer, we have added a dedicated section on limitations after the conclusions in the paper:

“Limitations: Aside from the general mass spectrometry limitations already addressed in the text (viz. metabolites must be sufficiently present in order to be detected and selected for MS/MS acquisition), the limitation of reverse metabolomics is that it requires physical material from which MS/MS data is generated. Herein, we addressed this limitation via combinatorial synthesis, but to use this strategy, the investigator must have some hypothesis with respect to structures of interest. Basic biological building blocks and simple metabolic transformations can lead to millions of candidate molecules but more complex molecules could become more difficult to synthesize and would become a much lower throughput project. Lastly, the 1.2 billion spectra that are part of MASST searches do not encompass all molecules. This limits reanalysis of public data in the discovery phase of this approach. One approach to address this limitation is to encourage public data deposition, particularly with metadata¹⁷. ”

- The sentence “ drugs such as rifamycins that target the pregnane X receptor (PXR), a bile acid receptor, are successfully used to treat IBD” (lines 243-245) does not seem right. Rifamycins are not used to treat IBD, and the cited references do not refer to it.

Thanks for pointing this out. To clarify the text has been modified to read:

“The pregnane X receptor (PXR), a bile acid nuclear receptor involved in xenobiotic transport and metabolism, is also thought to play a pivotal role in IBD. Specifically, PXR agonizing drugs that are semisynthetic derivatives of rifamycin (e.g. rifaximin, rifampicin) have been studied for the treatment of IBD in clinical trials and decreased PXR gene expression has been associated with IBD and Crohn’s disease 47,48,53–60. ”

Referee #2 (Remarks to the Author):

The letter by Gentry et al describe a “synthesis-based reverse metabolomics” approach whereby MS/MS spectra acquired from synthesized compounds are then searched against public metabolomics data to identify these compounds and associate them with disease. The

authors generated 145 amidated bile acids through amidation and confirmed the structures. Searching public databases by MASST and ReDU, as well as separate IBD cohorts, the authors report increased levels of these novel amidated bile acids in IBD. PXR is targeted for IBD treatment by rifampicin, and the authors show that these amino acid-bile acid conjugates activate PXR with similar affinity as rifampicin, although it is not clear what this might mean. Identifying bacteria that can generate these conjugates is also important in terms of linking this to the microbiome and future attempts to identify mechanism of conjugation. Overall, this is a useful approach that can be widely adapted to the study of other metabolites.

We want to thank the reviewer for their time and their comments and for recognizing the value of this approach.

We have now removed the statement about similar affinity to rifampicin.

There are some concerns about novelty given the previous report by the corresponding author of novel bile acid-amino acid conjugates in Nature recently (PMID: 32103176).

We can see how the reviewer came to this conclusion. New structure elucidation strategies are important. Here we not only provide a new structure elucidation platform, but we also provide insight into whether each molecule has ever been detected before, and enable the hypothesis generation about its function by exploiting sample information, disease associations, whether microbes can make the molecule etc. While we had previously published on some of the bile acids, and others confirmed their presence, we now show with other completely different classes of molecules the effectiveness of this metabolomics discovery strategy. In all, we now matched >600 molecules in one paper whose structure might otherwise remain a mystery, while other structure elucidation methods only solve one structure at a time.

Major criticisms

Line 207-209: It is stated that the novel amino acid conjugates are observed at higher frequency in IBD patients, and refers to Fig 2C. The heat map intensity is based on “% of matches”, but the intensity in IBD, UC, and CD are very low compared to “no disease reported”.

This question helped us to brainstorm how to get this point across better, and we have now revised this figure to improve the clarity. It is not correct that the IBD, UC and CD were low intensity, but they appeared as such due to ineffective use of normalization. In the original figure we only plotted the numbers of samples where these samples were found, irrespective of the number of samples in that group. However, the “no disease reported” group is much larger than the others. We therefore now normalized the results by the total number of samples in each group. In addition, we now plot the results as log-scale. We further analyzed

the peak intensities of one of the data sets that had CD, UC and non-IBD controls metadata information as well as publicly available pediatric IBD study (and the data generators are now co-authors) with and without antibiotic treatment. In addition to adding the data from additional classes of molecules, we have modified Figure 2d, added 2e and f and the text to address the above comment.

Figure 2: Repository-scale analysis of public mass spectrometry data. a) Heatmap representing the log value of spectral matches for 1472 *N*-acyl amides in the entire GNPS metabolomics repository. b) Heatmap showing the log value of unique MS/MS spectral matches for each amine conjugation in different tissues and biofluids using GNPS public data with metadata available in ReDU. c) Heatmap showing the log value of unique spectral matches for conjugated bile acids across different tissues and biofluids using GNPS public data with metadata

available in ReDU. d) Heatmap showing the proportion of samples that each MS/MS synthesized bile acid was detected for health phenotypes across the public metabolomics repositories that have metadata available in ReDU. CD, Crohn's disease; UC, ulcerative colitis; IBD, inflammatory bowel disease; disease NOS, disease not otherwise specified. From top to bottom for health phenotype, n=1556, n=679, n=84, n=144, n=207, n=46, n=713, n=317, n=195, n=56, n=59, n=13108 and n=6092. e) Relative MS1 abundances of conjugated bile acids across clinical groups in the single project MSV000084908. CD, Crohn's disease, n=103; UC, Ulcerative colitis, n=60. f) Relative MS1 abundances of conjugated bile acids in relation to antibiotic usage in data from a pediatric IBD cohort deposited as MSV000088735. Antibiotic use, n=72; no antibiotic use, n=45. Boxplots show first (lower) quartile, median, and third (upper) quartile and whiskers are 1.5 times the interquartile range. Statistical significance was tested using a pairwise two-sided Wilcoxon rank sum test. Only significant p-values are shown, which were adjusted using Benjamini-Hochberg correction.

However, in figure 3b. which shows "peak area", there is consistently higher levels relative to "nonIBD", which presumably is the same group as "no disease reported".

We can see how this is confusing, because they are not reporting the same thing. Original figure 2c, now 2d refers to the count of MS/MS matches from all the 1.2 billion MS/MS spectra from nearly 2500 public metabolomics projects that have the IBD metadata label. It enables the formulation of a hypothesis, while the peak area of the MS1 ion is done on a single project using the metadata within a single project.

To clarify, in the legend of Figure 2, we have now explicitly stated if the result was from across the entire GNPS repository, across the repository that has ReDU compatible metadata, or if it was from a single project. In addition, we now explicitly state if it is based on MS/MS or MS1. Please let us know if it is still not clear.

This is confusing, because in figure 3C and 3D it is suggested that these bile acid conjugates are just as effective, if not more, in several cases as rifampicin (treatment) at activating PXR. Then we are told that more studies are needed to determine what is going on with these conjugates and IBD. So it is unclear if these conjugates are low in IBD relative to health (Figure 2C), or significantly higher (Figure 3b), but serve as a potential treatment for the disease state in which they are abundant in (Figure 3C, 3D).

This apparent contradiction was a great catch by the reviewer, and much appreciated. The bile acids are high in IBD and they also are agonists of PXR, the target of rifampicin and analogs, a PXR agonist, which is being explored as a treatment. As usual - human biology is likely more interesting than this paper can describe. Not all IBD patients have large quantities of Glu or Thr conjugates, and they often have some of the other ones that do not show agonistic activity. Only some success was seen with PXR agonists in the clinic. Perhaps it will be a similar story to the PD1 checkpoint inhibitors for melanoma, where microbiota or their metabolites such as these bile acids can be used to predict effectiveness, and even

leveraged to improve effectiveness and will be interesting to learn from the community in the next couple years.

We are now very explicit about this potential hypothesis and have adjusted the text in the following way: “Taken together, these findings support the hypothesis that some of the newly discovered bile acids could play an important role in IBD through PXR and/or immune-mediated processes. We hypothesize that they could also explain why not all patients respond to rifaximin treatment, in cases where they already have large quantities of PXR agonists produced by their microbiota. More work is needed to fully understand our findings, but here we demonstrate that reverse metabolomics can discover new biologically active molecules.”

Line 297-299: It is suggested that 12alpha-dehydroxylation has not been reported. However, there have been reports of this, including recently (PMID:34463573; PMID:6492790).

Thanks for this comment. We are not entirely sure about the relevance of 12alpha-dehydroxylation in the context of the conjugated bile acids, but the first paper is very relevant and very exciting.

This first paper (PMID:34463573) came out during the first round of review - after our submission - but their results nicely support the finding that microbes do indeed make these bile acids as we show in this paper (we cultured 200+, we found 145 in public data, they had ~70 organisms and found ~40 new bile acids with none validated with synthetic standards while we found 145, now and an additional ~450 other molecules, and are all supported with synthetic standards). Without synthetic standards, it is not possible to unambiguously interpret the bile acids core that is attached to the conjugate and thus highlights the strength of our approach as the analysis starting point is the synthetic standard. I wished they had made their data public so we could validate their findings as well, and so it would be included as part of any future MASST search. This paper highlights a method that goes beyond bile acids as it can be applied quite broadly. We are less clear of the relevance of PMID:6492790 in the context of amidated bile acids, because that paper is about the “Estrone sulfatase activity in the human brain and estrone sulfate levels in the normal menstrual cycle.” We would be happy to address that paper if the context could be clarified.

We modified the text to read “While microbial conversion from CA to DCA has been known for decades, this is the first report of bacterial CDCA production from CA, though a recent publication infers that microbes are capable of this transformation (**Extended Data Figure 7b**)⁴⁴. Gram-positive bacteria belonging to the *Bifidobacterium*, *Enterococcus*, *Clostridium*, *Cellulosilyticum*, and *Catabacter* (recently renamed to *Christensenella*) genera⁶², most of which are members of the Firmicutes phylum, were found to produce the most conjugated bile acids in terms of abundance and diversity. These observations align with recent findings that were reported that screened 72 microbes that proposed structures of conjugated bile acids but none of them verified with standards⁴⁴. This work also revealed that gram-negative *Fusobacterium* can also produce conjugated bile acids. This analysis reveals that bacteria are involved in more bile acid

modifications than was previously appreciated, many of which are now also being seen in other studies^{45,63,64}.”

However, the medium that is used, reinforced clostridium medium (Line 567), has peptone. It has been reported that peptone from animal sources contains 9.53 mg/g bile acids (PMID:16347619). Our lab has confirmed this with peptone sources we have obtained by LC/MS. It will be important to know if CDCA is detected and at what concentration in the RCM blank.

Thank you for this comment, which is a good point for us to check. We have now included analysis of media blanks as part of our IMS studies (Extended Data Figure 7) and found that the median peak area of CDCA is at least a 1,000 fold higher in bacterial culture compared to media blanks. Therefore, we can say with confidence that the CDCA was produced by microbes and was not inherently from the peptone media or other sources. This is now shown in Extended Data Fig 7b.

Extended Data Fig. 7: LC-IMS-MS Analysis of Bacterial Cultures. a) Peak area abundances of conjugated bile acids in bacterial cultures vs. media blanks. b) Peak area abundances for unconjugated bile acids in bacterial cultures vs. media blanks.

Minor criticisms

-Line 55 and 300: bacterial genera should either be described as Bifidobacterium or bifidobacteria, for instance.

Thanks, they are now named Bifidobacterium.

-Are any of the bile acid conjugates in Figure 3 c and d significant relative to rifampicin treatment?

Yes Thr-CA would be significant relative to rifampicin treatment ($p=0.0019$) at 50uM when calculated using one-way ANOVA. We have now included this finding in the figure legend.

Extended Data Fig. 6: PXR activity of conjugated bile acids. a) PXR agonism at 10 (left) and 50 μM (right) concentrations of the 15 new conjugated bile acids in the iHMP2 dataset, reported as normalized luciferase luminescence (a.u.). Top 15 bile acids were chosen based on the upper value of their interquartile range. Each concentration was run in biological quadruplicates for every substrate tested. Rifampicin, a known PXR agonist, was used as a positive control for comparison. Thr-CA significantly increases PXR agonism versus rifampicin ($p=0.0019$) using a one-way ANOVA test. b) Concentration dependence of PXR activity for top candidate agonists compared to Rifampicin. c) Gene expression analysis of *Cyp3a11* in small intestinal organoids after exposure to conjugated bile acids at varying concentrations. Compounds are colored as Glu-CA (red), Glu-CDCA (orange), rifampicin (gray). Significance was calculated using one-way ANOVA and is denoted by asterisks: *, $P < 0.05$; **, $P < 0.01$; ***, $P < 0.001$; ****, $P < 0.0001$.

-with rifampicin treatment, the text does not establish the dose being tested is physiologically relevant.

This is a great point. We have now added the concentration of the bile acids in human feces, and can affirm that the doses tested are physiologically relevant. For example, we see activity of as low as 10 μM in intestinal organoid assays, and we see concentrations of up to 19 μM for Phe-CDCA in fecal samples from healthy individuals, if the density of feces is assumed to be 1.06 g/mL (<https://doi.org/10.1016/j.watres.2017.12.063>).

Referee #3 (Remarks to the Author):

Summary:

The workflow described in this paper is to synthesize putative biological compounds, obtain MS/MS on both the orbitrap and QTOF, run MS/MS searches in publicly deposited mass spec data to find matches and then link with phenotype through searching of public metabolomics datasets.

We thank the reviewer for taking the time to read and comment on the paper. These comments made it clear there were important points of clarification that were needed.

Comments:

The authors recently published on a novel class of amino acid conjugated bile acids (Nature, PMID: 32103176). It is reasonable to predict that other amino acid conjugations might also exist. In this work the authors made the conjugations, collected MS/MS data from the standards and then looked for evidence that they were present in public datasets.

We would like to make the important note that at least 20% of the public datasets that are in MASST were generated in the Dorrestein lab, many with consistent methods. Therefore, had we used only data from our own analyses, we would have obtained substantially the same result. However, we would not have had the cross-cohort analyses critical for showing robustness. Therefore this paper is an important demonstration of validating results across multiple labs. It also shows the value of contributing one's own data to larger public resources, as well as of the value of what can be achieved as those resources grow.

This is a sensible and compelling set of experiments to pursue. From this, however, the authors try to abstract a workflow that they call "reverse metabolomics" that they claim is broadly applicable to the discovery of many new classes of metabolites. This reviewer questions whether the presented method will be generally useful for other metabolites.

Thanks for this comment. The other reviewers made the same comment, and we agree that we had not yet demonstrated the generality comprehensively on initial submission. We have now synthesized an additional ~2000 compounds, which includes two classes of molecules that are not derived from bile acids, establishing greater generality. The results are shown in figure 2, extended figure 1 and 2 and we have modified the text accordingly (provided the three figures below).

Figure 2: Repository-scale analysis of public mass spectrometry data. a) Heatmap representing the log value of spectral matches for 1472 *N*-acyl amides in the entire GNPS metabolomics repository. b) Heatmap showing the log value of unique MS/MS spectral matches for each amine conjugation in different tissues and biofluids using GNPS public data with metadata available in ReDU. c) Heatmap showing the log value of unique spectral matches for conjugated bile acids across different tissues and biofluids using GNPS public data with metadata

available in ReDU. d) Heatmap showing the proportion of samples that each MS/MS synthesized bile acid was detected for health phenotypes across the public metabolomics repositories that have metadata available in ReDU. CD, Crohn's disease; UC, ulcerative colitis; IBD, inflammatory bowel disease; disease NOS, disease not otherwise specified. From top to bottom for health phenotype, n=1556, n=679, n=84, n=144, n=207, n=46, n=713, n=317, n=195, n=56, n=59, n=13108 and n=6092. e) Relative MS1 abundances of conjugated bile acids across clinical groups in the single project MSV000084908. CD, Crohn's disease, n=103; UC, Ulcerative colitis, n=60. f) Relative MS1 abundances of conjugated bile acids in relation to antibiotic usage in data from a pediatric IBD cohort deposited as MSV000088735. Antibiotic use, n=72; no antibiotic use, n=45. Boxplots show first (lower) quartile, median, and third (upper) quartile and whiskers are 1.5 times the interquartile range. Statistical significance was tested using a pairwise two-sided Wilcoxon rank sum test. Only significant p-values are shown, which were adjusted using Benjamini-Hochberg correction.

Extended Data Figure 1. Results of MASST searches for *N*-acyl amides. a) Pie chart representing sample types in which the spectra of synthesized amides were detected in ReDU. b) Heatmap representing log(number of spectral matches) across different sample types, organized by fatty acid chain identity. c) Heatmap showing the proportion of samples where *N*-acyl amides were detected in different health-related phenotypes (all the matches to the IBD was from a longitudinal study of a single person).

Extended Data Figure 2: Results of MASST searches for acyl esters. a) Representative heatmap showing log(number of spectral matches) across unique synthesized acyl esters. b) Pie chart representing the number of spectra for synthesized esters detected in different sample types from ReDU. c) Heatmap representing log(number of spectral matches) for acyl esters across different tissue types.

“To assess the capabilities of our “synthesis-based reverse metabolomics” strategy, we aimed to discover and characterize new molecules within four metabolite classes of interest – *N*-acyl amides, fatty acid esters, bile acid esters and conjugated bile acids. In total, MS/MS spectra for 2023 molecules were acquired and 31% of these spectra were detected in public metabolomics data using MASST. Until now, many of the compounds included in our synthetic mixtures had rarely, if ever, been reported in biological samples, largely because their MS/MS spectra were not yet available in public libraries.

Acyl amides and esters

Acyl amides and esters are important signaling molecules in humans whose structures have not been fully identified and characterized^{21–26}. Therefore, we synthesized a library of acyl amides and esters to search for in public metabolomics data via reaction of 46 fatty acyl chlorides with 32 amines and 17 hydroxy acids, respectively, under basic conditions (**Figure 1b**). In total, this synthetic scheme theoretically yielded 1472 acyl amides and 782 acyl esters. We were able to obtain MS/MS spectra of the $[M+H]^+$ adduct for 87% and 72% of all desired amides and esters, respectively. These spectra were searched for in public metabolomics data using MASST. We obtained 60,277 spectral matches to the synthesized *N*-acyl amides across 25,463 unique files and 31% of the compounds searched returned spectral matches (**Figure 2a**). For the esters, 5,884 spectral matches were found across 5,273 unique data files. However, only 15 esters or 1.9% of those synthesized were detected in public data (**Extended Data Figure 2a**), likely because esters do not ionize well in positive mode. However, we opted to consistently query the $[M+H]^+$ spectrum to maximize our search space, since ~90% of all public data is positive mode and little negative mode data is from fecal or tissue samples. Six of the acyl esters detected in public data are previously undescribed fatty acid bile acid conjugates (FABACs). Though several

FABACs have been synthesized for the treatment of gallstones, they have not yet been reported in biological samples, at least to our knowledge²⁷⁻²⁹.

Next, we investigated where these compounds are found in public data and how they associate with health phenotype, using sample information from ReDU. The majority of spectral matches to both the synthetic amides and esters were from animal samples, primarily those from humans and mice (**Extended Data Figure 1a, Extended Data Figure 2b**). The synthesized amides were most frequently observed in fecal, caecum and skin samples while esters were observed across many different tissue types (**Figure 2b, Extended Data Figure 2c**). Some of them were quite common in specific sample types. For example, 48% of human breast milk samples deposited in ReDU contained at least one synthesized ester, with oleic acid ester of hydroxy stearic acid (OAHSAs) being the most frequently observed. *N*-acyl amides were widely detected in microbial cultures (**Extended Data Figure 1b**), but not the esters as only two were found in microbial datasets. MS/MS matches to OAHSAs was detected in cultures of *Vibrio mediterranei* and *Leuconagaricus* and an MS/MS match to the palmitoleic acid ester of hydroxy palmitic acid was found in the *Leuconagaricus* data set. Some *N*-acyl amides were found more frequently in samples labeled as "IBD" (n=84) (**Extended Data Figure 1c**), but this category is comprised of only a single dataset and these associations did not hold up in either the "Crohn's disease (CD)" (n=207) or "Ulcerative colitis (UC)" (n=144) categories, which contain data from other IBD cohorts, highlighting the importance of being able to support an observation across many data sets. In this way, an important benefit of our method is that it enables more confident formulation of potential biomarkers by searching all public data and making associations across multiple datasets of similar or identical phenotype."

The bile acids are an exceptional group of compounds for a few reasons. First the authors already had strong reason to believe that additional conjugates would exist based on their prior work. Second the chloroformate mediated amide bond chemistry is unique and it was straightforward for the authors to produce many different conjugates with high yield. It is not clear that the same strategy will work with other classes of compounds with sufficient throughput, especially without a pre-defined hypothesis and modular chemical functional groups. To demonstrate that "reverse metabolomics" is an effective and efficient strategy to discover unknown molecules the authors need to demonstrate its utility beyond one unique class of compounds.

We agree with the reviewer, and we are pleased to report that our new results eliminate this concern by extending the generality as requested.

Another issue with "reverse metabolomics" as a broad approach is that no assessment is provided for how well it works. MS/MS for metabolites is not very specific and the problem with public databases is that there is no quality control, so the data might not be reliable.

We appreciate the reviewer's concern both about the MS/MS matching methods and the databases, which we have now addressed explicitly in the text.

Our MS/MS matching is exactly the same as the current state of the art used in all untargeted metabolomics papers that rely on MS/MS matching (and, indeed, proteomics projects that rely on MS/MS matching). Thus the underlying algorithms did not change, and we hope the reviewer would agree that these methods, although they have technical limitations, have yielded a large body of useful results and exciting new discoveries. The key advance in this work is how we summarize and how we leverage the resulting information. Because we share the concern about match accuracy, we use strict cutoffs that lead to underreporting, not overreporting.

We note that all untargeted data, no matter how good the data collection, includes poor quality data as there will always be some molecules that are at the signal to noise level, so it is important that the underlying algorithms and then the user-chosen settings minimize false discoveries. If the public data is poor, then it will not rise above the scoring threshold. Fundamentally, the way a molecule fragments under thermal activation does not change and computational approaches can improve matching when different collision energies and or instruments are used. Thus quality control of the spectra against which we search is managed at the MS/MS cleanup when it is uploaded to the GNPS ecosystem. When searching MS/MS at the repository scale that uses data from different instruments, filtering and data processing is key. This has been the key to the success of the GNPS/MassIVE ecosystem, and all its surrounding tools. Perhaps the most underappreciated part is that we apply a filter of keeping the top 6 ions for every 50 Da window and then apply a square root transform.

The effectiveness of such filtering, and the fact that such filtering allows effective cross-comparisons even when data is collected on different instruments, is documented in at least hundreds of papers and there is no other effective way to search 1.2 billion MS/MS spectra at this time (although we are also continuously working of faster approaches that scale to such datasets). We also agree that despite this processing, the user of the output still has to understand the limitations in terms of the depth of information that is obtained. For example, when we search the public data with the MS/MS only (and parent mass as filter) then it is difficult to differentiate cholic acid conjugate from muricholic acid conjugate, but because both of them are trihydroxylated bile acids we can still get a family match. Therefore, we describe them as di and tri hydroxylations in the paper (see Figure 2c, level 2 or 3 according to the 2007 metabolomics standards initiative). One must then perform follow-up studies with the standards to match them with retention time and/or ion mobility, as we did here in the two IBD cohorts and the microbial culture data, respectively so that level 1 identity can be made, again per the 2007 standards). The more MS/MS fragment ions a molecule has, the better the matches will be, as we showed in our 2017 FDR paper. In that work, we show the relationship among number of ions, score and FDR. On average, if one uses 5 ions and cosine >0.7 then one gets ~1% false discovery rate. In this work we only include MS/MS data that have 6 or more fragment ions (and with parent mass this would be a total of 7 ions). The majority of compounds we synthesized gave >6 ions. At the same time, to get the data to level 1, retention time analysis with standards are also performed.

We described these points previously in the text but have now further clarified this in the methods section, please let us know if it is still not sufficiently clear:

“Because it is not possible to perform retention time matching on the repository scale, and the MS/MS spectra of isomers were relatively indistinguishable, it is only possible to specify if the amino acids are conjugated to a di or tri-hydroxylated bile acid when performing a MASST search. This would be level 2 or 3 according to 2007 metabolomics standards initiative guidelines¹⁸. However, targeted analysis was also performed to confirm the presence of these bile acids in human fecal samples and to determine the absolute concentration of various conjugated and unconjugated bile acids representing level 1 (**Extended Data Fig. 3b**). They are not necessarily present in low quantities. Seven of the 15 human fecal samples contained one or more of the new conjugated bile acids at concentrations above their well-studied Gly and Tau counterparts.”

and

“To minimize false discoveries, MASST searches were performed using the following requirements. First, the precursor mass had to be within 0.01 or 0.02 m/z for QE and Q-ToF data respectively, and required a minimum of 6 fragment ions and cosine score of 0.7 to match. MS/MS matches that meet these specified criteria give rise to an FDR <1%³⁷, and matches were further confirmed via manual inspection of mirror plots. MS/MS spectra were also inspected for specific fragment ions diagnostic of their structures. Although this protocol does not guarantee against false discovery of closely related isomers or compounds, we found that it does prevent completely inaccurate identifications. For example, in the case of our ester searches, though we obtained hundreds of MS/MS spectra from the synthetic mixtures, there were only 13 that had matches to public data.”

When searching tens of thousands of public MS/MS patterns it is likely that there will be artificial matches. How frequently does this occur? How do the authors control for false positive hits? The authors should test molecules that do not occur in samples and see what hits are returned.

We thank the reviewer for raising this important question. We would like to clarify the different levels of false discoveries that can be considered. First, there is spectrum-level matching FDR, which we interpret as being implied by this question. Although this FDR is important and is the building block of higher level interpretation, the reverse metabolomics approach aims to assess which datasets and sample types each molecule appears in. One analogy is the difference between spectrum level FDR vs protein level FDR in proteomics. It is this higher-level dataset level FDR that we can assess. It is a very good point that the reviewer suggests: “The authors should test molecules that do not occur in samples and see what hits are returned.” We have performed exactly this test with the conjugated bile acids, which we know do not occur in plant samples. As expected, we do not observe any matches to these molecules in these datasets, giving us confidence that the dataset false discovery rate is low.

With our settings, the spectrum-level match FDR is expected to be <1%, as described in the text. This means <1 out of 100 matches is spurious. This is based on a method where a library is created by taking ions and then redistributing the ions as a new library, although other Bayesian methods also exist (see the cited 2017 Nature Methods paper). In total, of the 2023 MS/MS we searched with the additional molecules, 31% matched. Even with the chosen settings that result in an FDR of <1%, we also did a manual inspection, using mirror plots to ensure that the spectral match made sense (i.e. the ones that show the amino acid loss, the mass of the amino acid itself and the characteristic di or tri-hydroxy bile acid ions. Only the ones where the MS/MS makes sense (where most of the MS/MS ions are used) are included. Finally, one can look for trends across biospecimens because the public data contains data from all types of samples. These include samples from all over the planet, from phones, from homes, building materials, ocean soil, water and terrestrial environment, to cars, bikes, to plants, to microbial, animal data, and human data, and even from beyond our planet, such as samples from the International Space Station. The bile acids only matched to microbial, animal and human data, and only matched to specific organisms rather than at random. However, at this time, this is not quantifiable and requires human interpretation and understanding of the molecules themselves. As the data is clustered, it is possible to go through the mirror plots fairly rapidly, and thus we were able to validate by manual inspection every single plot that we included for the bile acids. Of all the matches that passed our manual mirror plot inspection, not a single bile acid was found in data sets where we did not expect to find them from first principles, such as plants (even though plants are among the most numerous data sources in the repository). The harder problem, that the metabolomics community has not dealt with adequately, is the lack of controlled metadata (contextual information about each biospecimen and the organism and site from which it was collected). ReDU, which we introduced in 2020, was the first metadata validator the community has seen, and it is enabling the interpretative part of reverse metabolomics. We are beginning to address this with ReDU, but also through the launch of FoodMASST, MicrobeMASST and PhyloMASST to drive uniformity in metadata collection to make it easier for the community to understand the inferences in the future. Thus the present work is the beginning of leveraging reverse metabolomics for the community.

A related question is in how many public datasets does the hit have to occur to be reliable? This should be assessed. The fact that some bile acids occur in one matrix for a few rare samples (for example blood or liver) demonstrates the complication. Why do bile acids occur in blood, liver and urine in these samples? Is that real? Is there a biological explanation? Most likely it represents false positives and highlights the need to implement some method to control for nonspecific hits. These questions show why it is important to look at more than one example for quantitative testing of false discovery and true positive statistics.

Related to the question above, it is important to understand what type of false positive matches are meant here. There are broadly two types of error that we can consider. First, a

match between a bile acid and a very similar bile acid that cannot be distinguished by mass spectrometry. Second, a match between a bile acid and a completely different molecule by chance.

In the first case, broadly indistinguishable molecules will be enantiomers or near isomers within the same molecular family, and we acknowledge the limitations of mass spectrometry here.

For the second, we first try to use match settings that approximate 1% FDR in spectral library matching (as written below). However, we acknowledge that searching spectral libraries (consisting of tens to hundreds of thousands of MS/MS spectra) compared to repository searching (millions to billions) presents a larger opportunity for false positive matches. To counteract these issues, we have ensured that the bile acid MS/MS spectra that we have chosen to MASST exhibit rich fragmentation. Given the rich fragmentation, the probability of noise producing a similar spectra is tiny. To get a sense empirically of potential false positives, all conjugated bile acids would only be expected to appear in human samples and not in plant samples. Correspondingly, we observed 0% of the bile acid matches appearing in plant samples, demonstrating a low false discovery rate against human datasets.

With the original set of molecules, the bile acids, we saw that 99% of the spectra matched, but with the additional molecules included this match rate was much lower because the majority did not find any matches (69% of the MS/MS spectra did not find matches in the public domain data). For the acyl esters, 98% of the MS/MS spectra searched did not have matches in public data. This addresses the reviewer's concern that chance matches against any molecule would be frequent across large datasets.

Regarding the observation of the new molecules in blood, liver or urine: the dominant ones are Glycine and Taurine conjugates in blood, liver, and urine, consistent with well established literature. For example it is well known that those specific bile acids are made in the liver. It is not clear why this result would be considered a false positive, because it confirms that the reverse metabolomics method works very well. Those molecules would be expected to be found in those sample types.

The major finding of this work is the report of new bile acids in human samples. To the best of my knowledge this is the first time that these compounds are being reported in human subjects. The burden of proof to show that these are indeed the compounds being measured is high. I am concerned by how poorly some of the MS/MS from standards match the MS/MS from samples. It could easily be that the compounds in the samples are some structural variation of the standard. To name a few--- Ala-aMCA has a fragment at 350 that is not in the sample. Fragments from 200-220 are much larger in the sample than in the standard. Fragment 426 is twice as large in the sample than the standard. Ala-bMCA has a major (almost most dominant fragment) at 126 in the sample that is not in the standard. Ala-CA also

has a major non-matching fragment at 126. The patterns for most Arg-species look totally different between the sample and standard. The ratios between fragments in the citrate gMCA case does not match between standards and sample. Glu-CDCA His-CA are other examples in which the intensities of the fragments actually flip. The low intensity fragment from the sample is the highest in the standard and vice versa. This data alone falls short of the evidence that is needed to convincingly support the publication of new compounds.

We thank the reviewer for going through these examples in such detail. We note that the mirror plots allow one to inspect the consistency of the amino acid conjugate, and whether the bile acid is a di- or trihydroxy bile acid (the level we can uncover when doing repository-scale searches). Aside from four others (one in the original MASST paper and three in the original bile acid discovery paper (Nature 2020), this is indeed the first demonstration that many of these bile acids are found in humans (although there are now other papers that are now confirming this in both humans and microbes). We now not only have the retention time matches from the PRISM and iHMP2 data, and RT matches for the LC-MS/MS and LC-IMS-MS microbial data, but have also quantified these bile acids in healthy human fecal samples using targeted QQQ methods and also quantified them in IBD samples with a qToF. Thus the presence of the bile acids is confirmed using four different mass spectrometry platforms, in four different human cohorts and also bacterial culture, and these identifications are considered level 1 (the best possible identification, according to the 2007 metabolomics standard initiative). Several additional platforms from other labs have now also been able to detect them too. Accordingly, we have now cited several preprints from other labs that are also finding these bile acids in humans.

Regarding the mirror plots, they can all be accessed through the MASST link jobs. But below are some of the examples as visualized using the GNPS dashboard (Petras, Nature Methods 2022). The way MASST (<https://www.ncbi.nlm.nih.gov/pmc/articles/PMC7236533>) currently works and are working on different approaches very similar to how BLAST works for genome repositories. There a precomputed network of sequence relationships can be searched with BLAST and then anything in the search space with your specific search criteria is then listed as a table. By analogy, MASST has a precomputed repository scale molecular network (Watrous PNAS 2012), where MS/MS spectra are searched. To create the network itself the MS/MS subjected to MSCluster (<https://pubmed.ncbi.nlm.nih.gov/18067247/>) and includes the filtering and square root adjustments. It is still among the best performing clustering algorithms. But in the end, it merges similar spectra and then keeps all ions, with the same parent mass, that are found in >60% of the MS/MS spectra that are clustered (while keeping track of what file and data set each of the underlying spectra came from with its metadata). This results in the removal of some ions that might not always be observed such as the ion the reviewer mentioned for Ci-trihydroxybile acids that was obtained through a match to Cit-MCA. Such clustering is always a balance of having too many non-clustered spectra or being greedy that merges related spectra, including isomers. We also apply a square root to lessen the contributions by the intense ions and increase the contributions of the lower ions and thus

intensities are weighed much less than the ion in the right location but allows for improved comparisons cross different instruments and collision energies. While it is possible to separate isomers when retention times are considered (<https://www.nature.com/articles/s41592-020-0933-6>), we currently do not have the technology to scale and build a network to such analysis at the repository scale. Because of this we need to be aware of what we can learn from the data when searched. With the bile acids we explicitly state that they are matched and thus provide annotations to the specific amino acid as you see both the ions at the parent mass loss and the ions at the mass of the aa itself and then we can assess if the characteristic 2 hydroxy and 3 hydroxy bile acids signatures with the ions at 337 = trihydroxy, 339 = dihydroxy are present. Isomeric molecules with the same parent mass are clustered but may have some of the ions being different (we see this with the 126 Da ion for example). It is similar to lipids with one double bond. One can get matches but one will not know exactly where the double bond is nor the stereochemistry and this is why they are defined as CXX:YY, such as C6:1 as a six carbon fatty acid with one double bond for example and not a 2,3-cis-hexenoic acid because the matching does not allow for more information, yet knowing that it is a hexenoic acid helps the interpretation. Finally, when clustering, we also filter out the ions at the parent mass as that is already the first pass filter we used before MSclustering is done as well any ion within the first 17 Da mass loss. Thus amine losses sometimes get filtered out. When displaying the mirror plot, the data is not shown as the square root, rather the actual intensities. The mirror plot shows the clustered and filtered spectra and the unfiltered regular spectrum, even though it is filtered before we do the searches. For the Arg-CA, the difference is that the spectrum shows the parent mass and that was filtered for the merged spectrum.

For example for the match of Arg-CA:

The link to the interactive mirror plot to Arg-CA can be found below (all mirror plots to every MS/MS match are available through the MASST data links, these are permanent links) The green spectra are the merged spectra from the clustered data from MASST while the top spectra in black:

Arg-CA (takes a few seconds to load): http://metabolomics-usi.ucsd.edu/dashinterface?usi1=mzspec%3AGNPS%3ATASK-f99ba4de25184ec29e654414cc959a0a-spectra%2F%3AAscan%3A1&usi2=mzspec%3AGNPS%3ATASK-f99ba4de25184ec29e654414cc959a0a-f.continuous%2Fclustered_data%2FMSV000083004_specs_ms.mgf%3AAscan%3A323198&width=10.0&height=6.0&mz_min=None&mz_max=600.0&max_intensity=150&annotate_precision=4&annotation_rotation=90&cosine=standard&fragment_mz_tolerance=0.1&grid=True&annotate_peaks=%5B%5B175.12054443359375%2C%20565.3980712890625%5D%2C%20%5B116.072998046875%2C%20175.11900329589844%2C%20227.15199279785156

%2C%20337.2510070800781%2C%20494.33599853515625%2C%20547.3839721679688%5D%5D

Cit-gMCA: http://metabolomics-usi.ucsd.edu/dashinterface?usi1=mzspec%3AGNPS%3ATASK-6cd88ffc9cea4b1c8a7c7cef936470ab-spectra%2F%3Ascan%3A1&usi2=mzspec%3AGNPS%3ATASK-6cd88ffc9cea4b1c8a7c7cef936470ab-f.continuous%2Fclustered_data%2FMSV000082969_specs_ms.mgf%3Ascan%3A55994&width=10.0&height=6.0&mz_min=None&mz_max=600.0&max_intensity=150&annotate_precision=4&annotation_rotation=90&cosine=standard&fragment_mz_tolerance=0.1&grid=True&annotate_peaks=%5B%5B159.07801818847656%2C%20319.2448425292969%2C%20337.25518798828125%2C%20495.3228759765625%2C%20469.3442077636719%2C%20513.3331909179688%5D%2C%20%5B159.07699584960938%2C%20337.25201416015625%2C%20319.2430114746094%2C%20469.343994140625%2C%20495.3210144042969%5D%5D

It is surprising that the authors didn't confirm that the MS/MS from their synthesized standards matched MS/MS from human samples collected in their own lab. I understand that it is intriguing to use public data but there are too many questions about its quality and provenance.

Because we showed that they were found across multiple platforms and cohorts in other labs, we originally considered this step redundant, especially because it would have incurred a long IRB delay. In response to this comment, we have now quantified the conjugated bile acids in IBD samples analyzed in our own lab and confirmed that these compounds are indeed elevated in Crohn's disease as we found in the independent iHMP2 and PRISM datasets.

To confirm identification of new compounds, confirmation is needed whereby the authors collect data on samples and standards using the same methods, instruments, voltages etc. The authors should ensure that the MS/MS, retention time and NMR spectra from the standards match human samples analyzed in their own lab. This is the longstanding convention when identifying new natural products.

Confirmation was done with the PRISM and iHMP2 external validation cohorts and with the cultured microbial data that were both collected in our lab and also in an external lab. It strikes us as odd that this reviewer is stuck on the data collection to be in our own lab vs independent external validation, but we now have quantitative data from an additional IBD cohort analyzed in our own lab. The bile acids have now been validated by retention time and MS/MS matching in 4 labs, in multiple cohorts and in some cases on multiple platforms in each lab. The data itself is all publicly available, including the synthesized standards and human datasets, and all can be reused by the community.

We agree that sometimes the only way to determine a structure is to isolate and perform NMR, but that is not the only way to solve a structure, especially when the predicted molecules can be synthesized. I remember a review for a Nature Microbiology paper where the reviewer stated that structures cannot be solved by NMR, too many erroneous structures were solved by NMR, and that the only way to determine a structure is by synthesizing it and comparing retention times: I mention this in order to note that there is diversity of views in the field on the right way to solve structures. In our view, both methods are appropriate, and hundreds of thousands of other papers have used one or the other with success.

The bile acids have been validated by 4 independent mass spec labs - something that far exceeds any untargeted metabolomics structural characterization paper that we are aware of. If there exists in the literature any untargeted metabolomics papers that discovered hundreds of molecules that have been validated so thoroughly, we would very much appreciate it if the reviewer could provide the relevant citations in case we missed something.

The authors make a weak claim that the bile acids are associated with inflammatory bowel disease. To confirm the association, the authors need more rigor using testing and validation cohorts, targeted analysis, absolute quant etc.

We now provide absolute quantification of bile acids using targeted analysis from healthy donors (purchased from a commercial vendor by the Patterson lab) and a new IBD cohort whose samples were collected to assess whether there are metabolomics differences among the dysbiotic state of CD and UC. In some cases, the new bile acids are found to be more abundant than Gly or Tau conjugates. In addition, we now provided additional relative quantitative analysis of two public IBD data sets, which shows that antibiotics alters the relative levels and the levels are higher in CD relative to UC or non-IBD controls. The PRISM and iHMP2 were already two independent validation cohorts outside of our lab, and outside the entire GNPS/MASST ecosystem. In short, all 5 cohorts, irrespective of how they were collected, all tell the same story. These bile acids are not only present in humans but also the levels are higher in fecal samples from individuals with Crohn's. We note that this level of validation cohorts far exceeds what is typical in the field, where one independent validation cohort is often considered sufficient.

Another key question is whether these new bile acids have any biological function. Discovering new molecules that do not mediate any biology is not very interesting. Especially since these compounds are at low levels, it's possible that they are made promiscuously and do not have any biochemical functions.

We appreciate the concern, and as outlined in the reviewer responses above, we have now validated that they are active at physiologically relevant concentrations.

From a mechanistic perspective the work as it stands now is weak. The authors do test the ability of the conjugated bile acids to function as a PPAR agonist in a luciferase assay but that only measures binding and not any potential biological effects. It would also be important to know if the bile acids change PXR signaling or have any down stream effects such as NF-KB or lipid metabolism. Since the bile acids are at low levels and were not detected in the liver it is unclear if any biological effect would occur in the intestine or inflammatory cells in the intestine.

The paper is indeed not intended to be a mechanistic paper, but rather to introduce new methods. We apologize for the lack of clarity on that point. It would clearly be impractical to do full mechanistic follow-up for >600 molecules, but we do now include data concerning additional downstream effects of PXR signaling in small intestinal organoids and we further have effects on IL-17 and IFN- γ production by T cells. We have now also shown that they are not in low concentrations. We hope this is sufficient to address concerns about whether mechanistic follow-up is possible, and point the way towards the large task ahead of the whole field in elucidating the exact mechanisms.

What is the site and mode of activity of these compounds? What is the actual concentration of the compounds?

These are great questions. We have now supplied the concentrations, as requested. Determining the site and mode of activity of each compound is, we believe, beyond the scope of the current work but an important topic for follow-up work.

References, clarity and context:

Some of the language could be toned down. The largest expansion in structures in the 170 year history of bile acids might be a meaningful phrase if the compounds have function, but this has not been demonstrated.

We have removed this statement as requested.

The statement that "identification of metabolites in humans remains challenging" is misleading in that many metabolites (acylcarnitines, amino acids, etc) can be routinely identified and quantified in clinical labs.

We have modified the statement to read "Identification of metabolites in untargeted metabolomics remains challenging." We hope the reviewer will agree that this is accurate, and apologize that the intent of the statement was not clear in the original context.

From a quick search, I also see that the expressions reverse metabolomics and reverse lipidomics have been used by others before. Reverse metabolomics was used in a different

context, but reverse lipidomics has been used in the same way (PMID: 28457845). The authors should acknowledge this.

Thanks for providing the reference. We note that this paper is a perspective, and although it mentions a term “reverse lipidomics”, it is used to explain an entirely different concept. We could not find a paper on reverse metabolomics, though we found a lot of reverse phase metabolomics papers referring to reverse phase chromatography). Four MS labs that are part of this paper also discussed this, and did an extensive literature search, and only found an abstract to a conference but no paper on reverse metabolomics. Therefore, we did not change this nomenclature or add the suggested reference. However, if the reviewer has a more relevant reference we would be happy to cite it, and we are also open to alternative nomenclature for the new approach we are introducing here, although the analogy to “reverse genetics” is important to us and we believe will help others understand its intent.

Reviewer Reports on the First Revision:

Referees' comments:

Referee #1 (Remarks to the Author):

The authors addressed my comments.

Referee #2 (Remarks to the Author):

The authors did an admirable job of extending the work and addressing reviewer comments. This reviewer still has one major issue with the data and description of experiments relating to conversion of CA to CDCA.

Reviewer original comment: "Line 297-299: It is suggested that 12alpha-dehydroxylation has not been reported. However, there have been reports of this, including recently (PMID:34463573; PMID:6492790)."

Thanks for this comment. We are not entirely sure about the relevance of 12alpha-dehydroxylation in the context of the conjugated bile acids, but the first paper is very relevant and very exciting.

This first paper (PMID:34463573) came out during the first round of review - after our submission - but their results nicely support the finding that microbes do indeed make these bile acids as we show in this paper (we cultured 200+, we found 145 in public data, they had ~70 organisms and found ~40 new bile acids with none validated with synthetic standards while we found 145, now and an additional ~450 other molecules, and are all supported with synthetic standards). Without synthetic standards, it is not possible to unambiguously interpret the bile acids core that is attached to the conjugate and thus highlights the strength of our approach as the analysis starting point is the synthetic standard. I wished they had made their data public so we could validate their findings as well, and so it would be included as part of any future MASST search. This paper highlights a method that goes beyond bile acids as it can be applied quite broadly. We are less clear of the relevance of PMID:6492790 in the context of amidated bile acids, because that paper is about the "Estrone sulfatase activity in the human brain and estrone sulfate levels in the normal menstrual cycle." We would be happy to address that paper if the context could be clarified.

Response: Not sure what happened with the PMID number, but the paper was "PMID: **6492798**" "Dehydroxylation of cholic acid at C12 and epimerization at C5 and C7 by Bacteroides species" by Edenharder in 1984. In the paper, he notes that the result could not be repeated. The paper 34463573 doesn't "show their work" relating to these transformations, and some of the results (dehydroxylation followed by hydroxylation) strain the imagination. There is also a report in 1969 regarding 7 and 12 dehydroxylation of bile acids by a soil aerobe, which is not surprising given soil aerobes can completely mineralize bile acids (5378382).

We modified the text to read "While microbial conversion from CA to DCA has been known for decades, this is the first report of bacterial CDCA production from CA, though a recent publication infers that microbes are capable of this transformation (**Extended Data Figure 7b**)⁴⁴.

Response: The experiment is not well described and raises a number of questions in the reviewer’s mind. The cells are grown in RCM and then, “After 48h of incubation, 5uL was taken out of each well and used to inoculate 200uL of fecal culture media (FCM) made according to the method published by McDonald et al66.” In reference 66, the medium composition is described in the following table:

Reagent	Weight (g)
Peptone water ²	2
Yeast extract ³	2
NaHCO ₃ ²	2
CaCl ₂ ¹	0.01
Pectin (from citrus) ¹	2
xylan (from beechwood) ¹	2
Arabinogalactan ¹	2
Starch (from wheat, unmodified) ¹	5
Casein ⁴	3
Inulin (from Dahlia tubers) ⁴	1
NaCl ¹	0.1
K ₂ HPO ₄ ¹	0.04
KH ₂ PO ₄ ²	0.04
MgSO ₄ ⁵	0.01
Hemin ⁵	0.005
Menadione ¹	0.001
Bile salts ¹	0.5
L-cysteine HCl ²	0.5
porcine gastric mucin (type II) ¹	4

So, this leads to confusion about “media blank” and “bacterial culture” actually mean. Does the “media blank” contain the bile salts, or are they omitted? Further, what is the source of the “Bile salts”? The authors of Ref 66 obtained them from Sigma Aldrich. I didn’t see a mention of the vendor where the “bile salts” are purchased from. If you look at some of these (there are many to choose from), they say that they are obtained from dehydrated bile (likely cow). Some say sodium salts, but of what? CA and DCA or TCA and TDCA? If the latter, how do we know that what you are measuring is not the deconjugation of TCDCA resulting in CDCA formation?

Original Comment: “However, the medium that is used, reinforced clostridium medium (Line 567), has peptone. It has been reported that peptone from animal sources contains 9.53 mg/g bile acids

(PMID:16347619). Our lab has confirmed this with peptone sources we have obtained by LC/MS. It will be important to know if CDCA is detected and at what concentration in the RCM blank.”

Thank you for this comment, which is a good point for us to check. We have now included analysis of media blanks as part of our IMS studies (Extended Data Figure 7) and found that the median peak area of CDCA is at least a 1,000 fold higher in bacterial culture compared to media blanks. Therefore, we can say with confidence that the CDCA was produced by microbes and was not inherently from the peptone media or other sources. This is now shown in Extended Data Fig 7b.

Extended Data Fig. 7: LC-IMS-MS Analysis of Bacterial Cultures. a) Peak area abundances of conjugated bile acids in bacterial cultures vs. media blanks. b) Peak area abundances for unconjugated bile acids in bacterial cultures vs. media blanks.

For microbiologists, concentrations would be much more intuitive than peak areas. Again, this figure is not clear based on what is described in methods and supplemental information. Why are you measuring (and detecting) muricholic acids, UDCA, HDCA? What is being added to the culture medium (X micromolar CA + Y micromolar DCA)? It sounds like, from ref 66 that a crude suspension of unknown bile composition is being added. Did you measure TCA, GCA, TCDCA, GCDCA, TDCA, GDCA in the “media blank” as well as CA, CDCA, and DCA? It strains belief that 80% of the cultures convert CA to CDCA and this has been missed for decades. Seems more likely that TCDCA levels are high in the “media blanks”, and CDCA is low. Most bugs you are adding have bile salt hydrolase activity and convert TCDCA to CDCA which you then detect. How do you explain measuring 10X higher CA when you add bacteria than what you find in the “blank media”? If CA is being converted to CDCA, as you state, there should be a series of dots at the low peak area range in CA that correspond with the high CDCA levels you show. It looks like your “bile salts” contain TCA or GCA, and again, bacteria are liberating CA through deconjugation.

What would be convincing is to show which bacteria are converting a known amount of pure, authentic cholic acid to a known amount of CDCA. Doing this over time would be more convincing. To accompany this data should be a blank medium control (no CA added), medium + bacteria (no CA added), medium + CA (no bacteria added). Otherwise, like the other reports of 12-dehydroxylation by gut bacteria, without sufficient support, this strains credulity.

Referee #3 (Remarks to the Author):

It is nice that the authors expanded their study to include two additional classes of compounds. This reviewer still contends that metabolites that can be made with combinatorial synthesis are the minority rather than the majority. The vast majority of polar metabolites are not structurally related by simple modular functional groups. That is not to say that "reverse metabolomics" is insignificant, but its applicability needs to be properly contextualized.

I agree that MS/MS matching is and has been "useful". Stating that something is useful is very different than stating that it is independently sufficient to establish high confidence. This is especially true based on the MS/MS patterns shown in the last submission where many of the fragments did not even correlate. Disappointingly, all of the MS/MS patterns have now disappeared in the new submission. The absence of MS/MS data in the current version of the paper is concerning. In their response the authors include a lot of computational rationalization to justify why certain fragmentation patterns were different between the molecules they synthesized and the molecules in samples. I am weary about having to extensively manipulate the raw data to get matching MS/MS patterns if the compounds are indeed the same. I remain skeptical that the MS/MS data (only shown in the last submission) represent actual metabolite identifications.

The authors have included a new class of compounds in the second submission, FAHFAs. FAHFAs are a good example of why imperfect MS/MS matches can be misleading and cause misidentification. FA dimers can form during LC-MS and produce compounds with the same exact mass and highly similar (but not identical) MS/MS data as FAHFAs. It is essential that the MS/MS patterns be exact matches to support the identification of a new compound. These should be included in the paper.

In my last evaluation I suggested that the authors match MS/MS data from their authentic standards to research samples. They did not do that. Instead they used a QQQ instrument to validate and quantify the concentration of the molecules in their own samples. The major issue in using a QQQ is that it can disguise poorly matched MS/MS patterns. What is necessary to establish high confidence in a new compound identification is a full MS/MS match, which should be included in the paper.

The authors write "it strikes us as odd that this reviewer is stuck on data collection to be in our own lab vs independent external validation". The problem is that the MS/MS data shown in the last submission

did not provide a compelling match between the synthesized standards and the research samples. The only way to convincingly prove that the differences are due to different instruments and workflows is to eliminate the "external" variable by doing the experiments in an identical fashion. If internal MS/MS data, retention time, and collision cross section of the synthesized standard do not exactly match internal MS/MS data, retention time, and collision cross section of the putative compound in the research sample, then it is a different compound.

The authors write: "To minimize false discoveries....the precursor mass had to be within 0.01 m/z for QE". That is a large m/z range for the QE. Why did they make the error so big?

They also write: "To minimize false discoveries....[we] required a minimum of 6 fragment ions". This is a limitation because many metabolites will not have 6 fragment ions.

Regarding the biochemistry part of the manuscript. The positive control (rifampicin) for the dose response is way outside the EC50~500nM-1600nM), it is impossible to determine the actual biological action of these bile acids since the response appears similar to an extremely high dose of rifampicin.

The reporting of the T cell assay is confusing and hard to interpret. Reporting the response to bile acid treatment as the fold induction of IL-17+ CD4 T cells doesn't make sense. These results are typically reported as the percent of IL-17+ or IFN-g+ T cells present after stimulation compared to controls rather than a fold change. What is untreated-naive, unstimulated T cells, stimulated T cells with no bile acids? The normalization is confusing. The manuscript indicates a change in secretion of cytokines but the amount of cytokine secretion wasn't measured only the differentiation of naive T cells into the IL-17+/IFN-g+ CD4+ T cells after stimulation.

Reviewer Reports on the First Revision:

Referees' comments:

Referee #1 (Remarks to the Author):

The authors addressed my comments.

Thanks

Referee #2 (Remarks to the Author):

The authors did an admirable job of extending the work and addressing reviewer comments. This reviewer still has one major issue with the data and description of experiments relating to conversion of CA to CDCA.

Reviewer original comment: "Line 297-299: It is suggested that 12alpha-dehydroxylation has not been reported. However, there have been reports of this, including recently (PMID:34463573; PMID:6492790)." Thanks for this comment. We are not entirely sure about the relevance of 12alpha-dehydroxylation in the context of the conjugated bile acids, but the first paper is very relevant and very exciting and we had cited it in the previous round of the review. This first paper (PMID:34463573) came out during the first round of review - after our submission - but their results nicely support the finding that microbes do indeed make these bile acids as we show in this paper (we cultured 200+, we found 145 in public data, they had ~70 organisms and found ~40 new bile acids with none validated with synthetic standards while we found 145, now and an additional ~450 other molecules, and are all supported with synthetic standards). Without synthetic standards, it is not possible to unambiguously interpret the bile acids core that is attached to the conjugate and thus highlights the strength of our approach as the analysis starting point is the synthetic standard. I wished they had made their data public so we could validate their findings as well, and so it would be included as part of any future MASST search. This paper highlights a method that goes beyond bile acids as it can be applied quite broadly. We are less clear of the relevance of PMID:6492790 in the context of amidated bile acids, because that paper is about the "Estrone sulfatase activity in the human brain and estrone sulfate levels in the normal menstrual cycle." We would be happy to address that paper if the context could be clarified.

Response: Not sure what happened with the PMID number, but the paper was “PMID: 6492798” “Dehydroxylation of cholic acid at C12 and epimerization at C5 and C7 by Bacteroides species” by Edenharder in 1984. In the paper, he notes that the result could not be repeated. The paper 34463573 doesn’t “show their work” relating to these transformations, and some of the results (dehydroxylation followed by hydroxylation) strain the imagination. There is also a report in 1969 regarding 7 and 12 dehydroxylation of bile acids by a soil aerobe, which is not surprising given soil aerobes can completely mineralize bile acids (5378382). We modified the text to read “While microbial conversion from CA to DCA has been known for decades, this is the first report of bacterial CDCA production from CA, though a recent publication infers that microbes are capable of this transformation (Extended Data Figure 7b)44.

Thanks for the correct PMID and the other additional references. Based on these helpful comments we have now removed the discussion on CDCA production altogether (but left the observation in) because the discussion is beyond the scope of the paper. We aim to introduce the concept of reverse metabolomics, and the CDCA production observation was not observed through reverse metabolomics. The goal of the experiment was simply to show that many of the new bile acids are made by microbes.

Response: The experiment is not well described and raises a number of questions in the reviewer’s mind. The cells are grown in RCM and then, “After 48h of incubation, 5uL was taken out of each well and used to inoculate 200uL of fecal culture media (FCM) made according to the method published by McDonald et al66.” In reference 66, the medium composition is described in the following table (I’ve attached a word version with the table):
So, this leads to confusion about “media blank” and “bacterial culture” actually mean. Does the “media blank” contain the bile salts, or are they omitted? Further, what is the source of the “Bile salts”? The authors of Ref 66 obtained them from Sigma Aldrich. I didn’t see a mention of the vendor where the “bile salts” are purchased from. If you look at some of these (there are many to choose from), they say that they are obtained from dehydrated bile (likely cow). Some say sodium salts, but of what? CA and DCA or TCA and TDCA? If the latter, how do we know that what you are measuring is not the deconjugation of TCDCA resulting in CDCA formation?

Original Comment: “However, the medium that is used, reinforced clostridium medium (Line 567), has peptone. It has been reported that peptone from animal sources contains 9.53 mg/g bile acids (PMID:16347619). Our lab has confirmed this with peptone sources we have obtained by LC/MS. It will be important to know if CDCA is detected and at what concentration in the RCM blank.” Thank you for this comment, which is a good point for us to check. We have now included analysis of media blanks as part of our IMS studies (Extended Data Figure 7) and found that the median peak area of CDCA is at least 1,000 fold higher in bacterial culture than in

media blanks. Therefore, we can say with confidence that the CDCA was produced by microbes and was not inherently from the peptone media or other sources. This is now shown in Extended Data Fig 7b. Extended Data Fig. 7: LC-IMS-MS Analysis of Bacterial Cultures. a) Peak area abundances of conjugated bile acids in bacterial cultures vs. media blanks. b) Peak area abundances for unconjugated bile acids in bacterial cultures vs. media blanks.

For microbiologists, concentrations would be much more intuitive than peak areas. Again, this figure is not clear based on what is described in methods and supplemental information. Why are you measuring (and detecting) muricholic acids, UDCA, HDCA? What is being added to the culture medium (X micromolar CA + Y micromolar DCA)? It sounds like, from ref 66 that a crude suspension of unknown bile composition is being added. Did you measure TCA, GCA, TCDCA, GCDCA, TDCA, GDCA in the “media blank” as well as CA, CDCA, and DCA? It strains belief that 80% of the cultures convert CA to CDCA and this has been missed for decades. Seems more likely that TCDCA levels are high in the “media blanks”, and CDCA is low. Most bugs you are adding have bile salt hydrolase activity and convert TCDCA to CDCA which you then detect. How do you explain measuring 10X higher CA when you add bacteria than what you find in the “blank media”? If CA is being converted to CDCA, as you state, there should be a series of dots at the low peak area range in CA that correspond with the high CDCA levels you show. It looks like your “bile salts” contain TCA or GCA, and again, bacteria are liberating CA through deconjugation.

What would be convincing is to show which bacteria are converting a known amount of pure, authentic cholic acid to a known amount of CDCA. Doing this over time would be more convincing. To accompany this data should be a blank medium control (no CA added), medium + bacteria (no CA added), medium + CA (no bacteria added). Otherwise, like the other reports of 12-dehydroxylation by gut bacteria, without sufficient support, this strains credulity.

We thank the reviewer for these comments, which raise many good points. We have now clarified the growth conditions further describing the sources in more detail. As the reviewer correctly observed, the background of the media is more complex than we initially appreciated, as the description by SIGMA describes it as 50:50 CA and DCA (and thank you for pointing this out), and indeed it does contain other bile acids but not any of the new conjugates. For the bile acids we used Sigma: <https://www.sigmaaldrich.com/US/en/product/sial/48305>. It was 500mg/L so roughly ~1.2mM each. The product lists the “composition as cholic acid sodium salt, ~50%, deoxycholic acid sodium salt, ~50%”. We are measuring other bile acids in this same experiment because this is untargeted metabolomics data, and therefore we can go back to see if other molecules are present based on the MS/MS, RT and/or CCS values, as we did based on the reviewer’s original request for a different set of bile acids. You can think of such an untargeted metabolomics experiment akin to a metagenomics experiment. Maybe one reports on the presence of one gene initially, but it is still possible to look and find other genes in that metagenomics data set once you know what gene you want to look for. For this experiment, looking up the presence of the other bile acids is a manual process but can be done, as we did in the previous round of review. We checked for evidence of TCA, GCA, TDCA, GDCA, TCDCA and GCDCA, and have now provided this additional information (see figure below). What we can say with 100% certainty is that the new conjugates are not detected without the addition of microbial cultures or at t=0 after adding the microbes. We agree with the reviewer there is a lot more microbiology to understand, and a lot more work is ongoing beyond the initial discoveries made by reverse metabolomics. Given the length of this review process, separate papers that are now drilling much deeper into the microbiology. For example, demonstrating that a single dual function enzyme (BSH) can produce the new conjugates from the taurine conjugates and free acids (<https://www.researchsquare.com/article/rs-2050406/v1>,

<https://www.researchsquare.com/article/rs-2050120/v1>, and others). The quantification of conversions by microbes is also being done in those papers. It will be really interesting to begin understanding the substrate specificities of the 2300+ BSH genes found in human microbiomes, especially now that we have discovered another 5,734 modifications using a new algorithmic approach. We agree this data generated many microbiology questions beyond the simple yes or no question: do microbes produce these bile acid conjugates? We have therefore now removed discussion of the idea that there is potential CDCA production from CA, although we still believe it is relevant, because it distracts from the reverse metabolomics method. Reverse metabolomics is the key tool we are introducing in this paper, and it deserves its own separate paper that is not distracted by this observation. We have now explicitly described that the media contains Taurine and Glycine bile acids as well and have adjusted the text and methods accordingly (and added another SI figure 7 shown below).

A third paper used the standards created in this work to show the impact of BSH on bile acid production and germination of *C. diff* (see <https://www.nature.com/articles/s41564-023-01337-7>). We also know this is only the tip of the iceberg because we now have a new algorithm that is allowing us to observe a lower bound of 5,734 different bile acid modifications - this paper will be submitted in the next few weeks as well. Both the much larger diversity of bile acid discovery and the discovery of the dual function of bile salt hydrolase and the large number of modifications became possible due to the reverse metabolomics-based discovery outlined in this paper. We therefore re-focused our response to provide a better description of the media (within the constraints that the media and our data allows) and its source, and added the additional information with respect to the Gly and Tau conjugates we had seen in the original data.

We have adjusted the main text as follows:

Each strain was cultured in duplicate in amino acid-rich media that included mucin, and ~500 mg/L supplemental bile acid mixture including CA, DCA, CDCA and their taurine and glycine conjugates for 72h (**Extended Data Figure 7a,b and c**).

Similar to our LC-MS/MS results in positive ionization mode, we observed a variety of amino acid conjugations in the culture extracts, 18 in total, that matched to CA, DCA. There were also CDCA amidates, specifically those conjugated with polar amino acids Arg, Asp, and Glu. Notably, Arg conjugates were exclusively matched to the CDCA form⁴⁴ (**Figure 4c, Extended**

Data Figure 7a). None of the new bile acid conjugates were observed in media blanks or at t=0 time point of incubation.

The methods section now reads:

Anaerobic Culturing of HMP Isolates

200µL of reinforced clostridial media (RCM, BD Difco) was added to each well within a shallow 96-well plate, then 5µL of inoculum containing different HMP strains was carefully transferred to the plates using a multichannel pipette, with very gentle mixing. Then, the culture plate was sealed with a breathable film cover (Breathe-Easy) and allowed to incubate anaerobically for 48h at 37 °C taking the OD every 30 min. After 48h of incubation, 5µL was taken out of each well and used to inoculate 200µL of fecal culture media (FCM) made exactly according to the method published by McDonald et al⁷⁰. Bile salts suitable for microbiology, purchased from Millipore Sigma (Catalog No 48305), were components of this media. The product lists the composition as ~50% deoxycholic acid sodium salt and ~50% cholic acid sodium salt. Analysis of the media blanks shown in Extended Data Figure 7a,b and c reveal there are also taurine and glycine conjugates. It was added at a concentration of 500 mg/L. Three plates were prepared in parallel using the above methods. One plate (t=0) was covered with an aluminum sealing film and immediately frozen at -80°C upon inoculation to serve as a baseline sample. The other two plates were sealed with a breathable film cover and incubated anaerobically for 72h at 33 °C, with the optical density (OD) being recorded every 30 min. Following incubation, samples were sealed with an aluminum sealing film (AlumaSeal), frozen and stored at -80 °C until extraction. In each of these plates, there were 15 wells that were not inoculated with bacteria, which served as media blanks.

Referee #3 (Remarks to the Author):

It is nice that the authors expanded their study to include two additional classes of compounds. This reviewer still contends that metabolites that can be made with combinatorial synthesis are the minority rather than the majority. The vast majority of polar metabolites are not structurally related by simple modular functional groups. That is not to say that “reverse metabolomics” is insignificant, but its applicability needs to be properly contextualized.

Thanks for the comment on the expansion. Based on the reviewer’s comments, we have recontextualized the text, because it is the reverse metabolomics that is the important concept. First, we have removed synthesis from the title. It now reads “Reverse Metabolomics for the Discovery of Chemical Structures from Humans and Animals.”

As reverse metabolomics can leverage data from entire repositories, we have now recontextualized the synthesis better as being one approach to get the MS/MS spectra, because how we get the MS/MS spectra to search is not relevant to the search process itself. We do, however, still remain of the opinion – and thus disagree with the reviewer - that synthesis will open up a large portion of reverse metabolomics searches be it via one MS/MS at a time or large number of MS/MS at a time (as we only know of a handful of natural molecules that synthetic experts have not been able to synthesize) but that the MS/MS could also be obtained from in silico predictions, or pattern based analysis (the latter has now led us to find 5,734 bile acid modifications - a separate paper we are currently writing up, reflecting the fact that this research area is rapidly changing).

We made changes to the title and text to de-emphasize the synthesis part of the paper per the reviewer's comments.

I agree that MS/MS matching is and has been "useful". Stating that something is useful is very different than stating that it is independently sufficient to establish high confidence. This is especially true based on the MS/MS patterns shown in the last submission where many of the fragments did not even correlate. Disappointingly, all of the MS/MS patterns have now disappeared in the new submission. The absence of MS/MS data in the current version of the paper is concerning. In their response the authors include a lot of computational rationalization to justify why certain fragmentation patterns were different between the molecules they synthesized and the molecules in samples. I am weary about having to extensively manipulate the raw data to get matching MS/MS patterns if the compounds are indeed the same. I remain skeptical that the MS/MS data (only shown in the last submission) represent actual metabolite identifications.

In this paper, we are running against a synthetic standard, in four labs and using a total of 8 different mass spectrometry platforms. That is currently the highest level of identification using LC-MS methods and the highest standard of identification that the metabolomics standards initiative, generated by the metabolomics society, identifies at the present time. Similarly, Schymanski rules of confidence reporting with respect to annotation give this a level 1 as well (Schymanski, E. L. et al. Identifying Small Molecules via High Resolution Mass Spectrometry: Communicating Confidence. *Environ. Sci. Technol.* 48, 2097–2098 (2014)). The reviewer brings up many good points that are general to all targeted and/or untargeted metabolomics/mass spectrometry experiments, including what is already being used in targeted assays to diagnose inborn errors of metabolism (every baby born for the past few decades is analyzed by mass spectrometry methods that all suffer from some of those limitations). Every recent paper published in *Nature* that uses mass spec or LC- based metabolomics experiments (and we could even extend these same limitations to mass spectrometry-based proteomics experiments) method in any form also shares these limitations. Here are some representative recent papers for comparison: <https://pubmed.ncbi.nlm.nih.gov/34262212/>, <https://www.nature.com/articles/s41586-019-1237-9>, <https://www.nature.com/articles/s41591-022-01688-4>, <https://www.nature.com/articles/s41551-022-00999-8>, <https://www.sciencedirect.com/science/article/pii/S0092867420305080>. Because there are always some limitations to the information that mass spec can provide (there is no way around this issue), we clearly highlight the levels of annotation for each of the molecules according to the metabolomics standards initiative and now also the Schymanski rules for annotation in the additional text provided in this revision.

The standards for this paper were run in all four labs across the country so they could observe them in their data sets and perform RT matching and/or quantify the bile acids on 8 different instrument platforms. Now these standards they are in the hands of dozens of labs across the world, as this work is the only source that is making these standards available to people that request them, changing how we must think about bile acid biology. There are also a number of other papers in review at *Nature* (at least 3 that I am aware of, one of them has been accepted) and another published in *Nature Microbiology* that use these standards and usually only one of the mass spec methods, not all 8. But what we now believe is that this reviewer wants this content not in electronic format linked to the repository (as is mandated by federal agencies) but rather they want a PDF (which does not adhere to current federal policies because it is not reusable).

At the same time, we are also convinced that we don't have a fundamental disagreement with this reviewer about what is required for annotations in LC-MS metabolomics experiments, but we take a different approach as to how this information is conveyed and

documented and becomes accessible. We believe that this information should be computationally accessible so anyone could reuse this information (and thus reusable and allows for reinterpretation of every ion every feature, etc.), while this reviewer prefers a PDF of this information. We estimated that including every entry in every file would result in >100K page PDF as SI. In discussion with the editor it was considered acceptable to provide representative examples, and then the tables with information of parent mass, ppm error, RT, and CCS values, to provide a description on the general capabilities in mass spectrometry based experiments in the context of each experiment. We had asked if we could get this reviewer to write a specific commentary on this paper describing how annotation is done in metabolomics, but the editor suggested that we should provide this in the SI instead.

This is what we wrote in the SI with respect to the annotations (we apologize for the long description):

SI extended discussion about mass spectrometry (General description of annotations used in untargeted metabolomics with representative examples of what to pay attention to as one interprets mass spectrometry data)

As we went through the review process of this paper, it became clear that it was important to educate the general community about what is and is not possible within the context of each of the mass spectrometry experiments in this paper, most of which were only fleetingly described, due to space limitations in the paper. These are general aspects that should always be considered in mass spectrometry-based metabolomics – and similar considerations exist in other approaches such as NMR and proteomics. As the use of mass spectrometry-based metabolomics is rapidly expanding, to not properly contextualize the confidence of mass spectrometry annotations is raising concerns among leading mass spectrometry metabolomics practitioners and we hope that this description will help with ensuring that the community better understands how to properly interpret mass spectrometry based annotations^{5,6}.

This paper introduces reverse metabolomics analysis at the repository scale that uses MS/MS spectra to assess if previously unknown structures have been detected but not annotated in public data. This was simply not possible and this is the first demonstration that this can be done in the context of -and at the scale of- an entire repository. Using reference libraries for MS/MS matching is the very foundation of discovery based untargeted metabolomics analysis.

In general, mass spectrometry (or even NMR) based metabolomics annotations can be thought of as filters for results among all the possible hypothetical structural possibilities. The best annotation of a metabolite would be to isolate the molecule and do full NMR, X-ray, MS and biosynthetic gene cluster analysis. However, this is not compatible with thousands of metabolites in a single sample nor practical to large scale analysis and it is not compatible with reanalysis of public data. Care should be taken to not over interpret the results from a mass spectrometry experiment (holds true for mass spectrometry-based experiments beyond metabolomics as well).

Annotation in mass spectrometry is an exercise that aims to narrow down (filters) the best candidate ion from a structure(s) that is(are) consistent with the data and that are possible in the ocean of all possible structures and their ion forms. Annotations based on precursor mass (MS1) alone are generally considered unacceptable by the metabolomics community⁵. It is even hard to predict the ion form and molecular formula from MS1 alone^{7,8} However, there are exceptions. If you are working with specific organisms known to make a particular metabolite that is rather unique (e.g. surfactin for *B. subtilis*) then the MS1 filter plus the information about the organism may be sufficient to make a reasonable assignment. If one knowingly consumes Natto (a food made with *B. subtilis*), for example, and one sees the same masses and has *B. subtilis* in sequencing data, then it is also a reasonable inference based on MS1. However, for

samples and masses that are not so unique as surfactin, and where one does not have additional information, there are generally too many structures possible with MS1 matching when one looks at structural databases such as PubChem9 or HMDB10 and includes all possible ion forms. It is for this reason that it is best to avoid annotation based on MS1 alone. Practice of MS1 based annotations often leads to over interpretation⁵. MS1 based annotations are a level 5 annotation, based on the Schymanski rules¹¹, and level 4 according to the 2007 metabolomics standards initiative¹². MS1 based annotation was not performed in this paper.

The next level of annotation, which is the foundation of reverse metabolomics, is using MS/MS spectral reference libraries as we applied in Figure 2 a,b,c. This leads to level 2 according to both the 2007 metabolomics standards initiative as well as Schymanski rules of confidence and defined as probable annotation. However, as described in the paper, we often refer to it as a molecular family match or level 3 as MS/MS matching alone often does not differentiate isomers. Thus in LC-MS/MS based metabolomics experiments these annotations are accomplished by first filtering the data and the MS/MS library for the precursor match (MS1). This is then followed by MS/MS matching to assign putative structures. The false discovery rate (FDR) of the MS/MS spectral matching is sensitive to the minimum number of ions included in the MS/MS match and the magnitude of the user-defined cosine score. The FDR of spectral matching can be estimated using a decoy MS/MS library. Generally, a score of cosine of >0.7 and 6 or more MS/MS ions that are required to match lead to <1% FDR⁸. When fewer ions are used to match the cosine needs to be raised or when more ions are matched the cosine score can often be lowered to obtain reliable matches. MS1 and MS/MS matching against MS/MS of standards was done as part of the reverse metabolomics workflow. This allowed us to assess the distribution of MS1 and MS/MS patterns (here obtained from synthesis but could be from any other approach to get MS/MS spectra) among biofluids, tissues, organism, disease states etc. When the MS/MS spectra of our synthetic standards were matched with public data using MASST, we defined it as a specific class rather than assigning a specific structure. For example, if we got an MS/MS match for the phenylalanine cholic acid amidate, other phenylalanine conjugated tri-hydroxylated bile acids cannot be ruled out. Therefore, we describe such matches as trihydroxylated bile acid conjugates (abbreviated as Phe-(OH)₃) to include all possible isomers. As there is not yet a standardized nomenclature for bile acids, this is the nomenclature we used in Figure 2e and f. On the other hand, fatty acids have a more defined nomenclature. For MS/MS matching of fatty acids, we can establish the number of carbons and double bonds, but cannot at this level of comparison to standards, establish stereochemistry (cis/trans double bonds) nor the position of the double bond. Thus a notation of C_x:y is used where x is number of carbons and y is number of unsaturations (typically double bonds but could also be cyclopropyl or similar reasons for unsaturation). We used this nomenclature in Figure 2a. To provide an analogy for readers that are more familiar with microbiome sequencing data, this is akin to amplification of the 16S ribosomal RNA V4 region, which typically allows matching of taxonomy at the genus level but generally not at the species or strain level. In sequencing one, would need to sequence a larger portion of the 16S rRNA and or obtain additional data of other marker genes to be able to narrow down to the species, or in some cases, whole genome sequences might be needed to obtain strain information. Similarly, additional data will be needed to further narrow down the possible isomers.

To further narrow down the structural hypothesis, retention times against authentic standards run under the same chromatographic conditions can be used. This is considered the highest level of annotation, a level one annotation according to the 2007 metabolomics standard initiative as well as the Schymanski levels of confidence. While the majority of isomers can be resolved in this fashion, not all isomers can be distinguished, especially stereochemistry. This is the experiment that was used in this paper for the validation of the presence of the metabolites in IBD cohorts and microbial cultures (Figure 3a, 4a,b). In this case most bile acid isomers we had synthesized could be separated based on retention times. However, some isomers such as

Met-CDCA/Met-DCA or Tyr-HDCA/UDCA as denoted in the manuscript in Figure 3a were not separated by retention time. Both the metabolomics standards initiative nor the Schymanski rules are adequate descriptions for this ambiguity as MS1, MS/MS and retention time matching with standards is considered the highest level of annotation for untargeted metabolomics experiments. Although there are relatively few of such instruments available for routine metabolomics, ion mobility drift time against the drift time of authentic standards can be further used to resolve isomers^{16–18} This was the experiment used to further validate the structures of the bile acids produced by microbes in culture in this paper (Figure 4c). Even when applying this many filters (MS1, MS/MS, retention time and/or collisional cross section from ion mobility data) to narrow down the best candidate structure, any annotation obtained by mass spectrometry is - and will always be- a structural hypothesis. This is even true for NMR-based structure elucidation, IR and other structure elucidation and this is also true for targeted assays such as a triple Q used here for quantification and widely used in the clinic for diagnosis (e.g. every baby heel prick in the last 25 years that looks for in-born errors of metabolism).

We also added the following tables with representative matching figures.

Supplementary Table 6: PDF of representative mirror plots of MS/MS matches against the repository for acyl amides and acyl esters generated from the data links provided in the paper. Provided as separate document. A screenshot below.

Supplementary Table 7: PDF of the table of precursor masses, their ppm difference, the retention time matches and representative chromatography traces compared to standards to microbial cultures on a Q-TOF. Provided as separate document. A screenshot below.

Arginine conjugated deoxycholic acid (Arg-DCA)

Supplementary Table 8: PDF of the table of precursor masses, their ppm difference, the retention time matches and representative chromatography traces compared to standards if the iHMP IBD data on a QE. Provided as separate document. A representative screenshot.

Citrulline conjugated chenodeoxycholic acid (Cit-CDCA)

Citrulline conjugated cholic acid (Cit-CA)

Supplementary Table 9: Representative of predicted and observed precursor masses, their ppm errors, MS/MS and retention times of bile acid conjugates on a QE. Provided as separate document. Representative screenshot of the PDF below

Tyrosine conjugated chenodeoxycholic acid (Tyr-CDCA)

Supplementary Table 10: The precursor values, their ppm, their retention times, CCS values and key confirmatory MS/MS fragment on TIMS-TOF. Provided as separate document. See table below.

Molecule	Precursor m/z (M-H) ⁻	Retention Time	CCS	Precursor Mass Error (ppm)
Ala_CA	478.3174	5.6	206.1872937	0.7
Arg_CA	563.3814	5.55	221.3069634	1
Asn_CA	521.3232	5	216.441312	-3.3
Asp_CA	522.3072	4.5	216.4309174	-2.5
Glu_CA	536.3229	4.45	217.9987433	0.3
Gln_CA	535.3389	5.1	216.2968141	-2.7
His_CA	544.3392	5.3	219.7046339	1.1
Ile_CA	520.3644	7.45	219.9521522	1.6
Leu_CA	520.3644	7.35	219.9521522	1.6
Lys_CA	535.3753	5.5	219.7946156	2.5
Met_CA	538.3208	6.7	221.5507767	1.6
Phe_CA	554.3487	7.65	224.9618874	1.8
Pro_CA	504.3331	5.9	214.8380734	-2.3
Ser_CA	494.3123	5.2	208.6876676	-3.3
Thr_CA	508.328	5.45	213.8993266	1.9
Trp_CA	593.3596	7.45	227.2788952	-1.2

Tyr_CA	570.3436	5.8	226.5937087	-0.2
Val_CA	506.3487	6.7	214.8155765	-1.2
Cit_CA	564.3654	5.05	221.2978266	-1.2
Orn_CA	521.3596	5.5	216.4409268	0.2
DOPA_CA	586.3385	5.55	222.8841325	-1.9
Ala_CDCA	462.3225	6.95	205.4878342	1.2
Arg_CDCA	547.3865	6.95	221.4600311	-0.4
Asn_CDCA	505.3283	6.1	215.7214115	5
Asp_CDCA	506.3123	5.25	214.8159812	-0.1
Glu_CDCA	520.328	5.2	216.4518645	1
Gln_CDCA	519.344	6.2	217.2816853	3.2
His_CDCA	528.3443	6.5	219.8673042	2
Ile_CDCA	504.3694	8.4	218.3411748	2.2
Leu_CDCA	504.3694	8.3	218.3411748	2.2
Lys_CDCA	519.3803	7.5	217.2812968	-3.3
Met_CDCA	522.3259	7.8	222.6121694	0.4
Phe_CDCA	538.3538	8.5	225.1225458	0.6
Pro_CDCA	488.3381	7.2	211.441956	-0.3
Ser_CDCA	478.3174	6.4	207.9788803	0.6
Thr_CDCA	492.3331	6.8	212.2905087	0.4
Trp_CDCA	577.3647	8.3	233.6614775	-0.8
Tyr_CDCA	554.3487	7.1	228.5314431	1.5
Val_CDCA	490.3538	7.85	214.1039379	-1.4
Cit_CDCA	548.3705	6.2	222.3429826	-0.1
Orn_CDCA	505.3647	7.3	214.8265372	-3
DOPA_CDCA	570.3436	6.9	222.134774	-0.3
Ala_DCA	462.3225	7.3	202.7978842	1.5
Arg_DCA	547.3865	7.25	220.5673693	-1.1
Asn_DCA	505.3283	6.4	212.1435392	5.1
Asp_DCA	506.3123	5.45	213.9215587	1.1
Glu_DCA	520.328	5.45	214.6642829	1.1
Gln_DCA	519.344	6.6	215.5685034	3.5
His_DCA	528.3443	6.9	218.9738598	2
Ile_DCA	504.3694	8.75	213.943154	2.2
Leu_DCA	504.3694	8.6	216.6266921	2.2
Lys_DCA	519.3803	7.8	219.0689617	-3.5
Met_DCA	522.3259	8.1	214.6433132	0.2
Phe_DCA	538.3538	8.9	218.8713622	0.6
Pro_DCA	488.3381	7.6	213.2325256	-0.2
Ser_DCA	478.3174	6.75	204.3957071	0.7
Thr_DCA	492.3331	7.15	209.6052454	0.4
Trp_DCA	577.3647	8.6	223.8546134	-0.9
Tyr_DCA	554.3487	7.35	220.4999428	1.4

Val_DCA	490.3538	8.15	212.313568	-1.5
Cit_DCA	548.3705	6.5	220.5577369	0.2
Orn_DCA	505.3647	7.6	214.8265372	-3.3
DOPA_DCA	570.3436	7.2	224.8101348	-0.4
Ala_HDCA	462.3225	5.25	205.4878342	1.6
Arg_HDCA	547.3865	5.2	221.4600311	-0.7
Asn_HDCA	505.3283	4.7	216.6158796	5.2
Asp_HDCA	506.3123	4.3	214.8159812	1
Glu_HDCA	520.328	4.3	214.6642829	1.5
Gln_HDCA	519.344	4.8	219.0693534	3.6
His_HDCA	528.3443	4.9	220.7607486	2.2
Ile_HDCA	504.3694	7.05	218.3411748	2.2
Leu_HDCA	504.3694	6.95	218.3411748	2.2
Lys_HDCA	519.3803	5.7	219.0689617	-3.7
Met_HDCA	522.3259	6.2	220.8247624	0.3
Phe_HDCA	538.3538	7.2	225.1225458	0.7
Pro_HDCA	488.3381	5.6	212.3372408	-0.4
Ser_HDCA	478.3174	4.9	207.9788803	0.8
Thr_HDCA	492.3331	5.1	213.1855964	0.5
Trp_HDCA	577.3647	6.95	230.0953451	-0.9
Tyr_HDCA	554.3487	5.4	227.6390541	1.7
Val_HDCA	490.3538	6.3	214.1039379	-1.5
Cit_HDCA	548.3705	4.8	223.2356054	0.1
Orn_HDCA	505.3647	5.5	215.7210035	-3.1
DOPA_HDCA	570.3436	5.1	230.1608564	-0.3
Ala_UDCA	462.3225	5.1	205.4878342	1.4
Arg_UDCA	547.3865	5.05	224.1380165	-0.5
Asn_UDCA	505.3283	4.6	215.7214115	4.7
Asp_UDCA	506.3123	4.2	216.604826	1.3
Glu_UDCA	520.328	4.2	214.6642829	1.3
Gln_UDCA	519.344	4.7	217.2816853	3.1
His_UDCA	528.3443	4.85	220.7607486	2
Ile_UDCA	504.3694	6.8	217.4466621	2.1
Leu_UDCA	504.3694	6.7	218.3411748	2.1
Lys_UDCA	519.3803	5.5	219.9627941	-3.5
Met_UDCA	522.3259	6	219.0373554	0.4
Phe_UDCA	538.3538	7	223.3364933	0.3
Pro_UDCA	488.3381	5.3	212.3372408	-0.5
Ser_UDCA	478.3174	4.75	207.9788803	0.6
Thr_UDCA	492.3331	4.95	212.2905087	0.4
Trp_UDCA	577.3647	6.75	226.5292127	-0.8
Tyr_UDCA	554.3487	5.3	222.2847207	1.5
Val_UDCA	490.3538	6	214.1039379	-1.5

Cit_UDCA	548.3705	4.7	223.2356054	0
Orn_UDCA	505.3647	5.3	215.7210035	-3.1
DOPA_UDCA	570.3436	5.1	223.0265609	-0.5
Ala_MCA (a)	478.3174	4.4	207.083087	0.7
Arg_MCA (a)	563.3814	4.3	223.091053	1
Asn_MCA (a)	521.3232	4	217.2605803	-3.5
Asp_MCA (a)	522.3072	3.45	218.1438506	1.9
Glu_MCA (a)	536.3229	3.5	218.8918532	0.7
Gln_MCA (a)	535.3389	4.1	220.6881376	-2.7
His_MCA (a)	544.3392	4.2	222.3829841	1.1
Ile_MCA (a)	520.3644	5.65	220.8459414	1.5
Leu_MCA (a)	520.3644	5.55	219.9521522	1.5
Lys_MCA (a)	535.3753	4.6	219.7946156	2
Met_MCA (a)	538.3208	5	223.3368318	1.5
Phe_MCA (a)	554.3487	5.8	226.7466652	1.6
Pro_MCA (a)	504.3331	4.7	214.8380734	-2.8
Ser_MCA (a)	494.3123	4.1	209.5826589	-3.3
Thr_MCA (a)	508.328	4.3	214.793656	1.9
Trp_MCA (a)	593.3596	5.6	235.2976884	-1.1
Tyr_MCA (a)	570.3436	4.5	231.0526434	-0.2
Val_MCA (a)	506.3487	5	215.7099972	-1.1
Cit_MCA (a)	564.3654	4.1	224.8658586	-1.3
Orn_MCA (a)	521.3596	4.5	217.2601936	1
DOPA_MCA (a)	586.3385	4.4	223.7753499	-2
Ala_MCA (b)	478.3174	4.4	207.083087	0.8
Arg_MCA (b)	563.3814	4.35	222.1990082	1.1
Asn_MCA (b)	521.3232	4	217.2605803	-3.2
Asp_MCA (b)	522.3072	3.45	217.2501463	1.9
Glu_MCA (b)	536.3229	3.5	216.2869495	1.1
Gln_MCA (b)	535.3389	4.1	216.2968141	-2.8
His_MCA (b)	544.3392	4.2	221.4902007	1.1
Ile_MCA (b)	520.3644	5.7	219.9521522	1.9
Leu_MCA (b)	520.3644	5.6	219.9521522	1.9
Lys_MCA (b)	535.3753	4.1	216.2964485	0.1
Met_MCA (b)	538.3208	5	220.6577491	1.4
Phe_MCA (b)	554.3487	5.85	225.8542763	1.9
Pro_MCA (b)	504.3331	4.5	214.8380734	-2.1
Ser_MCA (b)	494.3123	4.1	209.5826589	-3.1
Thr_MCA (b)	508.328	4.3	214.793656	1.8
Trp_MCA (b)	593.3596	5.6	229.0608493	-0.7
Tyr_MCA (b)	570.3436	4.6	223.9183479	-0.1
Val_MCA (b)	506.3487	5.05	215.7099972	-0.9
Cit_MCA (b)	564.3654	4.15	223.9738506	-1

Orn_MCA (b)	521.3596	4	217.2601936	-0.5
DOPA_MCA (b)	586.3385	4.4	226.4490022	-1.7
Ala_MCA (g)	478.3174	4.9	207.9788803	1
Arg_MCA (g)	563.3814	4.8	224.8751426	1.5
Asn_MCA (g)	521.3232	4.5	216.441312	-3.4
Asp_MCA (g)	522.3072	4	216.4309174	-1.6
Glu_MCA (g)	536.3229	4	217.1056335	1.4
Gln_MCA (g)	535.3389	4.55	219.7949871	2
His_MCA (g)	544.3392	4.65	222.3829841	1.6
Ile_MCA (g)	520.3644	6.6	220.8459414	2
Leu_MCA (g)	520.3644	6.5	219.058363	2
Lys_MCA (g)	535.3753	4.55	219.7946156	1.9
Met_MCA (g)	538.3208	5.9	222.4438043	2
Phe_MCA (g)	554.3487	6.8	225.8542763	1.8
Pro_MCA (g)	504.3331	5.1	212.1545302	-2.6
Ser_MCA (g)	494.3123	4.55	208.6876676	-2.5
Thr_MCA (g)	508.328	4.8	213.8993266	2.1
Trp_MCA (g)	593.3596	6.55	235.2976884	-0.6
Tyr_MCA (g)	570.3436	5.1	230.1608564	0.1
Val_MCA (g)	506.3487	5.8	215.7099972	-0.7
Cit_MCA (g)	564.3654	4.55	225.7578666	-0.9
Orn_MCA (g)	521.3596	4.5	216.4409268	1.4
DOPA_MCA (g)	586.3385	5	224.6665674	-1.3

Supplementary Table 11: Representative ion mobility traces compared to standards on a TIMS-TOF. Provided as separate document. Representative screenshot.

RT, Accurate Mass, DT and MSMS Validation

Ala-DCA in strain P1-E12
HM 1041 *Clostridium cadaveris* CC88A

Ala-DCA from synthetic reaction mixtures

Supplementary Table 12: Representative Lumos mirror plots and Triple TOF MS/MS spectra of the bile acids (two tabs in the table). Provided as separate document. Screenshot below.

Formula	ChemDraw/MSI	DrugBank	PubChem	HM ID	massed_atom_formula	WZ	MS2	MS2 mirror (approx. standard, bottom is a real sample)
<chem>C27H45NO8</chem>				480.331976	478.317424	478.3175	0.1588903	MS2 5753
<chem>C27H45NO8</chem>				480.331976	478.317424	478.3175	0.1588903	MS2 5756
<chem>C27H45NO8</chem>				480.331976	478.317424	478.3175	0.1588903	MS2 5449
<chem>C27H45NO8</chem>				480.331976	478.317424	478.3175	0.1588903	MS2 5448

The authors have included a new class of compounds in the second submission, FAHFAs. FAHFAs are a good example of why imperfect MS/MS matches can be misleading and cause misidentification. FA dimers can form during LC-MS and produce compounds with the same exact mass and highly similar (but not identical) MS/MS data as FAHFAs. It is essential that the MS/MS patterns be exact matches to support the identification of a new compound. These should be included in the paper.

We had already provided the links to all MS/MS matches linked to the npj approved repository where this data is permanently archived, including the MS/MS matches. We have now added PDFs or tables with a selection of these as mirror matches (see description in previous

comments). As noted in the manuscript these are level 3 matches - or molecular family level matches - and therefore cannot differentiate isomers. This was already described in the paper but now have provided an expanded description in the SI. See response to previous comments. In GNPS we have dimers of fatty acids (<https://www.nature.com/articles/s41467-021-23953-9> and <https://pubmed.ncbi.nlm.nih.gov/36156231/>) and none matched to the FAHFA standards.

To see how we responded in the paper - see the text to the previous discussion that was added.

In my last evaluation I suggested that the authors match MS/MS data from their authentic standards to research samples. They did not do that. Instead they used a QQQ instrument to validate and quantify the concentration of the molecules in their own samples. The major issue in using a QQQ is that it can disguise poorly matched MS/MS patterns. What is necessary to establish high confidence in a new compound identification is a full MS/MS match, which should be included in the paper.

QQQ was used for quantification because it is 2-3 orders of magnitude more sensitive. We have provided representative MS/MS matches, RTs and CCS data as PDFs now (see discussion to previous comments).

The authors write "it strikes us as odd that this reviewer is stuck on data collection to be in our own lab vs independent external validation". The problem is that the MS/MS data shown in the last submission did not provide a compelling match between the synthesized standards and the research samples. The only way to convincingly prove that the differences are due to different instruments and workflows is to eliminate the "external" variable by doing the experiments in an identical fashion. If internal MS/MS data, retention time, and collision cross section of the synthesized standard do not exactly match internal MS/MS data, retention time, and collision cross section of the putative compound in the research sample, then it is a different compound.

Thanks for clarifying what this reviewer wanted and needed to see, as we were indeed mystified by the notion that it had to be done in our lab as opposed to having the detection of the bile acids validated outside of our lab and with multiple cohorts. We have now provided representative examples of traces MS/MS, RT and when appropriate CCS values and provided the table summarizing the MS1, RT and CCS values and well as a description for each experiment and what was done and the depth of information that such experiment provides per discussion with the editor. See also our responses to this in previous sections.

The authors write: "To minimize false discoveries....the precursor mass had to be within 0.01 m/z for QE". That is a large m/z range for the QE. Why did they make the error so big?

Thanks for catching this issue, as we can now see how the original text could be misread. The statement "minimize false discoveries" was not in reference to the 0.01 m/z but rather the number of ions and cosine we included, and the citation provides context to this statement. We used those values because they narrow down the search space within a repository of 2600 public projects that has 1.2 billion MS/MS spectra and is used to reduce the search space. This in turn reduces the computational burden for MS/MS matching (at the time this still took days per query on a 3000-node cluster- we have new unpublished algorithms) but this was not the case a few years ago when we performed these experiments).

We therefore rephrased it as "MASST searches were performed using the following requirements. First, the precursor mass was filtered 0.01 or 0.02 m/z for QE and Q-ToF data

respectively, and to minimize false discoveries, the matches required a minimum of 6 fragment ions and cosine score of 0.7 to match.”

They also write: “To minimize false discoveries....[we] required a minimum of 6 fragment ions”. This is a limitation because many metabolites will not have 6 fragment ions.

Computational searches, can be done, with less that 6 ions results but comes at a cost of increasing FDR if the same cosine is kept as has been demonstrated (<https://pubmed.ncbi.nlm.nih.gov/29133785/>). We added the discussion on number of fragment on to the SI extended discussion as part of the response to previous comments.

Regarding the biochemistry part of the manuscript. The positive control (rifampicin) for the dose response is way outside the EC50~500nM-1600nM), it is impossible to determine the actual biological action of these bile acids since the response appears similar to an extremely high dose of rifampicin.

The EC50 for rifampicin in this assay was 2.067 uM. This is consistent with EC50 values of rifampicin determined with PXR reporter assays in other papers:

- 1.2 uM (<https://doi.org/10.1210/me.2006-0323>)
- 3.5 uM (10.1124/dmd.110.035105)
- 0.4 uM (<https://doi.org/10.1016/j.bcp.2008.06.016>)

These experiments are not treating disease, but rather serve as a positive control. It is a very common control, and used at those doses, described in many papers. Here is a representative list of papers using the 50 mg/kg dose.

1. <https://www.ncbi.nlm.nih.gov/pmc/articles/PMC8081023/>
2. <https://www.ncbi.nlm.nih.gov/pmc/articles/PMC7337396/>
3. <https://aacrjournals.org/cancerres/article/69/11/4760/549579/Rifampicin-as-an-Oral-Angiogenesis-Inhibitor>

We therefore did not change this point.

The reporting of the T cell assay is confusing and hard to interpret. Reporting the response to bile acid treatment as the fold induction of IL-17+ CD4 T cells doesn't make sense. These results are typically reported as the percent of IL-17+ or IFN-g+ T cells present after stimulation compared to controls rather than a fold change. What is untreated-naive, unstimulated T cells, stimulated T cells with no bile acids? The normalization is confusing. The manuscript indicates a change in secretion of cytokines but the amount of cytokine secretion wasn't measured only the differentiation of naive T cells into the IL-17+/IFN-g+ CD4+ T cells after stimulation.

These are very standard assays. Different communities report results of these very common assays in different ways, and we personally don't have a preference because they convey the same message about the same data. Here are two examples of papers that report in fold change. <https://www.tandfonline.com/doi/full/10.1080/2162402X.2017.1372081> Figure 3 <https://www.spandidos-publications.com/10.3892/ol.2020.11882> Figure 6.

Here are two examples that report in % <https://www.ncbi.nlm.nih.gov/pmc/articles/PMC7540721/> See Fig 1, 2, and 4 showing % cell populations <https://www.ncbi.nlm.nih.gov/pmc/articles/PMC6949019/> All figures show % cell populations, including cytokines.

As we do not have a preference of the fold change vs % reporting we will leave it to the editor to see what they prefer, as this preference varies among editors and journals.

Reviewer Reports on the Second Revision:

Referees' comments:

Referee #2 (Remarks to the Author):

The authors have addressed the points that I have raised. I was hoping to see strong evidence of 12-dehydroxylation, but this will have to wait. I agree that it is a distraction from what should be a focused manuscript.

Referee #3 (Remarks to the Author):

As I understand it, the concept of 'reverse metabolomics' is to use combinatorial chemistry to synthesize a collection of related authentic standards. Metabolomics data is then collected for these standards and used to discover previously unidentified molecules in biological samples. In this reviewer's perspective, the discovery of previously unidentified molecules of biological relevance is what makes the method exciting. As the authors write: "Until now, many of the compounds included in our synthetic mixtures had rarely, if ever, been reported in biological samples." Without this, the work simply amounts to collecting reference data on synthesized standards which, while important, is not novel or innovative.

The fundamental question is, have the authors convincingly demonstrated that they have identified compounds never before reported in biological samples? I remain skeptical.

In my last review, I made what seemed like a relatively straightforward suggestion to match full high resolution MS/MS data from synthesized standards to full high resolution MS/MS data from research samples. I suggested that the comparison be made by analyzing samples in the same lab using the same methods, as differences in these variables can lead to differences in MS/MS data.

It is unclear to me whether the authors actually did this. Their response to my point was 11 pages and contained many convoluted arguments that were hard to follow or had little relevance to the comment (such as references to other manuscripts in review at Nature that I have no way of reading).

They do provide MS/MS mirror plots, but the resolution in the PDF that I received is so low that most of them cannot be read and I am unable to interpret them. One of the two plots that I can read is Supplementary Table 8. This MS/MS data is not convincing as the fragmentation voltage was not high enough to effectively fragment the precursor molecule. Additionally the mass error in the 339 fragment seems too big to be a match for the high resolution system used.

The authors indicate that I asked them to provide PDFs of the MS/MS data, but I am not sure where this idea came from. **My advice is to provide the MS/MS data for newly identified molecules as .csv files. Alternatively the authors could provide a Github link containing high resolution png images of MS/MS mirror plots.**

The authors write in line 117 that reverse metabolomics provides "annotations of level 2 or 3". Level 2 and level 3 annotations correspond to putatively annotated compounds and putatively characterized compound classes. This evidence falls short of the claim that the authors have discovered new metabolites never reported before in biological samples. They state that they have identified a subset of compounds with level 1 confidence by targeted analysis on line 230. However, as support they reference extended data figure 3b which is quantification of conjugated bile acids in healthy human fecal samples. Quantification is unrelated to confidence level, so I am unclear why the data is called out there. No MS/MS data is provided to justify the level 1 assignment. Although the authors state that they have reached the highest level of confidence as recommended by the metabolomics standards initiative, I do not see the evidence to support such a claim. The suggestion above surrounded by asterisks is what is needed.

In response to my comments, the authors compare their work to newborn screening, but the discussion makes a false equivalency. The goal of newborn screening is quite different from the goal of the authors' paper. Newborn screening is not applied to discover new compounds never reported before in babies. Newborn screening uses methods that have already been validated to be specific to well characterized metabolites. Its goal is quantitative profiling. Similarly, the other papers referenced by the authors are also profiling known metabolites and not claiming to discover previously unreported metabolites.

Finally, to support the claim that reverse metabolomics can discover biologically interesting metabolites, it is important to show relevance. The biological assays (T cells assay) are presented as fold change and not quantitative which seems to be a theme throughout the paper (for example no quant of bile acid derivatives as mentioned by reviewer 2). Induction of a small percentage of IL-17+/IFN-g+ cells may not be physiologically relevant. To understand how strongly the molecules induce this T cell population, the data should be presented as percentages.

Referee #4 (Remarks to the Author):

Summary: I believe that the reviewer is exceptionally cautious in their approach, but that some of their concerns are valid. I interpret the novelty of the idea as the potential to improve identifications by using combinatorial chemistry. Some of your mass spectral comparisons have some differences between the standard and the sample fragmentation data which may suggest that they may be similar but not identical compounds. Phe-Ca and Phe-DCA are examples. This is not so unusual in mass spectrometry and in both of these cases, the fragmentation may be affected by the close elution time of a rival compound. Ideally, for at least a couple of the compounds, you could have used a complementary method e.g. NMR to fully confirm your identities. However, I am surprised, since it is mentioned, that you do not provide the CCS data alongside the fragmentation data (although you provide it in one table, it is not clear whether this applies to both standard and samples and whether there was any difference).

In addition, I would recommend:

Clearly linking all data to the repository where it is kept.

Include with each example exact details on:

Which instrument it was collected on

The mass accuracy and resolution of that instrument (both theoretical, and preferably also as it was measured with SST or equivalent on the day of collection)

Whether the standard was measured on the exact instrument

Collision energy applied

CCS data where applicable

While I agree with the authors that public repositories for data are definitely a better idea than pictures, the request from reviewer 2 for a couple of examples (2-3) to be included in the paper is reasonable and would allow the authors to add extra annotations to explain fragmentation patterns or differences. Such pictures do need to be of high resolution quality to be of use; at the moment, what has been provided is poor quality and could be improved. Clear comparisons of similarities and differences between the standards and samples (fragment ions, intensities etc) would be useful, and giving the accurate masses and preferably also proposed fragment identities would also help. Also a clear statement about whether they were collected on the same instrument in the same batch and at the same collision energy. You may also consider including the fragmentation patterns of any isomers (if known) for comparison.

Additional comments (in response to referee #3 concerns):

The fundamental question is, have the authors convincingly demonstrated that they have identified compounds never before reported in biological samples? I remain skeptical.

I am less sceptical, but I agree that the reviewer has some justified concerns about the quality of the evidence. The plots provided as pdfs are poor quality and some basic information is missing, such as instrumentation (including theoretical and measured mass accuracy and actual measured resolution) and fragmentation energy and type applied. However, better quality plots are available in their linked data storage (although still lacking basic information in a human-readable format). Some of the fragmentation patterns are not totally identical between standard and biological sample, but this is not unusual in mass spectrometry. However, it does allow an element of doubt about the identifications in some cases. Some labelling of the exact masses (as measured) and the theoretical masses based on the theoretical fragment ions would help here. I would also recommend that (where they exist), the fragment patterns of known isomers could also be shown for comparison. It is sufficient to do this for one or two compounds since this is very time consuming.

In my last review, I made what seemed like a relatively straightforward suggestion to match full high resolution MS/MS data from synthesized standards to full high resolution MS/MS data from research samples. I suggested that the comparison be made by analyzing samples in the same lab using the same methods, as differences in these variables can lead to differences in MS/MS data.

I agree with the reviewer that having the standards measured on the same instrument with the same method as some biological standards is a reasonable level of evidence to ask for. My understanding is that this is what the reviewers have done and describe in Line 413 onwards. A few lines clarifying this in the results section would be useful.

It is unclear to me whether the authors actually did this. Their response to my point was 11 pages and contained many convoluted arguments that were hard to follow or had little relevance to the comment (such as references to other manuscripts in review at Nature that I have no way of reading).

They do provide MS/MS mirror plots, but the resolution in the PDF that I received is so low that most of them cannot be read and I am unable to interpret them. One of the two plots that I can read is Supplementary Table 8. This MS/MS data is not convincing as the fragmentation voltage was not high enough to effectively fragment the precursor molecule. Additionally the mass error in the 339 fragment seems too big to be a match for the high resolution system used.

I am not sure which 339 fragment the reviewer is referring to. However, I also note that for some of the compounds, the mirror MS/MS plots are not absolutely identical. Phe-Ca and Phe-DCA as examples. This is not so unusual in mass spectrometry and in both of these cases, the fragmentation may be affected by the close elution time of a rival compound. I am surprised that the authors have not included the CCS evidence alongside these plots since this would be more conclusive. I agree with the reviewer that the mirror plots here are small and difficult to read; many of the numerical labels have been cut off the diagrams. It is not clear where they are stored in the repository, so I was unable to check this version of the data.

The authors indicate that I asked them to provide PDFs of the MS/MS data, but I am not sure where this idea came from. ****My advice is to provide the MS/MS data for newly identified molecules as .csv files. Alternatively the authors could provide a Github link containing high resolution png images of MS/MS mirror plots.****

The authors do seem to have provided a substantial number of plots in a GNPS file.

The authors write in line 117 that reverse metabolomics provides "annotations of level 2 or 3". Level 2 and level 3 annotations correspond to putatively annotated compounds and putatively characterized compound classes. This evidence falls short of the claim that the authors have discovered new metabolites never reported before in biological samples. They state that they have identified a subset of compounds with level 1 confidence by targeted analysis on line 230. However, as support they reference extended data figure 3b which is quantification of conjugated bile acids in healthy human fecal samples. Quantification is unrelated to confidence level, so I am unclear why the data is called out there. No MS/MS data is provided to justify the level 1 assignment. Although the authors state that they have reached the highest level of confidence as recommended by the metabolomics standards initiative, I do not see the evidence to support such a claim. The suggestion above surrounded by asterisks is what is needed.

It is a challenge to provide level 1 confidence data. However, I agree with the reviewer that extended data Fig 3b does not really justify their claim. Instead, I would anticipate the authors

would provide raw mass spec, chromatographic data and CCS data of their synthesised standards and their presence in their pooled samples. Since we have also discovered that targeted methods can misidentify compounds due to in-source fragmentation, ideally, NMR or chiral column data would also be provided, but I think this is an expensive ask and I would not typically demand it unless the biological claims in the paper were very dependent on an absolute identification.

Finally, to support the claim that reverse metabolomics can discover biologically interesting metabolites, it is important to show relevance. The biological assays (T cells assay) are presented as fold change and not quantitative which seems to be a theme throughout the paper (for example no quant of bile acid derivatives as mentioned by reviewer 2). Induction of a small percentage of IL-17+/IFN-g+ cells may not be physiologically relevant. To understand how strongly the molecules induce this T cell population, the data should be presented as percentages.

Quantitative data is always preferable, but in this case, would very likely require several months, if not years of extra work. I am not sure it is sufficiently warranted to justify this.

Author Rebuttals to Second Revision:

Referee expertise:

Referee #3 (Remarks to the Author):

As I understand it, the concept of 'reverse metabolomics' is to use combinatorial chemistry to synthesize a collection of related authentic standards. Metabolomics data is then collected for these standards and used to discover previously unidentified molecules in biological samples. In this reviewer's perspective, the discovery of previously unidentified molecules of biological relevance is what makes the method exciting.

We thank the reviewer for spending the time to do another round of review, and for the clear understanding of the exciting component of the method, i.e. that it enables discovery of previously unknown molecules of biological relevance.

As the authors write: "Until now, many of the compounds included in our synthetic mixtures had rarely, if ever, been reported in biological samples." Without this, the work simply amounts to collecting reference data on synthesized standards which, while important, is not novel or innovative.

We disagree, in that the present work provides a repository-scale analysis that has not previously been possible. This aspect of the work is novel, innovative and enabled the discovery of hundreds of new human-derived molecules, including many specifically associated with IBD (although we acknowledge that substantial additional work will be required to isolate them for NMR analysis for these molecules, beyond the scope of the present paper). We therefore see the present work as laying the groundwork for a large part of the future of metabolomics.

The fundamental question is, have the authors convincingly demonstrated that they have identified compounds never before reported in biological samples? I remain skeptical.

We note that this skepticism is not shared by Ref 4 or by the four independent mass spec labs that performed the work represented in this paper. A typical process in annotating unknowns in metabolomics is to predict molecules, then synthesize them to support structure elucidation. Specifically, the predictions are matched to synthetic standards (or commercial standards) and match RT, IMS drift time if available, and MS/MS for annotations. Here we do match directly to synthetic standards that we have confirmed by NMR, and we have moved some of this information into the main text from the SI as detailed below in the response to Ref 4.

Because this paper has been in review for multiple years, and because we have an open data policy where we share data with the scientific community before publication, others

have used our data to “discover molecules” in their data set using the MS/MS reference data from this paper. One published example is: <https://www.nature.com/articles/s41586-023-05989-7>. Another is <https://www.nature.com/articles/s42255-023-00777-z> – because our reference spectra in GNPS are exchanged with MONA, anyone using MONA (or using MZMine 3 or MS-Dial) can see matches to our reference spectra in their data. The present work contains far more evidence than anything documented in either of those papers, because they use the data in our present work to support their findings. Another paper that this team contribute to is <https://www.nature.com/articles/s41564-023-01337-7>, which has leveraged the synthetic library we created in this paper for the initial annotations. We now also know the enzymes involved (<https://www.researchsquare.com/article/rs-2050406/v1> and <https://www.researchsquare.com/article/rs-2050120/v1>). Thus, not only are we providing a resource, but others in the field are already confirming our structural annotations and discoveries and publishing on these findings.

Additionally, other papers, including several in Nature, use this synthesis approach for the same classes of molecules and more in their work. But they do not do reverse metabolomics – which provides the ability to search 1.2 billion spectra. Other labs are also already confirming the newly discovered metabolites from preprints of our paper, for example <https://www.medrxiv.org/content/10.1101/2023.05.17.23290088v1>, underscoring the impact of the work and its reproducibility.

In my last review, I made what seemed like a relatively straightforward suggestion to match full high resolution MS/MS data from synthesized standards to full high resolution MS/MS data from research samples. I suggested that the comparison be made by analyzing samples in the same lab using the same methods, as differences in these variables can lead to differences in MS/MS data.

We did provide a selection of these (the complete set of all matches would be hundreds of thousands of pages of data, as we noted). We also note that this reviewer did not want the data to be linked electronically in previous reviews on the assumption that we could track who accessed the data – which we actually cannot. However, we have now followed the suggestion by Ref 4 to show representative examples in the paper which we have now added, and then clearly state where the data is housed in public repositories. We hope this addresses this concern in a satisfactory way.

It is unclear to me whether the authors actually did this. Their response to my point was 11 pages and contained many convoluted arguments that were hard to follow or had little relevance to the comment (such as references to other manuscripts in review at Nature that I have no way of reading).

We apologize if the links did not work for the reviewer for some reason, but all these papers are publicly available as preprints. For example, the links in the section reproduced from the previous review below all worked when we tried them.

“For example, demonstrating that a single dual function enzyme (BSH) can produce the new conjugates from the taurine conjugates and free acids (<https://www.researchsquare.com/article/rs-2050406/v1>, <https://www.researchsquare.com/article/rs-2050120/v1>, and others).” A third paper used the

standards created in this work to show the impact of BSH on bile acid production and germination of *C.diff* (see <https://www.nature.com/articles/s41564-023-01337-7>)." This paper is now published <https://www.nature.com/articles/s41564-023-01337-7>.

They do provide MS/MS mirror plots, but the resolution in the PDF that I received is so low that most of them cannot be read and I am unable to interpret them. One of the two plots that I can read is Supplementary Table 8. This MS/MS data is not convincing as the fragmentation voltage was not high enough to effectively fragment the precursor molecule. Additionally the mass error in the 339 fragment seems too big to be a match for the high resolution system used.

We are not sure what figure the reviewer refers to here, but this fragment is a key ion of a dehydroxylated bile acid. We hope that the improved plots in response to Ref 4 adequately address this issue in general.

The authors indicate that I asked them to provide PDFs of the MS/MS data, but I am not sure where this idea came from. **My advice is to provide the MS/MS data for newly identified molecules as .csv files. Alternatively the authors could provide a Github link containing high resolution png images of MS/MS mirror plots.**

We apologize for the lack of clarity in our prior response: the issue is more about the volume of data than the specific format. All data are in the public domain – either in metabolomics workbench or Massive - and both are npj approved repositories. The reviewer, previously, did not accept the dashboard links, which are links directly from the repositories. All matches from GNPS are linked and every single mirror plot from the MASST searches can be examined without revealing the reviewer's identity. However, we have provided examples in the main text in response to Ref. 4's request below.

The authors write in line 117 that reverse metabolomics provides "annotations of level 2 or 3". Level 2 and level 3 annotations correspond to putatively annotated compounds and putatively characterized compound classes. This evidence falls short of the claim that the authors have discovered new metabolites never reported before in biological samples. They state that they have identified a subset of compounds with level 1 confidence by targeted analysis on line 230. However, as support they reference extended data figure 3b which is quantification of conjugated bile acids in healthy human fecal samples. Quantification is unrelated to confidence level, so I am unclear why the data is called out there. No MS/MS data is provided to justify the level 1 assignment. Although the authors state that they have reached the highest level of confidence as recommended by the metabolomics standards initiative, I do not see the evidence to support such a claim. The suggestion above surrounded by asterisks is what is needed.

We agree with the point that quantification is unrelated to confidence level, and had meant to delete this in the last round. We appreciate the detection of this error on our part. Ref 4 asked to remove the discussion for level 1, even for RT, drift time and MS/MS matched examples and we have done this as requested. We hope that the examples Ref 4 requested that we add to the main text also address this concern (see response below).

In response to my comments, the authors compare their work to newborn screening, but the discussion makes a false equivalency. The goal of newborn screening is quite different from the goal of the authors' paper. Newborn screening is not applied to discover new compounds never reported before in babies. Newborn screening uses methods that have already been validated to be specific to well characterized metabolites. Its goal is quantitative profiling. Similarly, the other papers referenced by the authors are also profiling known metabolites and not claiming to discover previously unreported metabolites.

We apologize for the lack of clarity. The quantification was done after we had discovered the compounds in the human samples. At this point, the procedure is identical to newborn screening using standards as reference.

Finally, to support the claim that reverse metabolomics can discover biologically interesting metabolites, it is important to show relevance. The biological assays (T cells assay) are presented as fold change and not quantitative which seems to be a theme throughout the paper (for example no quant of bile acid derivatives as mentioned by reviewer 2). Induction of a small percentage of IL-17+/IFN-g+ cells may not be physiologically relevant. To understand how strongly the molecules induce this T cell population, the data should be presented as percentages.

We are puzzled by this comment because we did change the presentation of these data to percentages in response to the last round of review. We note that whether to present this type of data as percentages or in other ways varies by field. We also note that Ref 2 is now satisfied with our responses.

Referee #4: metabolomics, MS

Referees' comments:

Referee #4 (Remarks to the Author):

Summary: I believe that the reviewer is exceptionally cautious in their approach, but that some of their concerns are valid. I interpret the novelty of the idea as the potential to improve identifications by using combinatorial chemistry. Some of your mass spectral comparisons have some differences between the standard and the sample fragmentation data which may suggest that they may be similar but not identical compounds. Phe-Ca and Phe-DCA are examples. This is not so unusual in mass spectrometry and in both of these cases, the fragmentation may be affected by the close elution time of a rival compound. Ideally, for at least a couple of the compounds, you could have used a complementary method e.g. NMR to fully confirm your identities. However, I am surprised, since it is mentioned, that you do not provide the CCS data alongside the fragmentation data (although you provide it in one table, it is not clear whether this applies to both standard and samples and whether there was any difference).

We thank the reviewer for the clear guidance and for the detailed and useful review. We agree with all comments and have addressed them in detail in the responses below.

In addition, I would recommend:

Clearly linking all data to the repository where it is kept.

We appreciate this suggestion, and have now provided descriptions for each data set and instrument on which it was collected in the data availability section, and the methods. We have also added the accession number beneath each of the figure legends in the paper.

Which instrument it was collected on

We have specified the instrument, ionization mode and accession numbers associated with each experiment contributing data to each figure.

The mass accuracy and resolution of that instrument (both theoretical, and preferably also as it was measured with SST or equivalent on the day of collection)

We had provided the information regarding comparison with standards to what was experimentally observed in samples as SI tables 6-8 (in the fourth revision this was table 7-9). In those tables we provide predicted m/z, observed m/z, ppm errors, RT of standard and RT observed in the sample. We have added a new SI table 9 that provides errors in ppm. In addition, the links in GNPS from MASST or molecular networking searches from the tables or the manuscript link out to the reference library information (which also gives theoretical mass based on the structure) and the observed mass. GNPS also gives direct provenance to each spectral annotation and the MS/MS match of the standard and provides mass difference, ppm difference, instrument info, origin of the MS/MS that it is matching against, TIC, RT, number of MS/MS ions that match in culture extracts and hyperlinks to mirror matches of MS/MS of standard and from the sample (this can be seen, for example, by loading the record for accession MSV000084475), etc. See representative link for DCA amidates.

https://gnps.ucsd.edu/ProteoSAFe/result.jsp?task=1e07d6df47924aec951739fb3695b44a&view=view_all_annotations_DB#%7B%22main.Compound Name input%22%3A%22-DCA%22%2C%22table sort history%22%3A%22main.MQScore dsc%22%7D.

We made sure that the headers of each table were clear (see example for Table 7 below). If the table headers are still not sufficiently clear with these revisions, we would appreciate further guidance to make it accessible to the broadest possible audience.

Conjugated bile acids found in HMP bacterial cultures

Bile Acid	predicted m/z [M+H] ⁺	observed m/z [M+H] ⁺	abs. ppm diff	standard RT (min)	observed RT (min)
Ala-CA	480.3321	480.3326	1.04	3.9	3.9
Ala-DCA	464.3371	464.3375	0.86	4.8	4.8
Arg-CA	565.3961	565.3962	0.18	3.1	3.1

Whether the standard was measured on the exact instrument

All new data that was collected (i.e. results that did not come from mining public data) was compared to standards within the same experiment on the same instrument. We have now added this information specifically to the methods, stating that data on standards was collected in the same experiment and instrument in each description of the method where appropriate.

Collision energy applied.

We left this information in the methods section but realized this was missing for the QToF, and have now added the table below in the methods (as supplementary table 10) because methods cannot have display items. We apologize for the omission.

Isolation mass value (m/z)	Charge state value (z)	Isolation width value (m/z)	Collision energy value (eV)
100	1	4	22
300	1	5	27
500	1	6	35
1000	1	8	45
2000	1	10	50
100	2	4	18
300	2	5	22
500	2	6	30
1000	2	8	35
2000	2	10	50

CCS data where applicable.

This is only relevant for the microbial culture data. We have provided a specific example in figure 4 showing the drift time of the standard and sample. Furthermore, the results from the LC-IMS-MS experiment are summarized in supplementary table 5 for the standards vs. pooled culture extracts, and supplementary table 9 for all observed bile acids in each of the microbial cultures (not pooled). We also provided a better description of how the standard data was used in the methods section (pasted below).

“In order to consider a MCBA from the HMP samples a match to the synthetic library, mass accuracy had to be within less than 10 ppm (although most were within 5ppm), idotp had to be greater than 0.8 (isotopic distribution match to the MF), RT within +/- 0.1 min, and IMS drift time had to fall within the window set in Skyline for an IMS resolving power of 40.”

While I agree with the authors that public repositories for data are definitely a better idea than pictures, the request from reviewer 2 for a couple of examples (2-3) to be

included in the paper is reasonable and would allow the authors to add extra annotations to explain fragmentation patterns or differences. Such pictures do need to be of high resolution quality to be of use; at the moment, what has been provided is poor quality and could be improved. Clear comparisons of similarities and differences between the standards and samples (fragment ions, intensities etc) would be useful, and giving the accurate masses and preferably also proposed fragment identities would also help.

We greatly appreciate the suggestion to provide representative examples in the main figures. Therefore, we have now moved figures from previous supporting tables to the manuscript.

To figure 1, we added the following MS/MS comparisons:

We added 5 examples to Figure 3:

To figure 4, we added the comparison of standards to pooled culture extract and a representative MS/MS match:

To figure 4, we also added an example of drift time and RT of standard vs sample:

We have now also added representative examples that explain key diagnostic fragment ions as a supporting figure 1:

Also a clear statement about whether they were collected on the same instrument in the same batch and at the same collision energy. You may also consider including the fragmentation patterns of any isomers (if known) for comparison.

We have added such a statement to the methods section, and in Figure 3a we added two examples of isomers. In one of these pairs, the retention time is different but the MS/MS

spectra looks very similar. In the other pair, the retention time and MS/MS spectra are the same and thus they cannot be distinguished.

The text now reads, “Most isomers such as Glu-CDCA and Glu-DCA could be resolved by chromatographic separation (Figure 3a), but in a few cases, isomers were not separated by chromatography, such as the Met-CDCA or DCA and Tyr-HDCA or UDCA. In such cases both names are indicated (Figure 3a,b).”

Additional comments (in response to referee #3 concerns):

The fundamental question is, have the authors convincingly demonstrated that they have identified compounds never before reported in biological samples? I remain skeptical.

I am less sceptical, but I agree that the reviewer has some justified concerns about the quality of the evidence. The plots provided as pdfs are poor quality and some basic information is missing, such as instrumentation (including theoretical and measured mass accuracy and actual measured resolution) and fragmentation energy and type applied. However, better quality plots are available in their linked data storage (although still lacking basic information in a human-readable format). Some of the fragmentation patterns are not totally identical between standard and biological sample, but this is not unusual in mass spectrometry. However, it does allow an element of doubt about the identifications in some cases. Some labelling of the exact masses (as measured) and the theoretical masses based on the theoretical fragment ions would help here. I would also recommend that (where they exist), the fragment patterns of known isomers could also be shown for comparison. It is sufficient to do this for one or two compounds since this is very time consuming.

We appreciate that this reviewer understands the realities of mass spec, where fragmentation patterns can vary and it is therefore necessary to accept some degree of ambiguity. We appreciate the suggestion to label exact masses and measured ones, and to move this information to the main text rather than leaving them in the SI. We have now provided this information as part of Figure 1, 3 and 4 and in Figure 3a we included examples of isomers (see response above).

In my last review, I made what seemed like a relatively straightforward suggestion to match full high resolution MS/MS data from synthesized standards to full high resolution MS/MS data from research samples. I suggested that the comparison be made by analyzing samples in the same lab using the same methods, as differences in these variables can lead to differences in MS/MS data.

I agree with the reviewer that having the standards measured on the same instrument with the same method as some biological standards is a reasonable level of evidence

to ask for. My understanding is that this is what the reviewers have done and describe in Line 413 onwards. A few lines clarifying this in the results section would be useful.

Reviewer 4's interpretation is correct, and we describe this starting on line 278 and again on line 413 where we validate the results, using the standards we created, against human cohorts in the Xavier lab at the Broad institute, the Patterson lab at Penn State, and the Baker lab at UNC, in addition to the data obtained in the Dorrestein lab at UCSD. We shipped all labs the standards, and then they compared to their samples in the same experiments on the same instrumentation. We agree this is the approach that is necessary to yield reasonable evidence, and it is exactly what was done. We have now explicitly stated in the methods sections that standards are directly compared in the same experiment. The main text now reads "Fecal extracts were then pooled, and analyzed with the mixed bile acid standards to match retention times and MS/MS spectra in the same experiment and on the same instrument (Figure 3a)." and then again " To further characterize bacterial production of conjugated bile acids, we used LC coupled to ion mobility spectrometry-MS, which enables the matching of the precursor m/z, MS/MS, retention time, and drift time against standards run on the same instrument within the same experiment (Figure 4d,e, Supporting Table 5, and 9)." Together with the addition of the above figures in response to the previous comments, we believe we have substantially improved the clarity of this information.

It is unclear to me whether the authors actually did this. Their response to my point was 11 pages and contained many convoluted arguments that were hard to follow or had little relevance to the comment (such as references to other manuscripts in review at Nature that I have no way of reading).

They do provide MS/MS mirror plots, but the resolution in the PDF that I received is so low that most of them cannot be read and I am unable to interpret them. One of the two plots that I can read is Supplementary Table 8. This MS/MS data is not convincing as the fragmentation voltage was not high enough to effectively fragment the precursor molecule. Additionally the mass error in the 339 fragment seems too big to be a match for the high resolution system used.

I am not sure which 339 fragment the reviewer is referring to.

We are also not sure what figure is referenced here, but this is a diagnostic ion for di-hydroxylated or hydroxylated-mono-ketone bile acids and bile amidates when subjected to thermal activation in positive ionization mode.

However, I also note that for some of the compounds, the mirror MS/MS plots are not absolutely identical. Phe-Ca and Phe-DCA as examples. This is not so unusual in

mass spectrometry and in both of these cases, the fragmentation may be affected by the close elution time of a rival compound.

We fully agree and appreciate that this reviewer understands the realities of doing mass spectrometry, especially at the level of MS/MS comparisons at the repository scale involving tens of thousands of matches from complex samples.

We hope that the movement of the examples to the main figures and the explanation of key fragment ions in the extended data Fig 1 helps (see response to previous comment) provides a clearer explanation.

I am surprised that the authors have not included the CCS evidence alongside these plots since this would be more conclusive.

To clarify, the only experiment where we used ion mobility was for microbial cultures (carried out by the Baker lab). All other experiments were done in labs that did not have ion mobility and therefore were LC-MS only. To ensure it is understood what experiment was done in each figure, as discussed above, we have clarified in the figure legends what instrument and ionization mode was used. We have now also added a drift time example to Figure 4 (see response above). SI Table 5 (originally SI table 10) has the summary for the standards and then SI table 9 (this is a new table added) has all the results from Skyline for each sample.

I agree with the reviewer that the mirror plots here are small and difficult to read; many of the numerical labels have been cut off the diagrams. It is not clear where they are stored in the repository, so I was unable to check this version of the data.

Thanks for pointing out the lack of clarity regarding where each data set is stored in the repository. We have added examples to the paper, as noted above, which we hope improve the presentation. We have also now provided a description of these points in the data availability sections, and each figure legend provides the accession number to the data associated with the record, so there is no ambiguity.

The authors indicate that I asked them to provide PDFs of the MS/MS data, but I am not sure where this idea came from. **My advice is to provide the MS/MS data for newly identified molecules as .csv files. Alternatively the authors could provide a Github link containing high resolution png images of MS/MS mirror plots.**

The authors do seem to have provided a substantial number of plots in a GNPS file.

We have indeed provided all the interactive links for the reverse metabolomics data.

The authors write in line 117 that reverse metabolomics provides "annotations of level 2 or 3". Level 2 and level 3 annotations correspond to putatively annotated compounds and putatively characterized compound classes. This evidence falls short of the claim that the authors have discovered new metabolites never reported before in biological samples. They state that they have identified a subset of compounds with level 1 confidence by targeted analysis on line 230. However, as support they reference extended data figure 3b which is quantification of conjugated bile acids in healthy human fecal samples. Quantification is unrelated to confidence level, so I am unclear why the data is called out there. No MS/MS data is provided to justify the level 1 assignment. Although the authors state that they have reached the highest level of confidence as recommended by the metabolomics standards initiative, I do not see the evidence to support such a claim. The suggestion above surrounded by asterisks is what is needed.

It is a challenge to provide level 1 confidence data. However, I agree with the reviewers that extended data Fig 3b does not really justify their claim. Instead, I would anticipate the authors would provide raw mass spec, chromatographic data and CCS data of their synthesised standards and their presence in their pooled samples. Since we have also discovered that targeted methods can misidentify compounds due to in-source fragmentation, ideally, NMR or chiral column data would also be provided, but I think this is an expensive ask and I would not typically demand it unless the biological claims in the paper were very dependent on an absolute identification.

We thank the reviewer for this assessment, and for the clear guidance as to what to do. We have removed level 1 from the following sentence. "However, targeted analysis was performed to confirm the presence of these bile acids in human fecal samples and to determine the absolute concentration of various conjugated and unconjugated bile acids representing level 1 (Extended Data Fig. 3b)." It now reads "However, targeted analysis was performed to determine the absolute concentration of various conjugated and unconjugated bile acids (Extended Data Fig. 3b)." We have further removed level 1 discussion from the rest of the paper and SI, and instead speak in more general terms about support for the annotation by comparing it to data obtained from standards within the same experiment.

We have now provided examples of key fragments in Supplementary Figure 1. Our goal in this paper was to provide as high a level of a confidence to their existence as possible. That is why we sent the bile acid standards to 4 different mass spec labs to see if they could also observe matches in their samples using their instruments - all labs are using different gradients and approaches. All raw data accessions are provided and linked in the paper, but to make this more clear we have now added the accession number to each data set in the legend for each figure. We also had provided the chromatographic and drift time data for the microbial culture data in SI table 5 (standards) and 9 (samples), for all the features that fit the following criteria: mass accuracy within 10 ppm (although most were within 5ppm),

idotp greater than 0.8 (isotopic distribution match to the MF), RT within +/- 0.1 min, and IMS drift time within the window set in Skyline for an IMS resolving power of 40. We really liked the suggestion by the reviewer to add a few examples to the main paper so people can evaluate how the data was matched to standards and from samples. See responses to the previous comments above.

Regarding NMR, no isolation and purification from human samples was performed. NMR's of the pure standards from the Dorrestein lab were provided, and we have now added NMR data of the pure standards synthesized in the Patterson lab that were used for quantification of bile acids in human fecal samples. These data can be found in the supporting information.

We have now also added the following sentence to the limitations section of the paper "Although our validation experiments used biological samples (human fecal and/or cultured microbial data) to match to the chromatographic retention times, MS/MS and drift time - where available- of the standards, to support the structural annotation, without isolation and NMR or X-ray structural analysis one cannot exclude the possibility that different isomers are represented by the data".

Finally, to support the claim that reverse metabolomics can discover biologically interesting metabolites, it is important to show relevance. The biological assays (T cells assay) are presented as fold change and not quantitative which seems to be a theme throughout the paper (for example no quant of bile acid derivatives as mentioned by reviewer 2). Induction of a small percentage of IL-17+/IFN-g+ cells may not be physiologically relevant. To understand how strongly the molecules induce this T cell population, the data should be presented as percentages.

Quantitative data is always preferable, but in this case, would very likely require several months, if not years of extra work. I am not sure it is sufficiently warranted to justify this.

Thanks for understanding and appreciating the time and cost associated with the quantitation. We agree with the reviewer that this large amount of time and cost is not justified in support of the present paper.

To respond to Ref 3's comment, in the last round of reviews we had changed the figure to % change and left it in this format for this round based on Ref 3's preference. Because we do not have a preference between reporting fold change or %, we leave this choice to the editor, because this preference varies among reviewers, editors and journals.

Reviewer Reports on the Third Revision:

Referees' comments:

Referee #3 (Remarks to the Author):

My primary concern has been the authors claim of identifying new compounds with the highest level of confidence possible, a concern that reviewer 4 shared. I agree that a suitable path forward is to remove this claim and present assignments at only level 2 or level 3 confidence.

Referee #4 (Remarks to the Author):

Very original approach and I am satisfied with the manuscript in the current form. The authors have addressed most of my comments.

I would mention two issues that they may wish to look at but which should not necessitate a further round of peer review:

- 1) There seemed to be a mix up between the figure names, as recorded on the Nature review system, and the file names for individual figures, especially for extended figures. Please check that figures are in their correct order.
- 2) Line 233 would recommend you review wording here: I think you mean to say "pooled faecal samples were analysed as part of the same analytical batch as the mixed bile acid standards" or similar. My initial reading was that you had pooled the faecal samples and the bile acid standards together.

Author Rebuttals to Third Revision:

Referee #3 (Remarks to the Author):

My primary concern has been the authors claim of identifying new compounds with the highest level of confidence possible, a concern that reviewer 4 shared. I agree that a suitable path forward is to remove this claim and present assignments at only level 2 or level 3 confidence.

Thanks for your comments. We are glad to have found a suitable path forward.

Referee #4 (Remarks to the Author):

Very original approach and I am satisfied with the manuscript in the current form. The authors have addressed most of my comments.

Thanks very much for your comments as they have been critical in getting our manuscript to its current form.

I would mention two issues that they may wish to look at but which should not necessitate a further round of peer review:

- 1) There seemed to be a mix up between the figure names, as recorded on the Nature review system, and the file names for individual figures, especially for extended figures. Please check that figures are in their correct order.

We apologize for this confusion and have corrected this mix up in the final submission.

2) Line 233 would recommend you review wording here: I think you mean to say “pooled faecal samples were analysed as part of the same analytical batch as the mixed bile acid standards” or similar. My initial reading was that you had pooled the faecal samples and the bile acid standards together.

The sentence has been reworded to “Fecal extracts were then analyzed as part of the same analytical batch as the bile acid standard mixtures to match retention times and MS/MS spectra.”
Thanks for this feedback.